# rRNA biogenesis regulates mouse 2C-like state by 3D structure reorganization of peri-nucleolar heterochromatin

Hua Yu[1,2,3,4,10], Zhen Sun[1,2,3,4,10], Tianyu Tan[1,2,3,4,10], Hongru Pan[1,2,3,4,10], Jing Zhao [1,2,3,4], Ling Zhang[1,2,3,4], Jiayu Chen [5], Anhua Lei[1,2,3,4], Yuqing Zhu[1,2,3,4], Lang Chen [1,2,3,4], Yuyan Xu[1,2,3,4], Yaxin Liu[1], Ming Chen[6], Jinghao Sheng[7], Zhengping Xu [7], Pengxu Qian[1,2,3], Cheng Li [8], Shaorong Gao [5], George Q. Daley [9] & Jin Zhang [1,2,3,4 ✉]

The nucleolus is the organelle for ribosome biogenesis and sensing various types of stress. However, its role in regulating stem cell fate remains unclear. Here, we present evidence that nucleolar stress induced by interfering rRNA biogenesis can drive the 2-cell stage embryo-like (2C-like) program and induce an expanded 2C-like cell population in mouse embryonic stem (mES) cells. Mechanistically, nucleolar integrity maintains normal liquid-liquid phase separation (LLPS) of the nucleolus and the formation of peri-nucleolar heterochromatin (PNH). Upon defects in rRNA biogenesis, the natural state of nucleolus LLPS is disrupted, causing dissociation of the NCL/TRIM28 complex from PNH and changes in epigenetic state and reorganization of the 3D structure of PNH, which leads to release of Dux, a 2C program transcription factor, from PNH to activate a 2C-like program. Correspondingly, embryos with rRNA biogenesis defect are unable to develop from 2-cell (2C) to 4-cell embryos, with delayed repression of 2C/ERV genes and a transcriptome skewed toward earlier cleavage embryo signatures. Our results highlight that rRNA-mediated nucleolar integrity and 3D structure reshaping of the PNH compartment regulates the fate transition of mES cells to 2C-like cells, and that rRNA biogenesis is a critical regulator during the 2-cell to 4-cell transition of murine pre-implantation embryo development.

[1] Center of Stem Cell and Regenerative Medicine, Department of Basic Medical Sciences, and Bone Marrow Transplantation Center of the First Affiliated Hospital, Zhejiang University School of Medicine, 310003 Hangzhou, China. [2] Zhejiang Laboratory for Systems & Precision Medicine, Zhejiang University Medical Center, 1369 West Wenyi Road, 311121 Hangzhou, China. [3] Institute of Hematology, Zhejiang University, 310058 Hangzhou, China. [4] Center of Gene/Cell Engineering and Genome Medicine, 310058 Hangzhou, Zhejiang, China. [5] Clinical and Translational Research Center of Shanghai First Maternity and Infant Hospital, Shanghai Key Laboratory of Signaling and Disease Research, School of Life Sciences and Technology, Tongji University, Shanghai, China. [6] College of Life Sciences, Zhejiang University, 310058 Hangzhou, China. [7] Institute of Environmental Medicine, and Cancer Center of the First Affiliated Hospital, Zhejiang University School of Medicine, 310058 Hangzhou, China. [8] Center for Bioinformatics, School of Life Sciences, Center for Statistical Science, Peking University, 100871 Beijing, China. [9] Stem Cell Transplantation Program, Division of Pediatric Hematology Oncology, Boston Children's Hospital, Department of Biological Chemistry and Molecular Pharmacology, Harvard Medical School, Boston, MA, USA. [10] These authors contributed equally: Hua Yu, Zhen Sun, Tianyu Tan, Hongru Pan. ✉email: zhgene@zju.edu.cn

Two-cell (2C) stage embryonic cells are totipotent cells in an earlier stage of embryo development and can generate all cell types of embryonic and extraembryonic tissues. In the culture of mouse embryonic stem (mES) cells, a rare population of mES cells sporadically transit into a 2C stage embryo-like (2C-like) cells with similar molecular features of totipotent 2C-stage embryos[1–3]. Recent works have demonstrated that the conversion of mES cells to 2C-like cells is regulated by a variety of factors related to epigenetic modification, including histone methylation and acetylation[1,4,5] and DNA methylation[6,7]. In addition, it was also found that RNA hydroxymethylation[8] and protein sumoylation[9] can affect the epigenetic state of chromatin to regulate the activation of ERV genes. The structure of chromatin is an important epigenetic factor and is closely related to the regulation of gene expression and cell fate transition. Interestingly, chromatin structure appears to emerge as an important factor in 2C gene regulation and transition of mES cells to 2C-like cells. For instance, 2C/ERV gene activation is regulated by a pioneer transcription factor DUX which increases chromatin accessibility[10–15], and two recent studies reported that the pluripotency factors DAPP2 and DAPP4 and the maternal factor NELFA can bind to the promoter region of *Dux* and directly trans-activate its expression[16,17]. Moreover, 2C-like cells can be induced by downregulation of chromatin remodeling factor CAF-1[18]. However, the molecular players of chromatin structure in mES cell to 2C-like cell transition have yet to be fully understood.

With the emergence and development of high-throughput chromatin conformation capture technology (Hi-C), the dynamic changes of higher-order chromatin structures during early embryonic development and stem cell differentiation have been elucidated[19–22]. Two-cell embryo or 2C-like cells show contrasting differences from inner cell mass or ES cells[21,23], suggesting that the 3D chromatin structure is a key factor mediating the transition of mES cells to 2C-like cells. Importantly, one of the mechanisms of *Dux* expression is dependent on a complex of nucleolin NCL and heterochromatin factor TRIM28 in the Peri-Nucleolar Heterochromatin (PNH) region[14,24,25]. However, it is not completely known how nucleolus integrity influences higher-order chromatin structure and how the chromatin structure determines *Dux* expression.

Here, we find that inhibition of nucleolar rRNA biogenesis triggered nucleolar stress, which activated 2C-like transcriptional program and induced an expanded 2C-like cell population in mES cells with a mechanism involving 3D structure reorganization of the PNH and the *Dux* expression. Consistently, the 3D structure of the PNH reorganizes after early 2-cell during murine early embryo development, which coincides with rRNA biogenesis and *Dux* repression. Moreover, in mouse early embryos, rRNA biogenesis and matured nucleolus are indispensable for the 2-cell to 4-cell transition. Taken together, our findings provide a mechanistic perspective of rRNA biogenesis in regulating the homeostasis between 2C-like and mES cells, and highlight that rRNA biogenesis in the nucleolus is a critical molecular switch from ZGA gene expressing 2-cell stage to nucleolus-matured blastocyst stage embryos.

## Results

### Inhibition of rRNA biogenesis activated the 2C-like transcriptional program and induced an expanded 2C-like cell population in mES cells.

We first explored whether nucleolar stress produced by inhibiting rRNA biogenesis could induce cell fate reprogramming to 2C-like cells (2CLCs) by performing RNA-seq analysis of mES cells treated by three inducers of cellular stress, including CX-5461, an RNA polymerase I (Pol I) inhibitor; rotenone, an electron transport chain complex 1 inhibitor, and rapamycin, a mTOR pathway inhibitor (CX-5461 treatment dosage: 2 µM, CX-5461 treatment time: 12 h; rotenone treatment dosage: 1 µM, rotenone treatment time: 12 h; rapamycin treatment dosage: 2 µM, rapamycin treatment time: 12 h). We found that nucleolar stress induced by CX-5461[26,27] activated the 2-cell marker genes *Zscan4d*, *Dux*, and *Gm12794* and repressed pluripotent marker gene *Pou5f1*. However, the other two cellular stresses upon rotenone or rapamycin treatment did not influence the expression of these genes (Fig. 1a, b and Supplementary Fig. S1a, b). The 2C-like transcriptional program was characterized by activation of transposable elements (TEs), particularly major satellite repeats (GSAT_MM) and ERVL subclasses MERVL-int and MT2_Mm. We systematically examined ERV genes and found global up-regulation of GSAT_MM and each LTR class (Fig. 1c and Supplementary Fig. S1e), particularly GSAT_MM, MERVL-int and MT2_Mm sub-classes, in CX-5461-treated mES cells (Fig. 1c, and Supplementary Fig. S1c, d). However, the other two types of stress, did not activate these repeat elements (Fig. 1c, and Supplementary Fig. S1c, d). Using unsupervised K-means clustering analysis, we identified four gene clusters specifically expressed in different stages during mouse pre-implantation embryo development (Supplementary Fig. S1f)[28]. We found that CX-5461 treatment upregulated 2-cell expressing cluster 1 (C1) and 2-cell/4-cell expressing cluster 2 (C2) genes, and decreased genes expressed in other two stages (C3 and C4) (Fig. 1d and Supplementary Fig. S1f). Yet, other two cellular stresses did not induce this expression pattern (Fig. 1d and Supplementary Fig. S1f). Moreover, unsupervised hierarchical clustering of transcriptomes of pre-implantation embryos and mES cells from published studies of 2C-like cells confirmed that mES cells treated with CX-5461 were most like the sorted 2CLCs from mES cells, or genetically modified mES cells with 2CLC signatures from other studies as well as the 2C embryos[1,11,14,16–18,29–31] (Fig. 1e). As expected, we observed the abundance of rRNA is significantly reduced under CX-5461 treatment (Supplementary Fig. S1g). Together, these results demonstrate that nucleolar stress induced by rRNA biogenesis defect activated 2C-like transcriptional program in mES cells.

We next asked how the population homeostasis of 2CLCs and ES cells altered in response to rRNA biogenesis defect at the single cell level. In line with bulk RNA-seq data, we observed that the 2C marker genes are up-regulated and pluripotent genes were downregulated in CX-5461-treated mES cells (Supplementary Fig. S2a, b). As expected, we found a marked expansion of the population with MERVL expression in mES cells treated with CX-5461 (Fig. 2a). Strikingly, we observed that the expression level of MERVL genes showed significant negative correlation with ribosomal protein genes (Fig. 2b–d and Supplementary Fig. S2c). Using a 2C::tdTomato reporter in which a *tdTomato* gene is under control of a MERVL promoter, we examined 2C status of individual cells by fluorescence activated cell sorting (FACS) analysis. Consistent with single-cell RNA-seq data, we observed that CX-5461-treatment induced a significant increase in the number of *tdTomato* positive (2C::tdTomato+) mES cells in a dose-dependent manner (Fig. 2e and Supplementary Fig. S2d). Moreover, the percentage of *tdTomato* positive cells was largely maintained even at 24 h after CX-5461 withdrawal (Fig. 2f). Importantly, although CX-5461, mostly at a high concentration, induced mild cell apoptosis, the majority of *tdTomato* positive cells were negative for the apoptosis markers Annexin-V and DAPI[32] (Supplementary Fig. S2e–g), suggesting that the emergence of 2C::tdTomato+ cells under CX-5461 treatment was not due to the activation of apoptosis pathway. Collectively, these results demonstrate that inhibiting rRNA biogenesis induced a shift of the ES cell homeostasis toward the MERVL-expressing and ribosomal gene repressed 2CLCs[33,34].

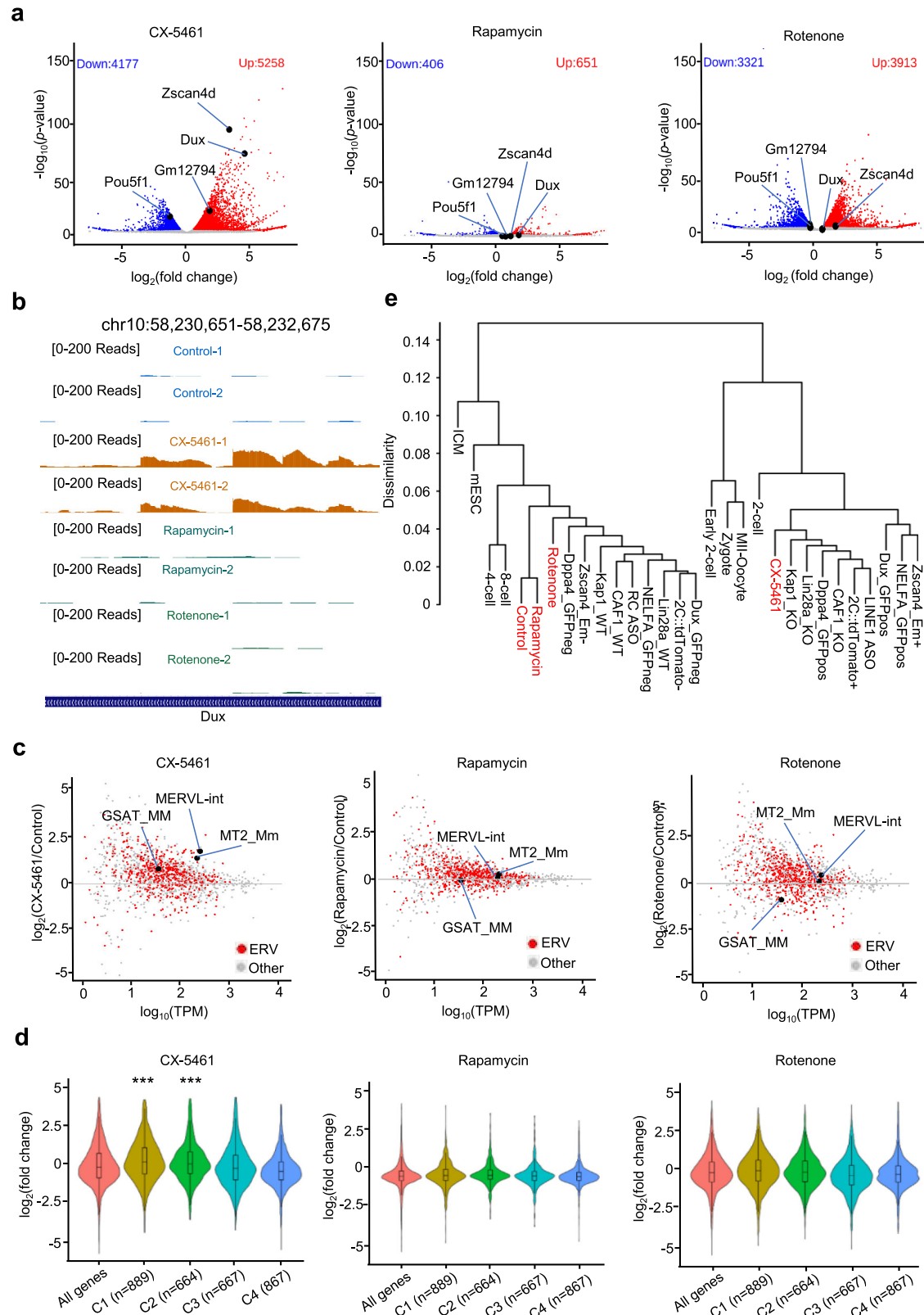

**Deficiency of rRNA biogenesis disrupted the normal state of nucleolar liquid–liquid phase separation (LLPS) and the epigenetic state of the PNH.** As it has been reported that ribosomal RNA plays a critical role in maintaining phase separation of nucleolus[35–37] and phase transition is involved in nucleolar stress[37,38], we examined whether nucleolar stress induced by rRNA biogenesis defect leads to changes of nucleolar phase separation. Using electron microscopy, we observed that CX-5461-treated mES cells displayed abnormal nucleolar structure, missing the outer layer usually associated with dense electron intensity (Fig. 3a). Immunofluorescence of key granular compartment (GC) and nucleolar dense fibrillar component (DFC) marker proteins NPM1 and NCL revealed that CX-5461 treatment led to disappearance of the NPM1-marked and

**Fig. 1 Inhibition of rRNA biogenesis activated 2C-like transcriptional program. a** Volcano plots of RNA-sequencing data comparing gene expression of control and cellular stress inducer-treated mES cells. GEO accession code GSE166041, $N = 2$ biologically independent RNA-seq experiments. **b** UCSC Genome Browser viewing of RNA-sequencing results at the *Dux* locus, GEO accession code GSE166041, $N = 2$ biologically independent RNA-seq experiments. **c** MA plots of RNA-sequencing data comparing repeat sequence expression of control and cellular stress inducer-treated mES cells. GEO accession code GSE166041, $N = 2$ biologically independent RNA-seq experiments. **d** Violin plots demonstrating the expressional changes of stage-specific gene clusters of mouse pre-implantation embryos under different types of cellular stress treatment; C1 vs. all genes: $p = 4.71E-11(***)$, C2 vs. all genes: $p = 2.96E-05(***)$, two-sided, Mann–Whitney *U*-test. GEO accession code GSE166041, $N = 2$ biologically independent RNA-seq experiments. **e** Hierarchical clustering of transcriptomes from our study, published 2C-like cell model studies and pre-implantation mouse embryos; Control, CX-5461, Rotenone and Rapamycin: GEO accession code GSE166041; 2C::*tdTomato*+ and 2C::*tdTomato*-: GEO accession code GSE33923; Zscan4_Em+ and Zscan4_Em-: GEO accession code GSE51682; Kap1_KO and Kap1_WT: GEO accession code GSE74278; CAF1_WT and CAF1_KO: GEO accession code GSE85632, Dux_GFPpos and Dux_GFPneg: GEO accession code GSE85632; LINE1 ASO and RC ASO: GEO accession code GSE100939; Dppa4_GFPpos and Dppa4_GFPneg: GEO accession code GSE120953, NELFA_GFPpos and NELFA_GFPneg: GEO accession code GSE113671; Lin28a_KO and Lin28a_WT: GEO accession GSE164420; MII-Oocyte, zygote, early 2-cell, 2-cell, 4-cell, 8-cell, ICM and mES cells: GEO accession code GSE66390. Gene expression levels of two biological replicates in same developmental stage were averaged for clustering. **d** The center line is the median, the bottom of the box is the 25th percentile boundary, the top of the box is the 75th, and the top and bottom of vertical line define the bounds of the data that are not considered outliers, with outliers defined as greater/lesser than ±1.5× IQR, where IQR inter-quartile range.

NCL-marked "ring" structure (Fig. 3b, c). Indeed, this molecular phenotype was observed in ~93% CX-5461 treated mES cells (26 out of 28 detected CX-5461 treated mES cells). Immunofluorescence of FBL and RPA194 protein also showed abnormal distribution and morphology with aggregated pattern in the nucleolus upon treatment (Fig. 3d, e). Consistently, fluorescence recovery after photobleaching (FRAP) analysis revealed that CX-5461 treatment markedly increased the mobility of NCL, NPM1, and FBL (Fig. 3f–h). Together, these data demonstrate that nucleolar stress caused disrupted assembly of the phase-separated nucleolar sub-compartments, which became fused and more dynamic liquid-like droplets[37,38].

As it has been observed that phase separation can regulate the epigenetic state of chromatin[39–41], we examined the epigenetic changes at the loci of the peri-nucleolar heterochromatin (PNH) region in CX-5461-treated mES cells. It has been reported that transcriptionally inactive genomic regions organize into Inactive Hubs around the nucleolus[42]. In addition, the nucleolar associated domains (NAD) or LINE1/L1 repeat sequence regions are also defined as repressive chromosomal segments enriched with peri-nucleolar heterochromatin[43–49]. We thus examined the epigenetic changes on these regions and found decreased H3K9me3 and H3K27me3 levels in CX-5461 treated cells (Fig. 4a, b and Supplementary Fig. S3a–c). Moreover, we observed increased H3K4me3 and H3K27ac levels and improved chromatin accessibility at Inactive Hub[42], NAD[43,45], and L1 regions (downloaded from UCSC Table Browser) in CX-5461 treated cells (Fig. 4c–g and Supplementary Fig. S3d–f). Previous work has reported that nucleolar protein nucleolin NCL and its interacting partner the heterochromatin protein TRIM28 repress 2C-like program by maintaining the PNH region in mES cells[14]. The disappeared NCL-marked "ring" structure suggested that rRNA biogenesis defect promoted the dissociation of NCL/TRIM28 complex from PNH region. To validate this, we conducted ChIP-seq experiment to investigate the binding changes of NCL/TRIM28 complex on the loci of the PNH region in CX-5461-treated mES cells and found decreased binding of NCL and TRIM28 proteins on Inactive Hub, NAD, and L1 regions (Fig. 4h, i and Supplementary Fig. S3g–i). Altogether, these results demonstrated the rRNA biogenesis defect affected the normal state of nucleolar phase separation and changed the epigenetic state of the heterochromatic regions at the periphery of nucleolus by breaking up the binding of NCL/TRIM28 complex on the PNH region.

**2C/ERV genes were activated through *Dux*.** Recent studies have reported that a pioneer transcription factor, the DUX protein,

directly binds to promoters and LTR elements on 2C genes and repetitive elements and activates their transcription[10–12]. As *Dux* expression has been reported to be influenced by nucleolar protein NCL[14], we speculated that *Dux* is the key molecular regulator for nucleolar stress-mediated activation of 2C-like transcriptional program. To this end, we performed the binding motif sequence enrichment analysis of transcription factors on 5258 genes induced by CX-5461. We found that the significantly enriched motifs include both P53 binding sites and DUX binding sites (Fig. 5a). In line with this, we found that 1229 CX-5461-induced genes are P53 direct target genes (Fig. 5b, hypergeometric test, *p*-value = 0)[50], consistent with the fact that p53 signaling is usually activated under nucleolar stress[51–57]. We further analyzed the overlap between CX-5461 treatment-induced genes and *Dux* over-expression-induced genes using published RNA-seq data[11], and found 621 genes are overlapped (Fig. 5b, hypergeometric test, *p*-value = 2.96e−195). Using P53 and DUX ChIP-seq data in mES cells[11,50], we further observed that both P53 and DUX showed a pattern of binding to the transcriptional start site (TSS) of CX-5461-induced genes (Fig. 5c). The binding pattern of DUX on these genes is weaker when comparing with the strong binding of P53 on CX-5461-induced genes. This was expected as CX-5461 induces many genes that are P53 target, but not DUX target. Interestingly, we found that P53 favors to bind specifically to CX-5461-induced genes, while DUX exclusively binds to the commonly induced genes between CX-5461-treated and *Dux*-over-expressed cells (Fig. 5d), and to ERV genes induced by CX-5461 (Fig. 5e, f). Consistently, a significant increase of chromatin accessibility in the promoter region is not observed for 4637 CX-5461-induced genes but is observed for 621 commonly induced genes or 10,173 CX-5461-induced ERV genes in *Dux*-over-expressed mES cells[11] (Supplementary Fig. S4a). In addition, we observed the decreased H3K9me3 and H3K27me3 levels and increased H3K4me3 and K3K27ac levels around *Dux* locus (Supplementary Fig. S4b). These analyses suggested that nucleolar stress-induced 2C activation is through *Dux*. To validate this hypothesis, we silenced *Dux* expression in CX-5461-treated mES cells and found that it reversed the 2C/ERV gene induction (Fig. 5g). We further performed ChIP-qPCR experiments to assess the changes of H3K9me3 & H3K27me3 levels and *Dux* binding of CX-5461-induced 2C marker genes after *Dux* silencing. We observed that, after *Dux* silencing, both H3K9me3 and H3K27me3 were increased. In contrast, *Dux* binding is decreased for CX-5461-induced 2C genes (Fig. 5h–j). When the *Dux* was silenced in CX-5461-treated mES cells, we observed that H3K9me3 and H3K27me3 levels, and DUX binding on CX-5461-induced 2C genes were reversed (Fig. 5h–j). We also performed

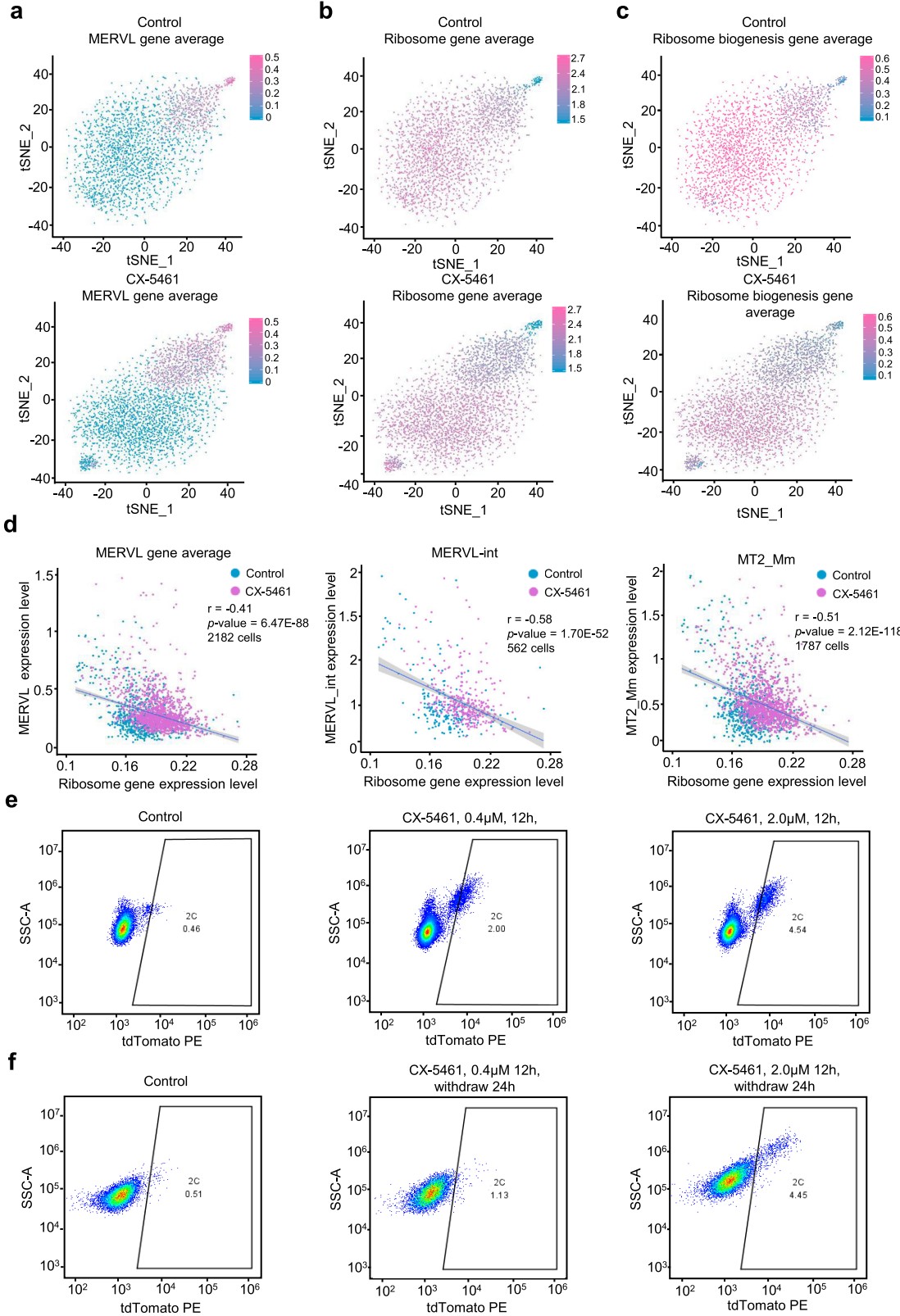

ATAC-seq experiment to assess the changes of chromatin accessibility after *Dux* silencing. In well support with these results, we observed that chromatin accessibility of CX-5461-induced 2C genes was reversed by *Dux* silencing (Fig. 5k). These results demonstrated that nucleolar stress induced 2C/ERV gene activation through *Dux*. In line with DUX's role in chromatin accessibility, we observed that the 621 commonly induced genes

and 10173 CX-5461 induced ERV genes have increased chromatin accessibility, and decreased H3K9me3 and H3K27me3 levels when comparing CX-5461 treated mES cells with control mES cells or comparing 2-cell embryos with ICM stage embryos[28,29,58] (Supplementary Fig. S4c–h). In addition, we also found increased H3K4me3 and H3K27ac levels of 621 commonly induced genes and 10,173 CX-5461 induced ERV genes in

**Fig. 2 Inhibition of rRNA biogenesis induced an expanded 2C-like cell population in mES cells. a** tSNE feature plots demonstrating the averaged expression levels of *MERVL* genes in 2981 control mES cells and 3219 CX-5461-treated mES cells. GEO accession code GSE166041. **b** tSNE feature plots demonstrating the averaged expression levels of ribosome genes in 2981 control mES cells and 3219 CX-5461-treated mES cells. GEO accession code GSE166041. **c** tSNE feature plots demonstrating the averaged expression levels of ribosome biogenesis genes in 2981 control mES cells and 3219 CX-5461-treated mES cells. GEO accession code GSE166041. **d** Scatter plots demonstrating negative correlation of expression level between MERVL/MERVL-int/MT2_Mm and ribosome genes; Each dot represents a single cell with detectable ERV expression; *r* denotes correlation coefficient; *p*-value was obtained by cor.test function in R software. GEO accession code GSE166041. **e** FACS analysis on 2C::*tdTomato*[+] mES cells upon different treatment doses of CX-5461, showing the change of percentage of 2C-like cells. **f** FACS analysis on 2C::*tdTomato*[+] mES cells after 12 h of treatment and 24 h withdrawal of CX-5461, showing the change of percentage of 2C-like cells.

CX-5461 treated mES cells (Supplementary Fig. S4i, j). Collectively, these results demonstrated that nucleolar stress induced a 2C-like transcriptional and epigenetic program in mES cells through *Dux*.

**rRNA biogenesis defect drove 3D chromatin structure reorganization of PNH and MERVL regions towards the 2C-like state.** The disassembly of PNH region suggested that the 3D chromatin structure within PNH region might have reshaped under CX-5461 treatment. To explore the reorganization of 3D chromatin conformation landscape of CX-5461-treated mES cells relative to control mES cells, we performed in situ Hi-C with more than four hundred million sequenced raw read pairs per sample. We observed markedly decreased higher-order chromatin interactions within PNH region indicated by the Inactive Hub, NAD, and L1 regions in the treated cells (Fig. 6a–c, compared with randomly selected genomic regions, Mann–Whitney *U*-test, the replicates of experiment *N* = 10, averaged *p* = 0, 0, 3.42E–06 for Inactive Hub, NAD, and L1, respectively). Moreover, the *Dux* locus is significantly further away from the PNH region as characterized by the largely decreased Hi-C contacts (Fig. 6a–c, g–i) compared with randomly selected genomic regions, Mann–Whitney *U*-test, the replicates of experiment *N* = 10, averaged *p* = 1.65E–05, 7.82E–05, 2.68E–03 for Inactive Hub, NAD, and L1, respectively). We further analyzed the 3D chromatin structural correlation within PNH region, and between the *Dux* locus and PNH region by comparing Hi-C pearson correlation coefficient (PCC) matrix of control and CX-5461-treated mES cells. We observed the obviously decreased 3D chromatin structural correlation within PNH region (compared with randomly selected genomic regions, Mann–Whitney *U*-test, the replicates of experiment *N* = 10, averaged *p* = 0 for all Inactive Hub, NAD, and L1) and between the *Dux* locus and PNH region (compared with randomly selected genomic regions, Mann–Whitney *U*-test, the replicates of experiment *n* = 10, averaged *p* = 1.86E–6, 5.03E–6, 4.02E–5 for Inactive Hub, NAD, and L1, respectively) (Fig. 6d–f, j–l). We further validated the above findings using DNA fluorescence in situ hybridization (FISH) and found that the *Dux* locus and a locus within PNH region located further from the peri-nucleolar region indicated by NCL staining in the CX-5461-treated cells (Fig. 6m). Interestingly, by performing the same analysis as above on public Hi-C data of mouse pre-implantation embryo development[21], we observed that a similar trend of 3D chromatin structure reorganization of PNH and *Dux* regions when comparing early two-cell embryos with ICM stage embryos, namely, the two-cell embryos showed less organized PNH and less contacts between the *Dux* Locus and the PNH (Supplementary Fig. S5a–d), suggesting a process of maturation of PNH 3D structure organization and of the *Dux* release from the PNH during embryo development. Interestingly, we observed that 3D structure reorganization of PNH was initiated during early two-cell to late two-cell transition, which coincides with the beginning of shutting down *Dux* gene

expression during murine early embryo development (Supplementary Figs. S5a–d and S9d).

As it has been reported that 2CLCs display increased three-dimensional structural plasticity relative to mES cells[23], we next asked whether the global 3D chromatin architecture is changed in CX-5461-treated mES cells. We compared control and CX-5461-treated mES cells Hi-C maps, with lymphoblastoid cells as a reference for fully differentiated cells[59]. A global analysis of A(active)/B(inactive) compartment strength showed a slight decrease of contacts within the B compartments in CX-5461-treated mES cells compared with control mES cells (Supplementary Fig. S6a–d). However, at topologically associating domain (TAD) or chromatin loop level, we found a mild increase in their strength in CX-5461-treated mES cells (Supplementary Fig. S6e–i). To specifically investigate 2C-related genes, we further performed an analysis of local architectural differences around MERVL loci. We found that the insulation scores[60] of chromatin around MERVL genes activated by CX-5461 treatment is markedly increased both globally (Supplementary Fig. S7a) and at local MERVL sites (Fig. 7a and Supplementary Fig. S7b), similar as observed in two-cell embryos compared with ICM[23] (Fig. 7a, b and Supplementary Fig. S7b), and the topological associated domain (TAD) structure around MERVL gene loci is more obvious (Fig. 7a, b and Supplementary Fig. S7b), showing a more similar pattern as that in two-cell embryos[21] (Fig. 7a, b and Supplementary Fig. S7b). The chromatin structure reorganization around MERVLs is accompanied with more open chromatin states around *MERVLs* both globally (Supplementary Fig. S7a) and locally (Fig. 7a and Supplementary Fig. S7b) and with their increased expression (Fig. 7a and Supplementary Fig. S7b). These results together demonstrate that nucleolar stress promoted the transformation of mES cells to 2C-like cells with reshaped 3D chromatin structure and its associated epigenetic status to facilitate gene expression, particularly at the PNH and MERVL regions.

**Genetic perturbation of rRNA biogenesis recapitulated CX-5461-induced 2C-like molecular phenotypes.** To further investigate the critical role of rRNA biogenesis in regulating the 2C program and the homeostasis between mES cells and 2C-like cells, we generated two rRNA biogenesis-inhibited mES cell lines: 1) a line with degraded Pol I protein (PRA1) by an auxin-inducible degron system[61] (Supplementary Fig. S8a), and 2) a snoRNA knockout line (SNORD113-114 gene cluster, a gift from Pengxu Qian) (Supplementary Fig. S8b), as snoRNAs are required for rRNA modification and biogenesis. Using these two cell lines, we performed DNA FISH experiments and found the similar molecular phenotypes of CX-5461-treated mES cells, i.e., the *Dux* locus and a representative locus within PNH region located further from the peri-nucleolar region (Fig. 8a–d). We then carried out FRAP experiments and consistently observed significantly increased mobility of NCL and NPM1 proteins in these two cell lines compared with their wild-type controls (Fig. 8e–h and Supplementary Fig. S8c–f). In line with these, we found that the 2C marker genes,

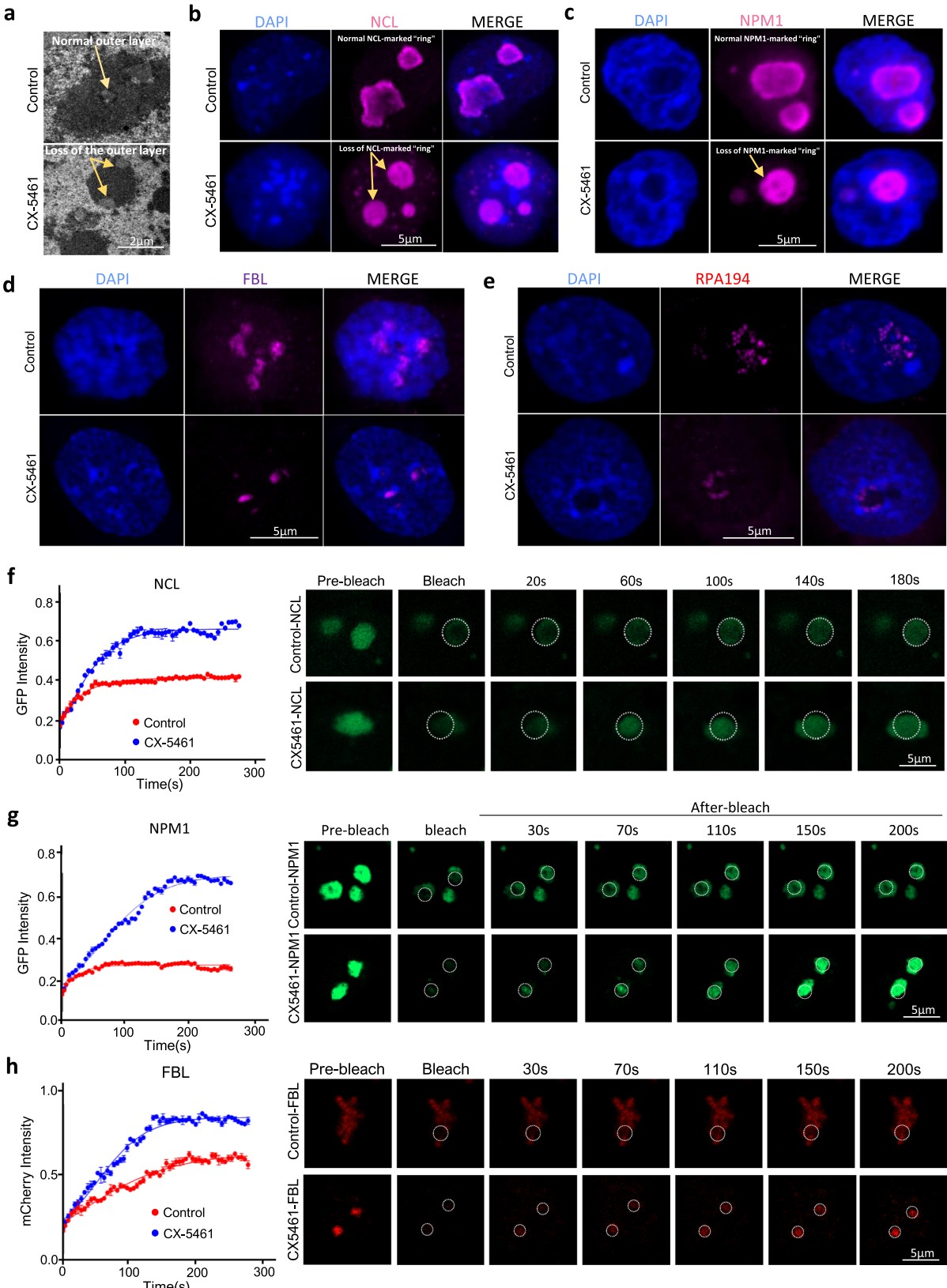

including *MERVL*, *Dux*, *Zscan4d*, *Gm12794*, and *Gm4340*, were significantly activated in these two cell lines (Fig. 8i, j). Moreover, using FACS analysis, we observed a significant increase of the percentage of *tdTomato* positive cells in these two cell lines compared with control cells (Fig. 8k, l and Supplementary Fig. S8g, h). Collectively, these results further confirmed that repressing rRNA biogenesis can activate 2C-like program and induce the transition of mES cells to 2C-like cells.

**Fig. 3 Deficiency of rRNA biogenesis disrupted the normal state of nucleolar LLPS. a** CX-5461 treatment causes abnormal nucleolar structure with electron microscopy; Experiment was repeated independently three times with similar results. **b** Immunofluorescence staining of NCL in control mES cells and CX-5461-treated mES cells; Experiment was repeated independently three times with similar results. **c** Immunofluorescence staining of NPM1 in control mES cells and CX-5461-treated mES cells; Experiment was repeated independently three times with similar results. **d** Immunofluorescence staining of FBL in control mES cells and CX-5461 treated mES cells; Experiment was repeated independently three times with similar results. **e** Immunofluorescence staining of RPA194 in control and CX-5461 treated mES cells; Experiment was repeated independently three times with similar results. **f** FRAP analysis showing CX-5461 treatment causes accelerated recovery after photobleaching of NCL; $N = 3$ biologically independent experiments; Data are presented as mean values $+/-$ SEM. SEM standard error of mean. Shown images are representative of three times of biologically independent experiments. **g** FRAP analysis showing CX-5461 treatment causes accelerated recovery after photobleaching of NPM1; $N = 3$ biologically independent ChIP-seq experiments; Data are presented as mean values $+/-$ SEM. SEM standard error of mean. Shown images are representative of three times of biologically independent experiments. **h** FRAP analysis showing CX-5461 treatment causes accelerated recovery after photobleaching of FBL; $N = 3$ biologically independent ChIP-seq experiments; Data are presented as mean values $+/-$ SEM. SEM standard error of mean. Shown images are representative of three times of biologically independent experiments. Source data are provided as a Source Data file.

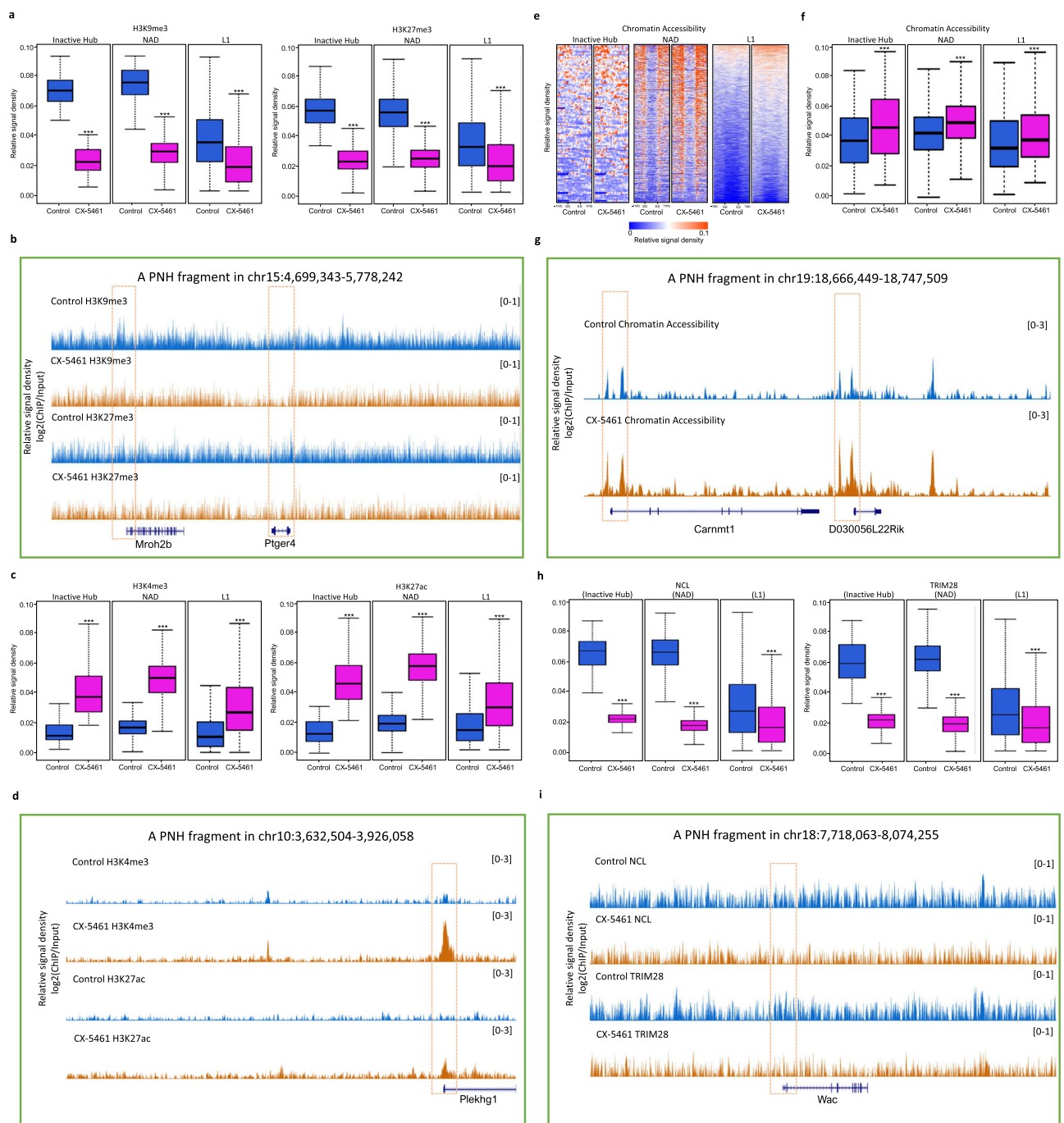

**Fig. 4 Deficiency of rRNA biogenesis changed the normal epigenetic state of PNH region. a** Boxplots demonstrate the averaged H3K9me3 and H3K27me3 levels of 101 Inactive Hub fragments, 578 NAD fragments and 34888 L1 sequences; H3K9me3 of PNH: $p = 2.56E-34(***)$, H3K9me3 of NAD: $p = 9.64E-185(***)$, H3K9me3 of L1: $p = 0(***)$, H3K27me3 of PNH: $p = 1.89E-33(***)$, H3K27me3 of NAD: $p = 2.35E-160(***)$, H3K27me3 of L1: $p = 0(***)$, two-sided, Wilcox signed rank test; $N = 1$ biologically independent ChIP-seq experiment. GEO accession code GSE166041 . **b** UCSC Genome Browser viewing of H3K9me3 and H3K27me3 ChIP-seq signals in control and CX-5461 treated mES cells around a representative PNH fragment at chr15:4,699,343–5,778,242. GEO accession code GSE166041. **c** Boxplots demonstrate the averaged H3K4me3 and H3K27ac levels of 101 Inactive Hub fragments, 578 NAD fragments and 34888 L1 sequences; H3K4me3 on PNH: $p = 1.86E-23(***)$, H3K4me3 on NAD: $p = 8.48E-172(***)$, H3K4me3 on L1: $p = 0(***)$, H3K27ac on PNH: $p = 6.46E-26(***)$, H3K27ac on NAD: $p = 2.22E-172(***)$, H3K27ac on L1: $p = 0(***)$, two-sided, Wilcox signed rank test, $N = 1$ biologically independent ChIP-seq experiment. GEO accession code GSE166041. **d** UCSC Genome Browser viewing of H3K4me3 and H3K27ac ChIP-seq signals in control mES cells and CX-5461 treated mES cells around a representative PNH fragment at chr10:3,632,504–3,926,058. GEO accession code GSE166041. **e** Heatmap plots demonstrate the levels of chromatin accessibility on within 1 mb region around start and end sites of Inactive Hub and NAD, and within 1 kb region around start and end sites of L1. The regions of different lengths of Inactive Hub and NAD fragments were fitted to 1 mb. The regions of different lengths of L1 sequences were fitted to 1 kb. SS: start site of a chromatin fragment of PNH; ES: end site of a chromatin fragment of PNH. The PNH fragment was defined as the L1 contained regions overlapped with Inactive Hub and NAD. GEO accession code GSE166041. **f** Boxplots demonstrate the averaged chromatin accessibility of 101 Inactive Hub fragments, 578 NAD fragments and 34,888 L1 sequences; chromatin accessibility of PNH: $p = 0(***)$, chromatin accessibility of NAD: $p = 2.65E-09(***)$, chromatin accessibility of L1: $p = 9.17E-74(***)$, two-sided, Wilcox signed rank test, $N = 1$ biologically independent ChIP-seq experiment. GEO accession code GSE166041. **g** UCSC Genome Browser viewing of ATAC-seq signals in control and CX-5461 treated mES cells around a representative PNH fragment at chr19:18,666,449–18,747,509. GEO accession code GSE166041. **h** Boxplots demonstrate the averaged binding signals of NCL and TRIM28 on 101 Inactive Hub fragments, 578 NAD fragments and 34,888 L1 sequences; NCL on PNH: $p = 2.56E-34(***)$, NCL on NAD: $p = 1.71E-179(***)$, NCL on L1: $p = 0(***)$, TRIM28 on PNH: $p = 3.16E-34(***)$, TRIM28 on NAD: $p = 1.04E-175(***)$, TRIM28 on L1: $p = 0(***)$, two-sided, Wilcox signed rank test, $N = 1$ biologically independent ChIP-seq experiment. GEO accession code GSE166041. **i** UCSC Genome Browser viewing of NCL and TRIM28 ChIP-seq signals in control and CX-5461 treated mES cells around a representative PNH fragment at chr18:7,718,063–8,074,255. GEO accession code GSE166041. **a**, **c**, **f**, **h** The center line is the median, the bottom of the box is the 25th percentile boundary, the top of the box is the 75th, and the whiskers define the bounds of the data that are not considered outliers, with outliers defined as greater/lesser than ±1.5× IQR, where IQR inter-quartile range.

**rRNA biogenesis is critically required at the two-cell-to-four-cell stage transition during pre-implantation embryo development.** To better understand whether the physiological function of rRNA biogenesis is to facilitate embryo development during and after the two-cell exit, we first inspected rRNA expression levels in all stages of pre-implantation mouse embryos. The precursor and matured rRNA levels were low in MII-oocyte, pronuclear zygote, early two-cell and middle two-cell stages but increased sharply from the late two-cell stage to the blastocyst stage (Fig. 9a). We then analyzed the expression levels of different subunit genes of RNA polymerase I (Pol I) in pre-implantation embryos. Different from rRNA expression, Pol I gene mRNA levels increased significantly from the late two-cell stage to four-cell stage but decreased markedly as embryos progressed through eight-cell, morula stages, and reached to blastocysts which still had a higher level than those stages before the late two-cell (Fig. 9b). Consistent with Pol I genes, we observed the similar pattern of increased expression of ribosome biogenesis genes (Supplementary Fig. S9a). This indicated that while the levels of rRNA was gradually accumulated during pre-implantation embryo development, the rRNA biogenesis rate reached to a peak during the late two-cell to the four-cell stage. In contrast, the ERV and 2C marker genes, such as *Dux*, *Zscan4d*, and *Gm12794* were significantly decreased during the late two-cell-to-four-cell stage (Supplementary Fig. S9b–d). This reciprocal expression pattern between 2C marker genes and rRNA biogenesis genes suggested that rRNA biogenesis may play a key role in shutting down the 2C program, as revealed by our 2CLC emergence analysis in cultured mES cells, and in promoting the transition from the two-cell to the four-cell stage. We next applied CX-5461 (an embryo tolerable concentration) to mouse early embryos as they progress through pronuclear zygotes to blastocysts. When compared with the control embryos, we found that CX-5461-treated embryos were indeed blocked before the four-cell stage (Supplementary Fig. S9e). We further divided mouse embryos into four groups according to the different stages of CX-5461 treatment, including transitions of zygote-to-two-cell, two-cell-to-four-cell, and morula-to-blastocyst, respectively (Fig. 9c).

Compared with the untreated control, we found that blastocyst formation rates of all three CX-5461-treated groups were decreased, and the two-cell-to-four-cell-treated group showed the strongest decrease of the blastocyst formation rate at both early and late blastocyst stages (Fig. 9d). This is consistent with the pattern of Pol I gene RNA expression and the pattern of PNH reshaping after early two-cell stage during pre-implantation embryo development (Supplementary Fig. S5a–d), indicating that rRNA biogenesis is most critically required during the two-cell-to-four-cell transition compared with other stages. In well support with this, the knockout of *Polr1a* gene led to mouse embryos arrested at two-cell[62]. Moreover, upon successful inhibition of rRNA biogenesis at the morula/blastocyst stage (Fig. 9e), we found the disappearance of the NPM1-marked and NCL-marked "ring" structure and abnormal localization and reduced signal density of FBL in the nucleolus (Supplementary Fig. S9f). Importantly, we observed increased expression of 2C genes such as *Zscan4d*, *Gm4340*, *Dux*, and *Mervl-pol* (Fig. 9f). RNA-seq also demonstrated upregulated C1 and C2 clusters of 2C genes (Fig. 9g) defined before (Supplementary Fig. S1f) as well as ERV genes (Fig. 6h–j) in CX-5461-treated embryos compared with controls, consistent with the results from mES cells described above (Fig. 1a–d and Supplementary Fig. S1a–e). Altogether, these data demonstrated rRNA biogenesis and nucleolar integrity is a molecular switch for the transition from the two-cell to the four-cell embryos.

## Discussion

Starting from zygotic genome activation (ZGA) at the early two-cell stage, an embryo undergoes multiple developmental stages to be ready for itself for implantation. Along with the pre-implantation embryo development, the activity of anabolic metabolism and translation is progressively increased. Nucleoli, a nuclear sub-compartment responsible for the biogenesis of ribosome, originally derived from nucleolar precursor bodies (NPB), become mature structurally and functionally during this process[63–65]. Interestingly, a recent study reported that TRIM28/Nucleolin/LINE1 complex can regulate both *Dux* repression and

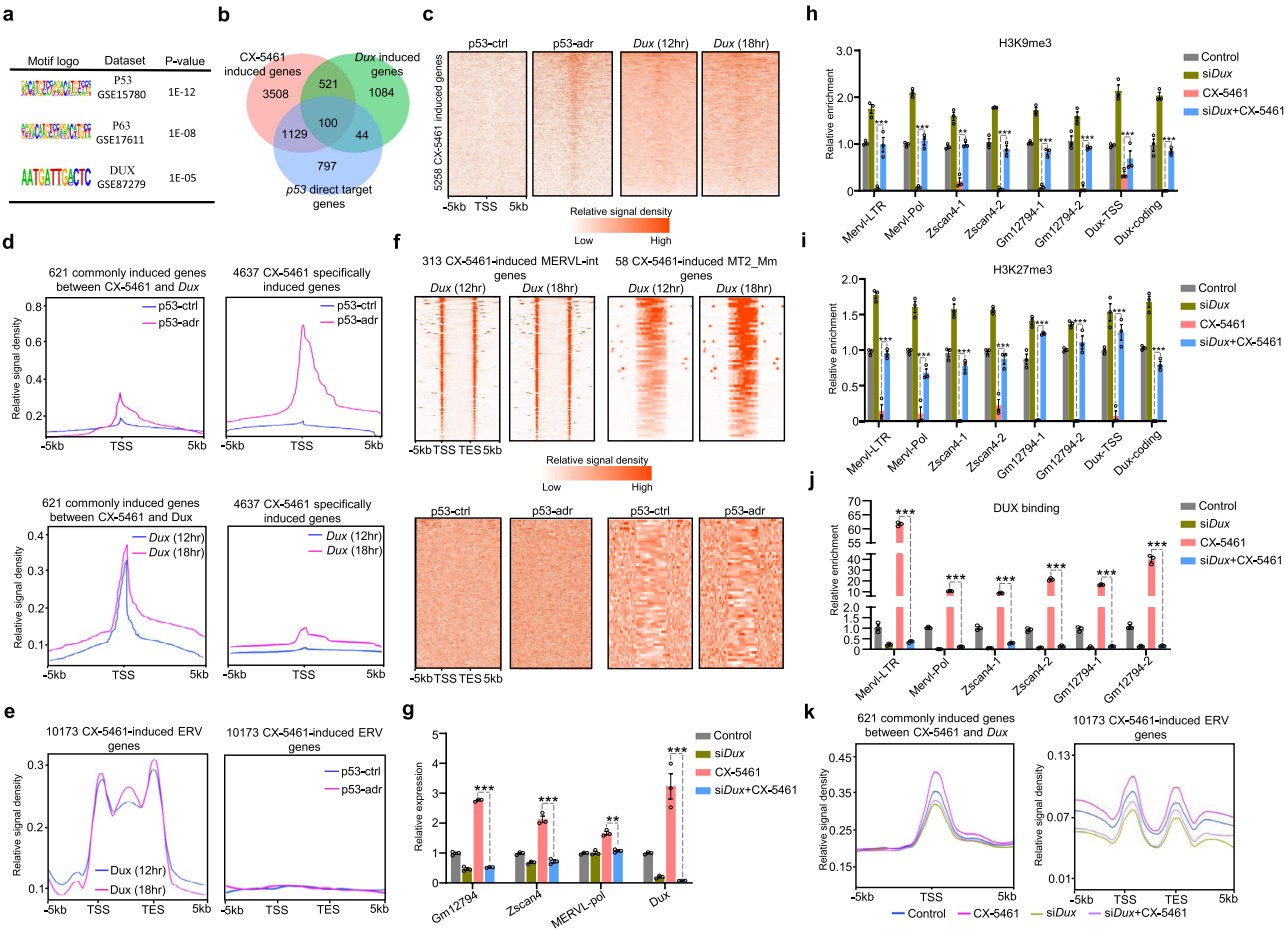

**Fig. 5 2C/ERV genes were activated through the *Dux*. a** Enriched binding motifs of 5258 genes induced by CX-5461 treatment. **b** Venn diagrams showing the overlap among CX-5461 treatment induced genes, *p53* activated direct target genes and *Dux*-overexpression induced genes. **c** Heatmap plots demonstrate the binding of *p53* and *Dux* proteins on within 5 kb region around transcription start sites of 5258 CX-5461 induced genes using published P53 ChIP-seq data and DUX ChIP-seq data. GEO accession code GSE26360 for P53 and GEO accession code GSE85632 for DUX . **d** Line plots demonstrate the meta-analysis results of P53 and DUX on within 5 kb region around transcription start sites of 621 commonly induced genes between CX-5461 treatment and *Dux* overexpression and 4637 specifically induced genes by CX-5461 using published P53 ChIP-seq data and DUX ChIP-seq data. GEO accession code GSE26360 for P53 and GEO accession code GSE85632 for DUX. **e** Line plots demonstrate the meta-analysis results of P53 and DUX protein on within 5kb region around transcription start and end sites of 10,173 CX-5461 induced ERV genes using published P53 ChIP-seq data and DUX ChIP-seq data. The regions of different lengths of gene body were fitted to 5 kb. GEO accession code GSE26360 for P53 and GEO accession code GSE85632 for DUX. **f** Heatmap plots demonstrate the binding of P53 and DUX proteins on within 5 kb region around transcription start and end sites of 5258 CX-5461 induced MERVL-int and MT2_Mm genes; The regions of different lengths of gene body were fitted to 5 kb. GEO accession code GSE26360 for P53 and GEO accession code GSE85632 for DUX. **g** qRT-PCR showing the expression of *Dux* or 2C-related genes in *Dux* silenced mES cells; GM12794: $p = 1.21E{-}13(***)$, Zscan4: $p = 3.00E{-}09(***)$, MERVL-pol: $p = 4.55E{-}03(**)$, *Dux*: $p = 9.60E{-}14(***)$, two-way ANOVA; $N = 3$ biologically independent ChIP-qPCR experiment; Data are presented as mean values $+/-$ SEM. SEM standard error of mean. **h** ChIP-PCR showing H3K9me3 levels of *Dux* or 2C-related genes in *Dux* silenced mES cells; MERVL-LTR: $p = 1.04E{-}11(***)$, MERVL-Pol: $p = 9.68E{-}12(***)$, Zscan4-1: $p = 2.51E{-}10(**)$, Zscan4-2: $p = 1.67E{-}11(***)$, Gm12794-1: $p = 1.90E{-}09(***)$, Gm12794-2: $p = 2.66E{-}11(***)$, Dux-TSS: $p = 1.35E{-}04(*)$, Dux-coding: $p = 4.92E{-}11(***)$, two-way ANOVA; $N = 3$ biologically independent ChIP-qPCR experiment; Data are presented as mean values $+/-$ SEM. SEM standard error of mean. **i** ChIP-PCR showing H3K27me3 levels of *Dux* or 2C-related genes in *Dux* silenced mES cells; MERVL-LTR: $p = 9.87E{-}12(***)$, MERVL-Pol: $p = 2.05E{-}08(***)$, Zscan4-1: $p = 1.12E{-}11(***)$, Zscan4-2: $p = 5.81E{-}10(**)$, Gm12794-1: $p = 9.56E{-}12(***)$, Gm12794-2: $p = 9.55E{-}12(***)$, Dux-TSS: $p = 9.55E{-}12(***)$, Dux-coding: $p = 1.02E{-}11(***)$, two-way ANOVA; $N = 3$ biologically independent ChIP-qPCR experiment; Data are presented as mean values $+/-$ SEM. SEM standard error of mean. **j** ChIP-PCR showing DUX protein binding levels on 2C-related genes in *Dux* silenced mES cells; MERVL-LTR: $p < 1.00E{-}15(***)$, MERVL-Pol: $p < 1.00E{-}15(***)$, Zscan4-1: $p < 1.00E{-}15(***)$, Zscan4-2: $p < 1.00E{-}15(***)$, Gm12794-1: $p < 1.00E{-}15(***)$, Gm12794-2: $p < 1.00E{-}15(***)$, two-way ANOVA; $N = 3$ biologically independent ChIP-qPCR experiment; Data are presented as mean values $+/-$ SEM. SEM standard error of mean. **k** Line plots demonstrate the meta-analysis results of chromatin accessibility in *Dux* silenced mES cells within 5 kb region around transcription start sites or transcription start and end sites of 621 commonly induced genes between CX-5461 treatment and *Dux* overexpression and 10,173 CX-5461 induced ERV genes. The regions of different lengths of ERV genes were fitted to 5 kb. GEO accession code GSE166041. p53-ctrl: untreated mES cells, p53-adr: mES cells treated with adriamycin, a DNA damage agent widely used to activate *p53*, *Dux* (12 h): mES induced with doxycycline for 12 h, *Dux* (18 h): mES induced with doxycycline for 18 h. Source data are provided as a Source Data file.

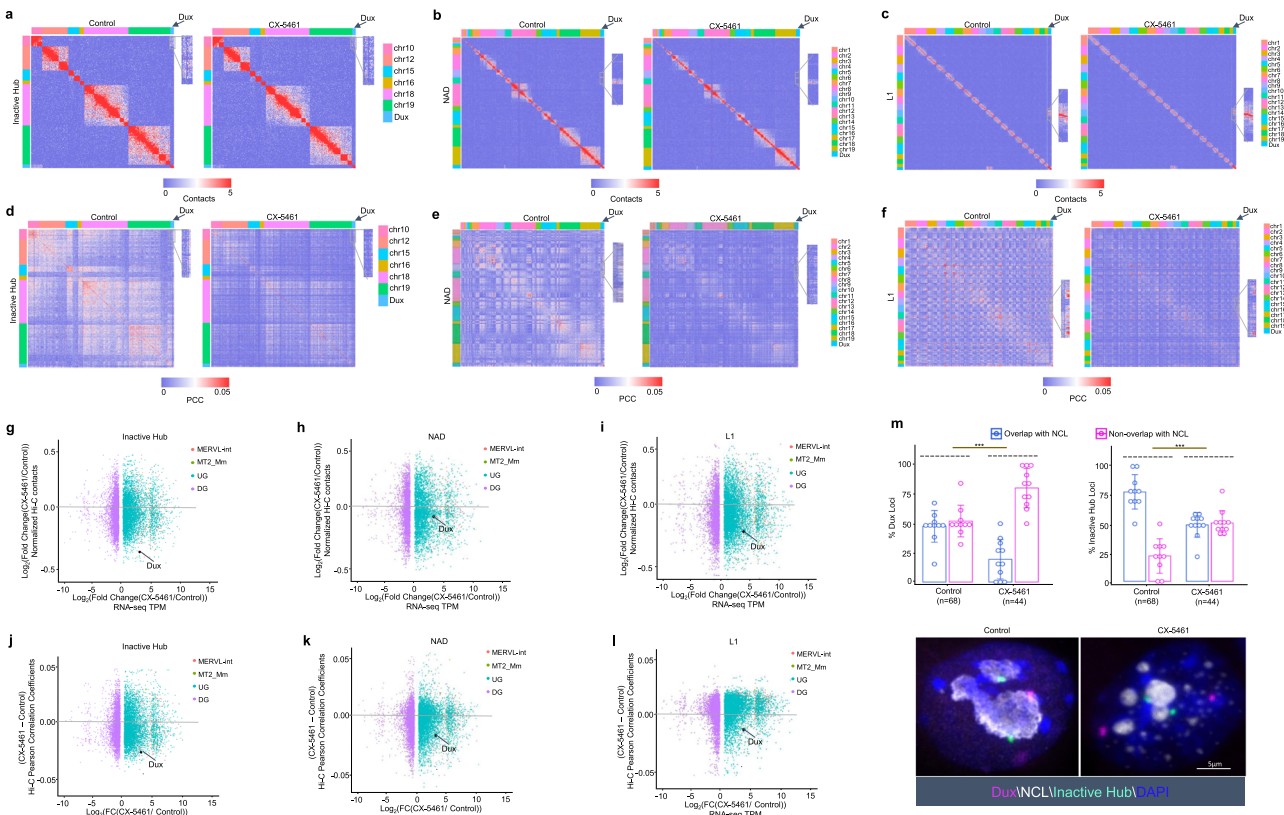

**Fig. 6 rRNA biogenesis defect drove 3D chromatin structure reorganization of PNH region towards the 2C-like state. a** Hi-C contact maps of Inactive Hub and 1.5 Mb genomic regions around *Dux* at 150 kb resolution. GEO accession code GSE166041. **b** Hi-C contact maps of NAD and 1.5 Mb genomic regions around *Dux* at 150 kb resolution. GEO accession code GSE166041. **c** Hi-C contact maps of L1 and 1.5 Mb genomic regions around *Dux* at 150 kb resolution. The zoomed-in regions aim to demonstrate the change of Hi-C contacts between *Dux* and chromosome 10 in control mES cells and CX-5461 treated mES cells. GEO accession code GSE166041. **d** Hi-C pearson correlation coefficient (PCC) heat maps of Inactive Hub and 1.5 Mb genomic regions around *Dux* at 150 kb resolution. GEO accession code GSE166041. **e** Hi-C pearson correlation heat maps of NAD and 1.5 Mb genomic regions around *Dux* at 150 kb resolution. GEO accession code GSE166041. **f** Hi-C pearson correlation heat maps of L1 and 1.5 Mb genomic regions around *Dux* at 150 kb resolution. The zoomed-in regions aim to demonstrate the change of Hi-C PCC between *Dux* and chromosome 10 in control mES cells and CX-5461 treated mES cells. GEO accession code GSE166041. **g** Scatter plot demonstrates the log$_2$(fold change) of Hi-C contacts between Inactive Hub and different types of genes in control and CX-5461 treated mES cells. GEO accession code GSE166041. **h** Scatter plot demonstrates the log$_2$(fold change) of Hi-C contacts between NAD and different types of genes in control and CX-5461 treated mES cells. GEO accession code GSE166041. **i** Scatter plot demonstrates the log$_2$(fold change) of Hi-C contacts between L1 and different types of genes in control and CX-5461 treated mES cells. GEO accession code GSE166041. **j** Scatter plot demonstrates the PCC difference between Inactive Hub and different types of genes in control and CX-5461 treated mES cells. GEO accession code GSE166041. **k** Scatter plot demonstrates the PCC difference between NAD and different types of genes in control and CX-5461 treated mES cells. GEO accession code GSE166041. **l** Scatter plot demonstrates the PCC difference between L1 and different types of genes in control and CX-5461 treated mES cells. The difference of PCC is defined as the average (PCC) of Inactive Hub regions and different types of genes in CX-5461 treated mES cells minus the average PCC of Inactive Hub regions and different types of genes in wild type mES cells. PCC Pearson correlation coefficient, MERVL-int: upregulated MERVL-int genes, MT2_Mm upregulated MT2_Mm genes, UG upregulated genes, DG downregulated genes. GEO accession code GSE166041. **m** DNA FISH analysis with a *Dux* locus probe and Inactive Hub locus probe, and co-immunostained with NCL protein. The percentage of Nucleolus-localized (overlapped with NCL) and Nucleoplasm-localized (nonoverlapped with NCL) of FISH signals is calculated. *Dux*: $p = 3.67\text{E}{-05}(\text{***})$, Inactive Hub: $p = 4.13\text{E}{-05}(\text{***})$, chi-square test; $N$ denotes the number of observed mES cells; $N = 11$ biologically independent observations; Data are presented as mean values $+/-$ SEM. SEM standard error of mean. GEO accession code GSE166041. Source data are provided as a Source Data file.

rRNA biogenesis[14]. This finding suggests a tight interconnection between turning off ZGA and initiating nucleoli formation. However, we did not completely known how the switch between ZGA and nucleolar formation occurs in the nucleus.

Here, we reported that nucleolar rRNA biogenesis and higher-order 3D chromatin structure remodeling of PNH might coordinate to develop during the two-cell to later stage transition, and we found that mES cells cultured in vitro can transform into 2C-like cells upon nucleolar stress caused by repressing rRNA biogenesis. We propose a mechanistic model for the role of rRNA biogenesis in regulating mouse 2C-like state (Fig. 10). In the unperturbed mouse embryonic stem cells, nucleolar integrity

mediated by rRNA biogenesis maintains the normal the liquid–liquid phase separation (LLPS) of nucleolus and the establishment of peri-nucleolar heterochromatin (PNH), and this normal nucleolar LLPS facilitates NCL/TRIM28 complex occupancy on the *Dux* locus and repression of *Dux* expression. In contrast, in the rRNA biogenesis-repressed mouse embryonic stem cells, the natural liquid-like phase of nucleolus is disrupted, causing the dissociation of the NCL/TRIM28 complex from PNH and alterations of epigenetic state and 3D structure of PNH, which finally leads to *Dux* released from PNH region, activation of 2C-like program and transition of mES cells to 2C-like cells. Given the dynamic regulation of nucleolus and rRNA gene

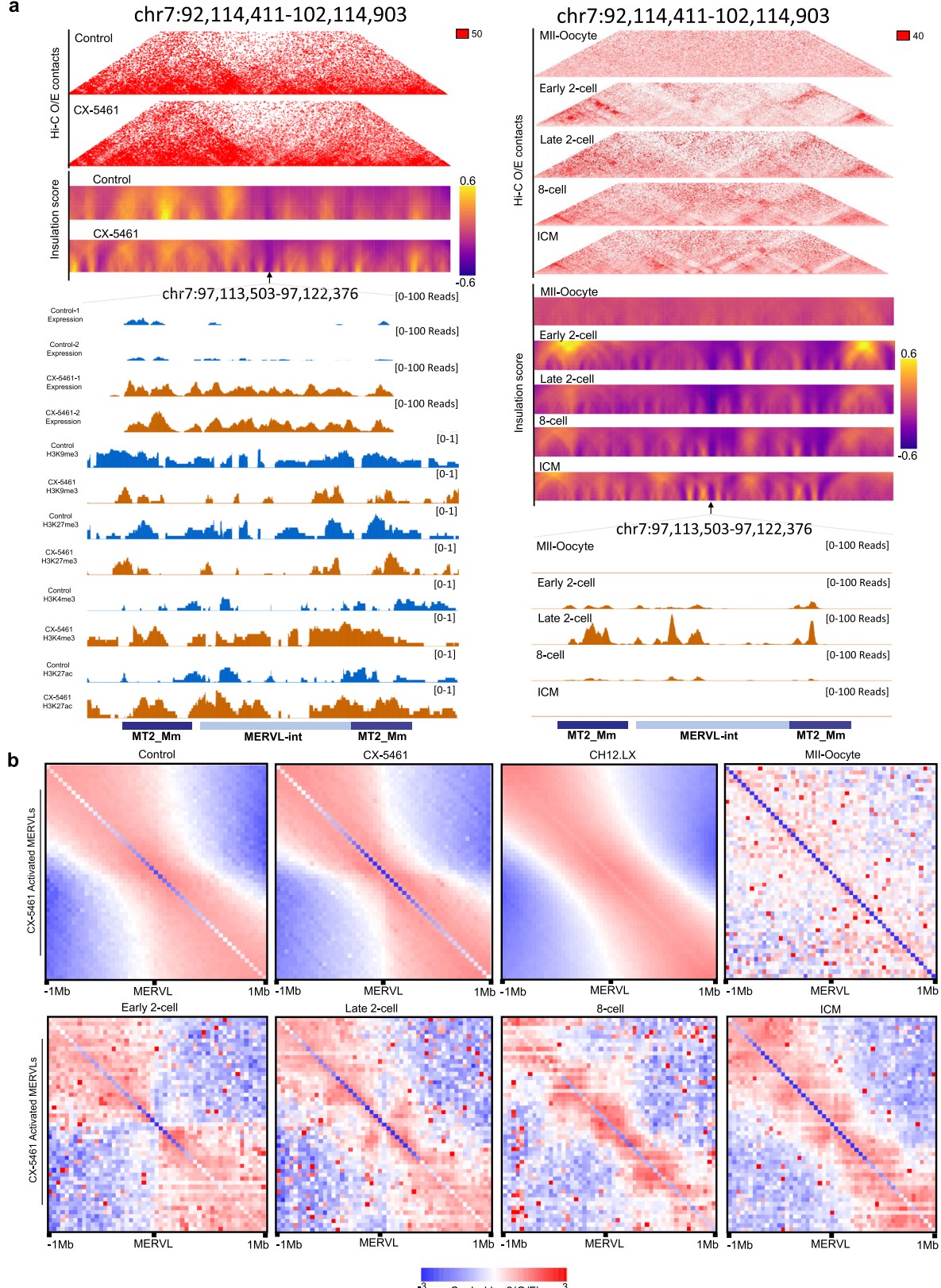

**Fig. 7 rRNA biogenesis defect drove 3D chromatin structure reorganization of MERVL region towards the 2C-like state. a** Aggregate Observed(O)/Expected(E) Hi-C matrices centered on CX-5461 induced MERVL genes in control, CX-5461 treated mES cells and mouse embryos throughout mouse pre-implantation embryonic development. GEO accession codes GSE166041, GSE63525, and GSE82185. **b** Representative 40 kb Hi-C O/E interaction matrices of a MERVL loci located at TAD boundaries (chr7:97,113,503–97,122,376) are shown as heatmaps, along with the insulation score and genome browser tracks of RNA-Seq, H3K9me3, H3K27me3, H3K4me3, and H3K27ac ChIP-Seq signals of the expanded genomic region containing the TAD boundary (arrows) in control and CX-5461 treated mES cells as well as in mouse early embryos. GEO accession code GSE166041 and GSE82185.

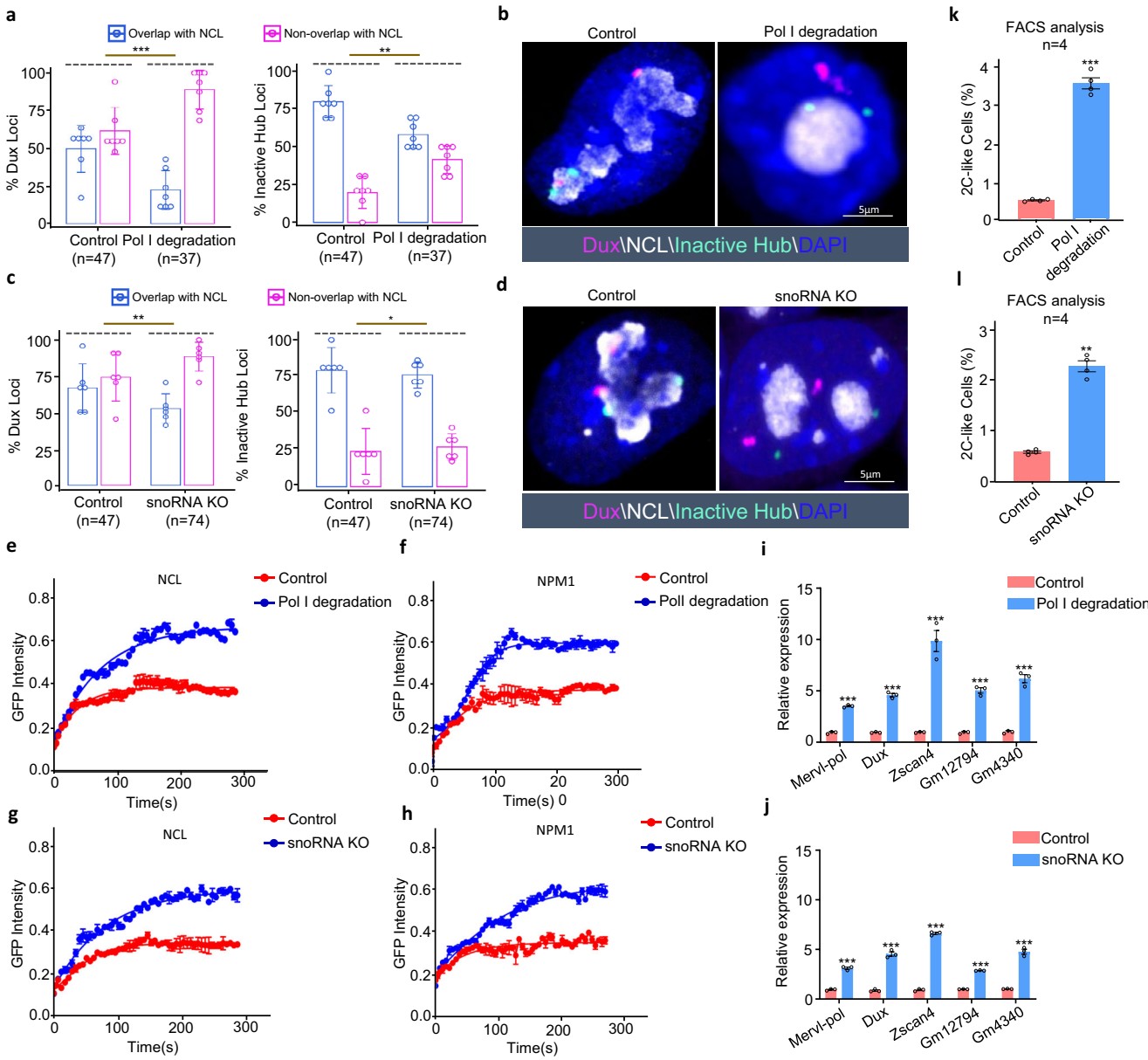

chromatin during early embryo development and the sensitivity of embryos to environmental stress at the early stages, it is conceivable that embryos may use the mechanisms elucidated above to ensure its safe development.

Nucleolus, the largest membrane-less condensate in a cell, is a stress-sensitive organelle and ensure quality control of nuclear proteome under stress[37,38]. Its association with heterochromatin in its periphery confers genetic regulation of key cell fate decision factors such as *Dux* in pluripotent stem cells. Previous studies on nucleolus in stem cells mainly focused on the role of rRNA and its associated chromatin in the context of ES cell self-renewal and differentiation or exit of pluripotency[44,57,66]. In contrast, our work provided a perspective in reprogramming mES cells back to 2C-like cell, and in nucleolar phase separation and 3D chromatin structure remodeling at the PNH. These findings are in line with the emerging notion that phase-separated condensates regulate transcription, epigenetics, and higher-order chromatin structure[39–41,67–70], and shed light on a previously neglected area of nucleolus-associated condensates in chromatin control during early development. It is worth-mentioning that we do not intend to overstate that the fate transition of mES to 2C-like cell

triggered by rRNA biogenesis defect is only explained by LLPS[71]. What we observed is that the integrity of nucleolus mediated by rRNA biogenesis maintains the normal nucleolar LLPS and 3D structure of the PNH. It is possible that 3D structure reshaping of the PNH mediated by nucleolar LLPS is a common thread of RNA and RNA binding protein-mediated Dux silencing and 2C repression (e.g., through rRNA, snoRNA, LINE1 RNA, NCL, TRIM28, or LIN28)[14,72]. The NCL/TRIM28 complex or other proteins localized at the nucleolus and its peripheral heterochromatin are possibly the key factors coordinately connecting nucleolar LLPS and the establishment and maintenance of the PNH. Our study provides initial evidence that the LLPS model is helpful in explaining how the assembly and function of nucleolus[37] participate in the gene regulation process in the nucleolus, and the quantitative models merit future investigations[73].

## Methods

**Cell culture**. E14 wild-type mES cells were cultured on 0.1% gelatin-coated plates with mouse embryonic fibroblast feeder cells in LIF/2i medium (1:1 mix of DMEM/F12 (11320-033, Gibco) and Neurobasal medium (21103-049, Gibco) containing

**Fig. 8 Genetic interferences of rRNA biogenesis recapitulated CX-5461-induced 2C-like molecular phenotypes. a** The percentage of Nucleolus-localized (overlapped with NCL) and Nucleoplasm-localized (nonoverlapped with NCL) of FISH signals in control and Pol I degradation mES cell lines; *Dux*: $p = 3.77E$ $-04(***)$, Inactive Hub: $p = 1.96E-03(**)$, chi-square test; $N$ denotes the number of observed mES cells; $N = 7$ biologically independent observations; Data are presented as mean values $+/-$ SEM. SEM standard error of mean. **b** DNA FISH analysis with a *Dux* locus probe and Inactive Hub locus probe, and co-immunostained with NCL protein in control and Pol I degradation mES cell lines; Experiment was repeated independently three times with similar results. **c** The percentage of Nucleolus-localized (overlapped with NCL) and Nucleoplasm-localized (nonoverlapped with NCL) of FISH signals in control and snoRNA knockout mES cell lines; *Dux*: $p = 7.09E-03(**)$, Inactive Hub: $p = 0.04(*)$, chi-square test; $N$ denotes the number of observed mES cells, $N = 6$ biologically independent observations; Data are presented as mean values $+/-$ SEM. SEM standard error of mean. **d** DNA FISH analysis with a *Dux* locus probe and Inactive Hub locus probe, and co-immunostained with NCL protein in control and snoRNA knockout mES cell lines; Experiment was repeated independently three times with similar results. **e** FRAP analysis showing Pol I degradation causes accelerated recovery after photobleaching of NCL; $N = 3$ biologically independent experiments; Data are presented as mean values $+/-$ SEM. SEM standard error of mean. **f** FRAP analysis showing Pol I degradation causes accelerated recovery after photobleaching of NPM1; $N = 3$ biologically independent experiments; Data are presented as mean values $+/-$ SEM. SEM standard error of mean. **g** FRAP analysis showing snoRNA knockout causes accelerated recovery after photobleaching of NCL; $N = 3$ biologically independent experiments; Data are presented as mean values $+/-$ SEM. SEM standard error of mean. **h** FRAP analysis showing snoRNA knockout causes accelerated recovery after photobleaching of NPM1; $N = 3$ biologically independent experiments; Data are presented as mean values $+/-$ SEM. SEM standard error of mean. **i** qRT-PCR quantification of 2C marker gene expression in control mES cells and Pol I degraded mES cell lines; MERVL-pol: $p = 3.15E-04(***)$, *Dux*: $p = 3.50E-06(***)$, *Zscan4*: $p = 8.25E-13(***)$, *Gm12794*: $p = 5.68E-07(***)$, *Gm4340*: $p = 1.26E-08(***)$, two-way ANOVA; $N = 3$ biologically independent experiments; Data are presented as mean values $+/-$ SEM. **j** qRT-PCR quantification of 2C marker gene expression in control mES cells and snoRNA knockout mES cell lines; MERVL-pol: $p = 4.34E-07(***)$, *Dux*: $p=2.61E-09(***)$, *Zscan4*: $p = 3.46E-11(***)$, *Gm12794*: $p = 3.70E-06(***)$, *Gm4340*: $p = 2.13E-09(***)$, two-way ANOVA; $N = 3$ biologically independent experiments; Data are presented as mean values $+/-$ SEM. **k** The percentage of 2C::*tdTomato* positive cells was quantified using FACS analysis in control mES cells and Pol I degraded mES cells; $p = 9.10E-04 (***)$, two-way ANOVA; $N = 4$ biologically independent experiments; Data are presented as mean values $+/-$ SEM. SEM standard error of mean. **l** The percentage of 2C::*tdTomato* positive cells was quantified using FACS analysis in control mES cells and snoRNA knockout mES cells; $p = 1.64E$ $-03(**)$, two-way ANOVA; $N = 4$ biologically independent experiments; Data are presented as mean values $+/-$ SEM. SEM standard error of mean. Source data are provided as a Source Data file.

1× N2 and B27 supplements (17502-048/17504-044, Life Technologies), 100 μM non-essential amino acids (GNM71450, GENOM), and 1000 U/ml LIF (PEPRO TECH), 1 μM PD03259010 and 3 μM CHIR99021 (STEMCELL Technologies) and 100 U/ml penicillin, 100 μg/ml streptomycin (15140-122, Gibco). For primed state media, 20 ng/ml Activin, 10 ng/ml FGF2, and 1% KSR were added to the 1:1 DMEM/F12 and Neurobasal medium containing N2 and B27. To investigate DUX binding, an N-terminal FLAG-DUX protein was expressed in our clonal cell lines. In control group, mES cells were treated with doxycycline for 12 h to induce FLAG-DUX expression and then treatment of negative-control Silencer Select siRNA. In siDux group, mES cells were treated with doxycycline for 12 h and then siDux for two days. In CX-5461 treatment group, mES cells were treated with doxycycline for 12 h and then treatment of CX-5461. In siDux+CX-5461 group, mES cells were treated with doxycycline for 12 h and then siDux for 2 days followed by treatment of CX-5461.

**Fluorescence activated cell sorting (FACS) analysis**. E14 wild-type mES cells, snoRNA KO and Pol I degraded mES cells were transfected with 2C::*tdTomato* using Lipofectamine 2000 and selected with 150 μg/ml hygromycin 48 h after transfection and for 7 days. Mouse E14 wild-type cells were subjected to 0.4 μM CX-5461 treatment for 12 h or 2 μM CX-5461 treatment for 12 h. Cells were isolated by FACS to measure the ratio of 2C-like cells. Apoptosis was measured using the Annexin V and DAPI Staining. Flow cytometry was performed on Beckman CytoFLEX LX (Version 9). FACS data was collected using (CytExpert, Version 2.3) and was processed using (FlowJo, Version 9).

**Cell line immunofluorescence staining**. E14 mES cells were grown on gelatin-coated glass coverslips with MEFs and cultured 12 h before fixed with 4% PFA for 10 min, and then permeabilized with 0.5% Triton X-100 in PBS for 20 min at room temperature (RT). The cell samples were blocked in blocking buffer (3% BSA, 2% donkey serum in PBS) for 10 min at RT and then stained with a primary antibody (1:100, Nucleolin, CST, cat. no. 145745; 1:500, NPM1, Sigma, cat. no. B0556; 1:100, Fibrillarin, Abcam, cat. no. ab4566; RPA194, 1:200, Santa Cruz, cat. no. sc-48385) for 12 h at 4 °C. After three washes with 0.1% Triton X-100 in PBS, cells were stained with a secondary antibody (1:200, Goat polyclonal Secondary Antibody to Mouse IgG, Abcam, cat. no. ab150113) for 2–12 h at 4 °C. Followed by washing three times with 0.5% Triton X-100/PBS, DAPI was used for nucleus staining. The samples were then imaged by Zeiss LSM880 fluorescence microscope at a 63× oil objective. For high regulation microscopy imaging, LSM800 with Airyscan module was used.

**Fluorescence recovery after photobleaching (FRAP) analysis**. E14 wild-type and CX-5461 treatment mES cells cultured on MEF cells were grown in LIF/2i conditions and maintained at 37 °C and with 5% CO$_2$ during imaging. Cells were transduced with Lenti-NCL-eGFP/Lenti-NPM1-eGFP/Lenti-FBL-mCherry lentivirus. FRAP experiments were performed on a ZEISS (Jena, Germany) LSM800 confocal laser scanning microscope equipped with a ZEISS Plan-APO 63×/NA1.46

oil immersion objective. Circular regions of constant size were bleached and monitored overtime for fluorescence recovery. Imaging was taken once every 5 s for a total of 10 min. Fluorescence intensity data was corrected for background fluorescence and normalized to initial intensity before bleaching using GraphPad software. Resulting FRAP curves were fitted with Four parameter logistic (4PL) curve.

**siRNA-mediated knockdown in mES cells**. siRNA was transfected into mES cells with Lipofectamine 2000 (Thermo Fisher Scientific). mES cells were seeded into 12-well plate and cultured in LIF/2i medium for overnight. The next day, 800 μl LIF/2i medium without antibiotics was added into each well. Then, the transfect mixture (40 pmol of three independent siRNA targeting each gene/a non-targeting siRNA (negative control, NC) and 2 μl of Lipo 2000 which was diluted in 200 μl Opti-MEM medium (Gibco)) was added into each well and incubated for 6 h at 37 °C followed by exchanging for fresh complete LIF/2i medium, and then cells were collected for RNA extraction about 48 h. The sequences of siRNA are listed in Supplementary Table 1.

**Cell line RNA extraction and qRT-PCR**. Total RNA was isolated from mES cells using miRNeasy kit (217004, QIAGEN) according to the manufacturer's protocol, and 1 μg RNA was reverse transcribed to cDNA with HiScript II Q RT Super Mix (R223-01, Vazyme). Gene expression was analyzed with SYBR-Green qPCR Master mix (Bio-Rad) on Bio-Rad PCR machine (CFX-96 Touch). Each gene was normalized to Actin or Gapdh. All primers used are listed in Supplementary Table 2.

**Oligopaint DNA FISH**. The Oligopaint DNA FISH probes were designed by OligoMiner[74–76]. The probe pools synthesized by Synbio Technologies were used as templates and the dye-labeled secondary probes were produced by Sunya Biotechnology. All sequences used in this work were listed in Supplementary Data 1. Briefly, the synthesized probe pool was first used as template to amplify via 30 PCR cycles and was subsequently purified by ammonium acetate precipitation. Then, the PCR products was used as template to amplify and convert into RNA via an in vitro transcription of high yield (New England Biolabs, E2040S); the RNA product above were converted back into single-stranded DNA via reverse transcription. At last, the product was subjected to the alkaline hydrolysis to remove template RNA and was further purified by ammonium acetate precipitation. For the secondary probe, a 30 bp random oligo was designed attached with Cy3 or Alexa Flour 647 at 5′ end.

For DNA FISH, the cell samples were fixed in 4% Paraformaldehyde (Sigma, 158127) for 10 min and washed two times with PBS, followed by permeabilizing with 0.5% Triton X-100 (Sigma, T8787) in PBS. Then the samples were incubated in 0.1% w/v sodium borohydride (Sigma, 71320) for 10 min and treated with 0.1 M HCl for 5 min. After that, the cell samples were incubated in 0.1 mg/ml RNase A diluted in PBS for 45 min in 37 °C. After washing three times in 2× SSCT (2× SSC + 0.1% Tween-20), the samples were immersed in 50% formamide diluted in 2× SSCT for 15 min at room temperature then transferred to 85 °C for 10 min. The

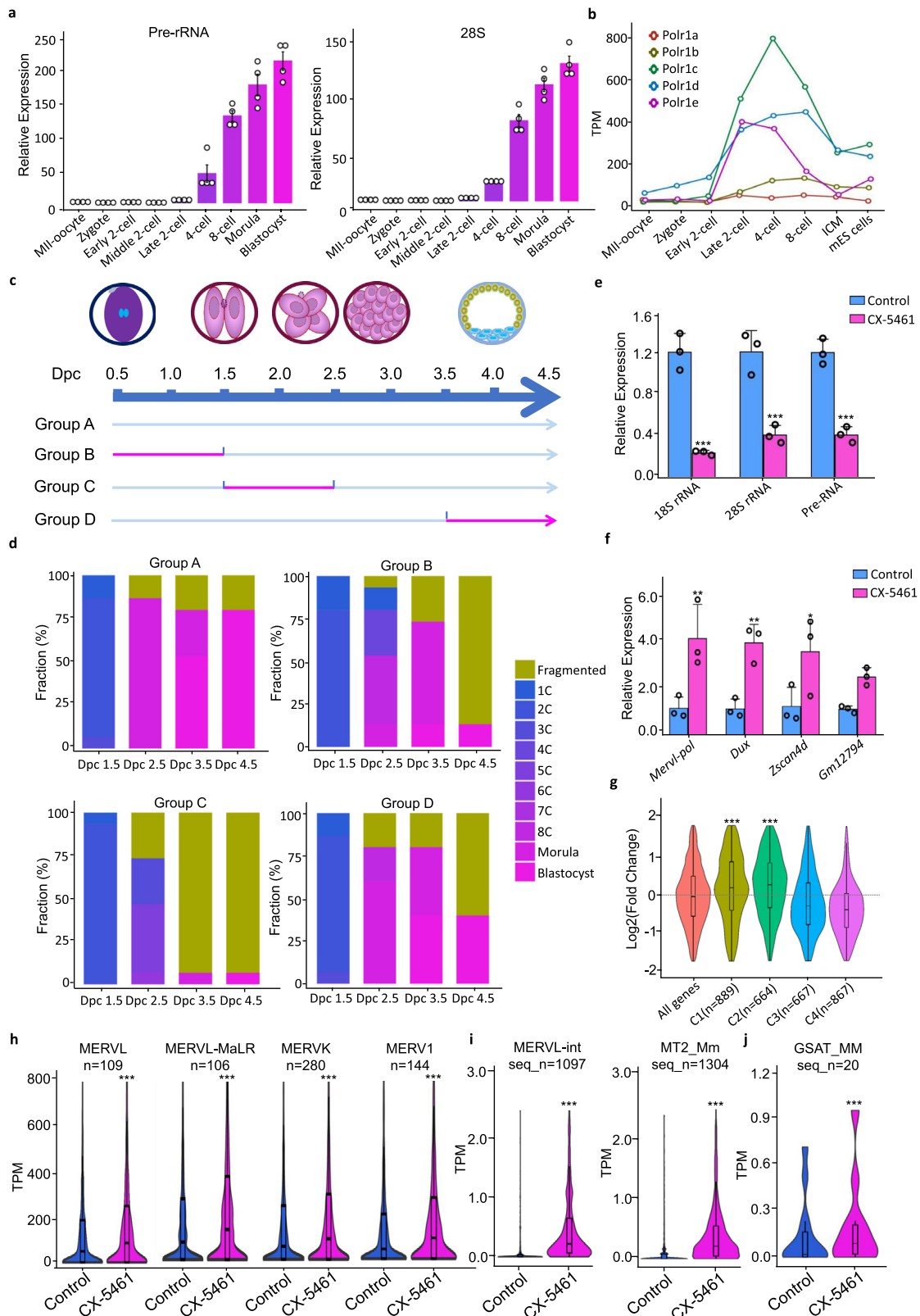

primary probe and secondary probe were freshly mixed into hybridization buffer (2× SSC, 50% formamide, 20% dextran sulfate) at 6 and 1 μM final concentration and dropped on samples. The samples were heated at 85 °C for 20 min and transferred to a 37 °C incubator before hybridization overnight. For the co-immunostaining, the cell samples above were washed three times in PBS and then incubated with primary antibody (1:200, Nucleolin, CST, cat. no. 145745) for 12 h at 4 °C. After three times of washing with PBS, Donkey anti-Rabbit secondary antibody (1:200, Abcam, cat. no. ab150077) was performed and

incubated for 4 h. After washing three times with PBS, DAPI was used for nucleus staining.

**Transmission electron microscope (TEM)**. For transmission electron microscopy, massive mouse E14 ES cells cultured in one 10 cm dish with three days, and then were collected to get rid of feeder cells, and subsequently fixed with 2.5% glutaraldehyde. After that, the cell mass was dispersed into several small pieces and fixed at least 6 h at 4 °C. Next, the cell samples were treated with standard

**Fig. 9 rRNA biogenesis is critically required at the two-cell-to-four-cell stage transition during pre-implantation embryo development. a** Expression of Pre-rRNA and 28S rRNA across different embryo developmental stages; $N = 4$ biologically independent experiments; Data are presented as mean values $+/-$ SEM. SEM standard error of mean. **b** Expression of different subunit genes of RNA polymerase I across different embryo developmental stages using public pre-implantation mouse embryos RNA-seq data; Gene expression levels of two biological replicates in same developmental stage were averaged for plotting. GEO accession code GSE66390. **c** Different schemes of treatment with CX-5461. The 24 h time window for CX-5461 treatment is highlighted in red; Dpc days post-coitum. **d** Stacked bar plots showing fraction of embryos at different developmental stages with the different CX-5461 treatment schemes in Fig. 6c. The numbers of embryos of group A to group D were all 15 embryos. **e** qRT-PCR showing rRNA expression level in blastocysts, after CX-5461 treatment of morula embryos followed by in vitro culture of the treated embryos; 18S rRNA: $p = 2.68\mathrm{E}-06$ (***); 28S rRNA: $p = 1.98\mathrm{E}-05$ (***), Pre-rRNA: $p = 2.17\mathrm{E}-05$ (***), two-way ANOVA; $N = 3$ biologically independent experiments; Data are presented as mean values $+/-$ SEM. SEM standard error of mean. **f** qRT-PCR showing 2C marker gene expression level in blastocysts, after CX-5461 treatment of morula embryos followed by in vitro culture of the treated embryos. Dux: $p = 4.09\mathrm{E}-03$(**), MERVL-pol: $p = 6.82\mathrm{E}-03$(**), Zscan4: $p = 2.56\mathrm{E}-02$(*), Gm12794: $p = 2.82\mathrm{E}-01$, two-way ANOVA; $N = 3$ biologically independent experiments; Data are presented as mean values $+/-$ SEM. SEM standard error of mean. **g** Violin plots demonstrating the expression level changes of stage-specific gene clusters of mouse pre-implantation embryos (as defined in Supplementary Fig. S1) in CX-5461-treated and control blastocyst embryos, C1 vs. All genes: $p = 4.55\mathrm{E}-09$, C2 vs. all genes: $p = 2.30\mathrm{E}-14$, two-sided, Mann–Whitney $U$-test, $N = 2$ biologically independent RNA-seq experiments. GEO accession code GSE166041. **h** Violin plots show expression levels of major ERV gene classes in control blastocyst embryos and CX-5461 treated blastocyst embryos; $N$ denotes the number of sub-classes of ERV genes; MERVL: $p = 5.11\mathrm{E}-05$(***), MERVL-MaLR: $p = 1.65\mathrm{E}-07$(***), MERVK: $p = 3.65\mathrm{E}-07$(***), MERV1: $p = 3.67\mathrm{E}-07$(***), two-sided, Wilcox signed rank test; $N = 2$ biologically independent RNA-seq experiments. GEO accession code GSE166041. **i** Violin plots show expression levels of ERV gene sub-classes of MERVL-int and MT2_Mm in control blastocyst embryos and CX-5461 treated blastocyst embryos; seq_n denotes the number of annotated MERVL-int and MT2_Mm sequences; MERVL-int: $p=2.65\mathrm{E}-18$(***), MT2_Mm: $p = 8.51\mathrm{E}-05$(***), two-sided, Wilcox signed rank test, $N = 2$ biologically independent RNA-seq experiments. GEO accession code GSE166041. **j** Violin plots show expression levels of ERV gene sub-classes of GSAT_MM in control blastocyst embryos and CX-5461-treated blastocyst embryos; seq_n denotes the number of annotated GSAT_MM sequences; $p = 3.01\mathrm{E}-04$(***), two-sided, Wilcox signed rank test, $N = 2$ biologically independent RNA-seq experiments. GEO accession code GSE166041. **g–j** The center line is the median, the bottom of the box is the 25th percentile boundary, the top of the box is the 75th, and the top and bottom of vertical line define the bounds of the data that are not considered outliers, with outliers defined as greater/lesser than ±1.5× IQR, where IQR inter-quartile range. Source data are provided as a Source Data file.

procedures and cut into ultra-thin sections meeting the imaging requirements. Finally, the obtained samples were imaged on FEI Spirit 120 kV LaB6 Routine Cryo-EM Capable Electron Microscope.

**Embryo collection and culture.** The animal facility of Zhejiang University was used to house the C57BL/6J mice. The C57BL/6J mice were cultivated on 12 h light/dark cycle. All embryo experiments were carried out according to the Animal Research Committee guidelines of Zhejiang University. To collect pre-implantation embryos, C57BL/6J female mice (4–6 weeks old) were intraperitoneally injected with 7.5 IU each of PMSG (San-Sheng Pharmaceutical) for 48 h followed by injection of 7.5 IU of hCG (San-Sheng Pharmaceutical). The superovulated female mice were mated with adult males overnight after hCG administration. Embryos at different stages of pre-implantation development were collected at defined time periods after the administration of hCG: 30 h (early two-cell), 44–48 h (two-cell), 54–56 h (four-cell), 68–70 h (eight-cell), 76–78 h (morula) and 92–94 h (blastocysts) in HEPES-buffered CZB medium. Zygotes were collected from ampullae of oviducts and released with hyaluronidase for removing cumulus cells.

**Embryo immunofluorescence staining.** Embryos were first fixed with 1 and 2% paraformaldehyde (PFA) in 1× PBS for 3 min sequentially, followed by treatment with 4% PFA for 30 min at room temperature (RT). Embryos were washed three times with 1× PBS, permeabilized for 15 min in PBS/0.25% Triton X-100 and blocked in blocking buffer (PBS/0.2% BSA/0.01% Tween-20) for 1 h at RT, followed by incubation overnight with primary antibodies (1:200, Nucleolin, CST, 145745; 1:400, NPM1, Sigma, B0556; 1:200, Fibrillarin, Abcam, ab4566; RPA194, 1:50, Santa Cruz, sc-48385) at 4° or for 1 h at 37 °C. Subsequently, embryos were washed four times for 10 min each and incubated with a secondary antibody (daylight 488-conjugated anti-rabbit, 1:100 or daylight 594-conjugated anti-mouse, 1:200) for 1 h at 37 °C and washed three times with PBS. Nuclei were stained with DAPI for 1 min. Embryos were observed under Zeiss LSM880 fluorescence microscope at 63× magnification with an oil immersion objective.

**Embryo collection, cDNA synthesis, and qRT-PCR.** Ten embryos were rinsed in 0.2% BSA/PBS without Ca2+ and Mg2+ and placed in 0.2 ml PCR tube, immediately transferred in liquid nitrogen, and stored at −80 °C. It was hybridized with 0.5 μl oligo-dT30 (10 μM, Takara) and 1 μl random (1 M) and 1 μl dNTP mix (10 mM) in 2 μl cell lysis buffer (2 U RNase inhibitor, 0.01% Triton X-100) at 72 °C for 3 min. Then, the reaction was immediately quenched on ice. After the reaction tube was centrifuged, 2 μl was used for reverse transcription with Super Script II Reverse Transcriptase 5× first strand buffer, 0.25 μl RNase inhibitor (40 U), 0.06 μl MgCl2 (1 M), 2 μl betaine (5 M), and 0.5 μl Reverse Transcriptase Superscript II (Takara). Reverse transcription was carried out in the thermocycler at 42 °C for 90 min, 70 °C for 15 min, and then 4 °C for holding. Subsequently, cDNA was diluted 1:10 (v/v) with RNase free water and used for a qPCR amplification in triplicate with SYBR

Green Master (Vazyme) in a final volume of 20 μl per reaction as manufacturer's instructions.

**Embryo RNA-seq library preparation and sequencing.** Embryos were collected (five embryos per sample) in 0.2 ml PCR tubes with a micro-capillary pipette and processed into cDNA with Superscript II reverse transcriptase. The cDNA is amplified with KAPA Hifi HotStart using 12 cycles. Sequencing libraries were constructed from 1 ng of pre-amplified cDNA using DNA library preparation kit (TruePrep DNA Library Prep Kit V2 for Illumina, Vazyme). Libraries were sequenced on a HiSeq-PE150, with paired end reads of 150 bp length each.

**Bulk RNA-seq library preparation and sequencing.** A total amount of 2 μg RNA per sample was used as input materials for the RNA sample preparation. mRNA was purified from total RNA using poly-T oligo-attached magnetic beads. Purified mRNA was fragmented at 94 °C for 15 min by using divalent cations under elevated temperature in NEBNext first strand synthesis reaction buffer (5×). First strand cDNA was synthesized using random primer and ProtoScript II reverse transcriptase in a preheated thermal cycler as follows: 10 min at 25 °C; 15 min at 42 °C; 15 min at 70 °C. Immediately finished, second strand synthesis reaction was performed by using second strand synthesis reaction buffer (10×) and enzyme mix at 16 °C for 1 h. The library fragments were purified with QiaQuick PCR kits and elution with EB buffer, then terminal repair, A-tailing and adapter added were implemented. The products were retrieved, and PCR was performed for library enrichment. The libraries were sequenced on an Illumina platform.

**10× single-cell mRNA library preparation and sequencing.** Single-cell suspensions of control and CX-5461 treated mES cells were resuspended in DPBS-0.04% BSA at $1 \times 10^6$ cells/mL. Then scRNA-seq libraries were generated from the 10× Single Cell 3′ Solution Reagents V2 according to the manufacturer's protocol (10× Genomics). After the GEM-RT incubation, barcoded-cDNA was purified with DynaBeads cleanup mix, followed by 10-cycles of PCR amplification (98 °C for 3 min; [98 °C for 15 s, 67 °C for 20 s, 72 °C for 1 min] × 10; 72 °C for 1 min). The total cDNA of single-cell transcriptomes was fragmented, double-size selected with SPRI beads (Beckman), followed by 12 cycles sample index PCR amplification (98 °C for 45 s; [98 °C for 20 s, 54 °C for 30 s, 72 °C for 1 min] × 10; 72 °C for 1 min), then another double-size selection with SPRI beads was performed before sequencing. Libraries were sequenced on the Illumina HiseqX10 platform according to the manufacturer's instructions (Illumina). Read 1 and Read 2 (paired end) were 150 bp, and the length of index primer was designed as 8 bp.

**ChIP-seq library preparation and sequencing.** mES cells were cross-linked in 1% formaldehyde for 10 min at 37 °C, followed by adding glycine to a final concentration of 125 mM and incubated for 5 min at room temperature. Spin the cells for 5 min at 4 °C, 111×g, and wash twice in ice-cord PBS. Cell pellet was resuspended with lysis buffer containing 1× Protease Inhibitor Cocktail and incubated on ice for 10 min, then

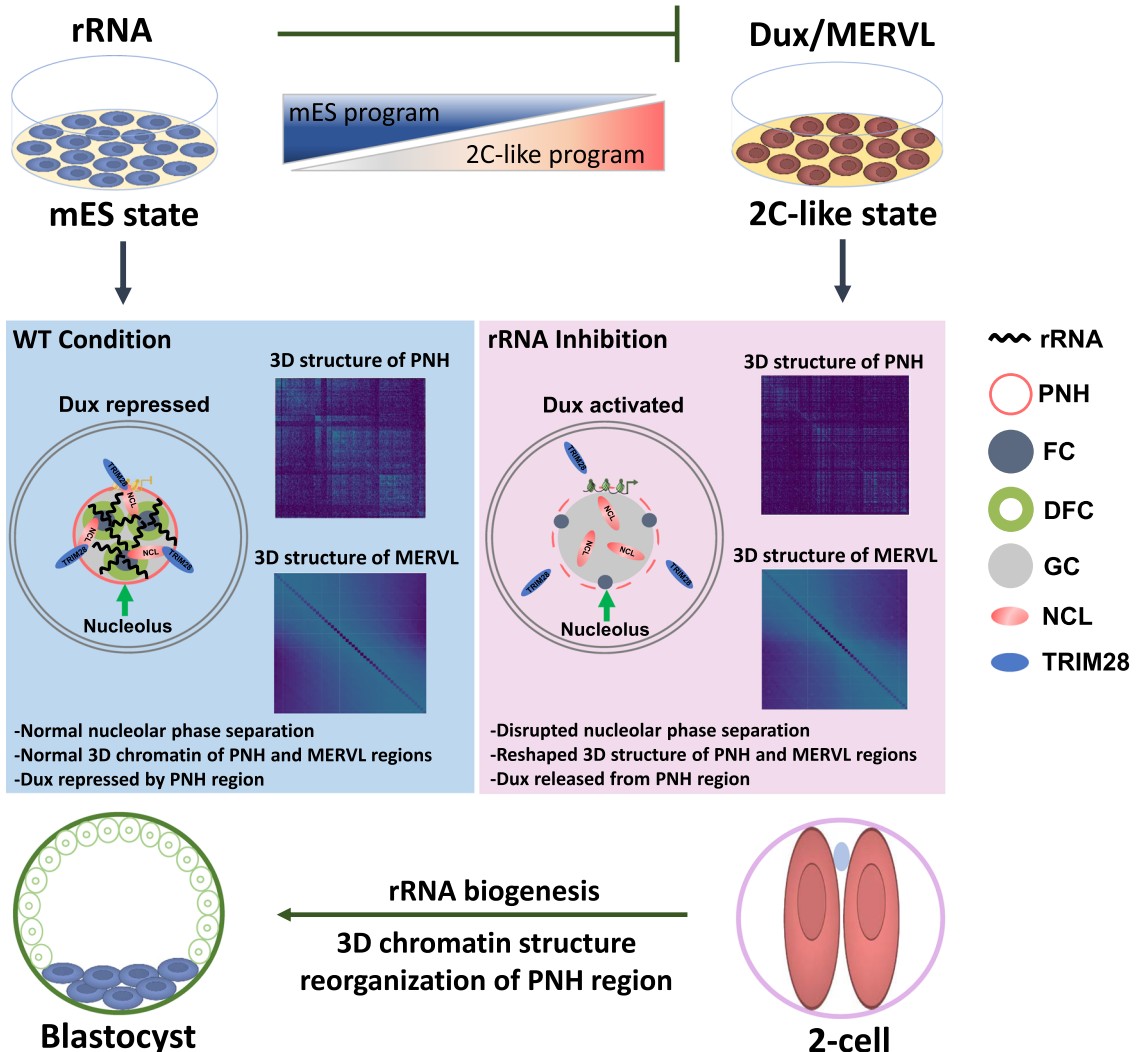

**Fig. 10 A mechanistic model for the role of rRNA biogenesis in regulating mouse 2C-like state.** In the unperturbed mES cells, nucleolar integrity mediated by rRNA biogenesis maintains the normal liquid-liquid phase separation (LLPS) of nucleolus and the formation of peri-nucleolar heterochromatin (PNH) containing *Dux*, and this normal nucleolar LLPS facilitates NCL/TRIM28 complex occupancy on the *Dux* locus and repression of Dux expression. In contrast, in the rRNA biogenesis-inhibited mES cells, the natural liquid-like phase of nucleolus is disrupted, causing dissociation of the NCL/TRIM28 complex from PNH and changes of epigenetic state and 3D structure of PNH, which eventually leads to *Dux* released from PNH, activation of 2C-like program and transition of mES cells to 2C-like cells.

vortexed vigorously for 10 s and centrifuged at 825×*g* for 5 min. The pellet was re-suspended in ChIP lysis buffer and incubated on ice for 10 min and vortexed occasionally. Afterwards, the chromatin lysate was transferred to a 1.5 mL centrifuge tube and chromatin sheared using water bath sonication with the following conditions: shear 15 cycles at 4 °C, 15 s on, 30 s off. Centrifuge and transfer supernatant to a new tube. Taking 5 μL (1%) from the 500 μL containing sheared chromatin as input. Each chromatin sample was incubated with antibodies for H3K9me3 Rabbit polyclonal antibody (1:100, abcam, cat. no. ab8898), H3K27me3 Rabbit mAb (1:50, CST, cat. no. 9733), H3K4me3 Rabbit mAb (1:50, CST, cat. no. 9751), H3K27ac Rabbit mAb (1:100, CST, cat. no. 8173), Nucleolin (D4C7O) Rabbit (1:100, CST, cat. no. 14574), TRIM28 Mouse monoclonal (20C1) (1:100, abcam, cat. no. ab22553) overnight on a rotating platform at 4 °C. The next day, the sample was incubated with protein A+G magnetic beads (HY-K0202, MCE) for 3 h at 4 °C with rotation. The beads-antibody/chromatin complex was washed three times with low-salt wash buffer and once with high-salt wash buffer and resuspended with elution buffer. The elute DNA was treated with RNase A at 42 °C for 30 min, then treated with protease K at 60 °C for 45 min followed by heat inactivation at 95 °C for 15 min. The purified DNA was subjected to library preparation or analyzed by qPCR. The libraries were sequenced on an Illumina platform. All primers of ChIP-qPCR used are listed in Supplementary Table 3.

**In situ Hi-C library preparation and sequencing.** 10⁶ cells were cross-linked for 10 min with 1% final concentration fresh formaldehyde and quenched with 0.2 M final concentration glycine for 5 min. The cross-linked cells were subsequently lysed in lysis buffer (10 mM Tris-HCl (pH 8.0), 10 mM NaCl, 0.2% NP40, and complete protease inhibitors (Roche)). The extracted nuclei were re-suspended with 150 μL 0.1% SDS and incubated at 65 °C for 10 min, then SDS molecules were quenched by adding 120 μL water and 30 μL 10% Triton X-100, and incubated at 37 °C for 15 min. The DNA in the nuclei was digested by adding 30 μL 10× NEB buffer 2.1 (50 mM NaCl, 10 mM Tris-HCl, 10 mM MgCl₂, 100 μg/mL BSA, pH 7.9) and 150U of MboI, and incubated at 37 °C overnight. On the next day, the MboI enzyme was inactivated at 65 °C for 20 min. Next, the cohesive ends were filled in by adding 1 μL of 10 mM dTTP, 1 μL of 10 mM dATP, 1 μL of 10 mM dGTP, 2 μL of 5 mM biotin-14-dCTP, 14 μL water and 4 μL (40 U) Klenow, and incubated at 37 °C for 2 h. Subsequently, 663 μL water, 120 μL 10× blunt-end ligation buffer (300 mM Tris-HCl, 100 mM MgCl₂, 100 mM DTT, 1 mM ATP, pH 7.8), 100 μL 10% Triton X-100 and 20 U T4 DNA ligase were added to start proximity ligation. The ligation reaction was placed at 16 °C for 4 h. After ligation, the cross-linking was reversed by 200 μg/mL proteinase K (Thermo) at 65 °C overnight. DNA purification was achieved through QIAamp DNA Mini Kit (Qiagen) according to manufacturer's instructions. Purified DNA was sheared to a length of ~400 bp. Point ligation junctions were pulled down by Dynabeads® MyOne™ Streptavidin C1 (Thermo-fisher) according to manufacturer's instructions. The Hi-C library for Illumina sequencing was prepped by NEBNext® Ultra™ II DNA library Prep Kit for Illumina (NEB) according to manufacturer's instructions. The final library was sequenced on the Illumina HiSeq X Ten platform (San Diego, CA, United States) with 150PEmode. Two replicates were generated for one group material.

**Bulk RNA-seq data analysis**. All bulk RNA-seq reads were trimmed using Trimmomatic software (Version 0.36) with the following settings "ILLUMINA-CLIP:TruSeq3-PE.fa:2:30:10 LEADING:3 TRAILING:3 SLIDINGWINDOW:4:15 MINLEN:36"[77] and were further quality-filtered using FASTX Toolkit (http://hannonlab.cshl.edu/fastx_toolkit/) fastq_quality_trimmer command with the minimum quality score 20 and minimum percent of 80% bases that has a quality score larger than this cutoff value. The high-quality reads were mapped to the mm10 genome by HISAT2 (v2.1.0), a fast and sensitive spliced alignment program for mapping RNA-seq reads, with -dta parameter[78]. PCR duplicate reads were removed using Picard tools (https://broadinstitute.github.io/picard/) (v2.18.2). For subsequent analysis on single-copy genes, only uniquely mapped reads were kept. Considering the multi-mapping of reads derived from repeat sequences, we used all mapped reads for further analysis. The expression levels of genes and repeat sequences were independently calculated by StringTie[79] (Version v1.3.4d) with -e -B -G parameters using Release M18 (GRCm38.p6) gene annotations downloaded from GENCODE data portal and annotated repeats (RepeatMasker) downloaded from the UCSC genome browser, respectively. To obtain reliable and cross-sample comparable expression abundance estimation for each gene and each family of repeat sequence, reads mapped to mm10 were counted as TPM (Transcripts Per Million reads) based on their genome locations. Differential expression analysis of genes in different samples was performed by DESeq2 (v1.32.0) using the reads count matrix produced from a python script "prepDE.py" in StringTie website (http://ccb.jhu.edu/software/stringtie/). We selected the genes with stage-specific scores larger than 0.2 to perform K-mean clustering analysis using pheatmap (v1.0.10) R package. The stage-specific scores of genes expressed during mouse early embryo development were obtained by entropy-based measure[80].

Unsupervised hierarchical clustering was carried out to compare the transcriptomes of mES cells from our study and other reports of 2C-like cell (2C::tdTomato$^+$ and 2C::tdTomato$^-$ (GEO accession GSE33923); Zscan4_Em$^+$ and Zscan4_Em$^-$ (GEO accession GSE51682); Kap1_KO and Kap1_WT (GEO accession GSE74278); CAF1_KO and CAF1_WT (GEO accession GSE85632), Dux_GFPpos and Dux_GFPneg (GEO accession GSE85632); LINE1 ASO and RC ASO (GEO accession GSE100939); Dppa4_GFPpos and Dppa4_GFPneg (GEO accession GSE120953), NELFA_GFPpos and NELFA_GFPneg (GEO accession GSE113671); Lin28a_KO and Lin28a_WT (GEO accession GSE164420). Additionally, pre-implantation mouse embryos of different developmental stages (GSE66532) were included for comparison. TPM were obtained for each sample using the StringTie described above. Only genes that were expressed TPM ≥ 0 were included for analysis. A log2 transformation was applied after adding one pseudo-count (log2[TPM+1]). The ComBat function from R package sva (https://bioconductor.org/packages/release/bioc/html/sva.html) (v3.40.0)[81] was applied on log2 expression values to correct for batch effects caused by different experiments and sequencing platforms.

**Single-cell RNA-seq data analysis**. For single cell RNA-seq, we used 10× Genomics system following the manufacturer's protocol. We followed the previously published pipeline[82] to produce digital gene expression matrices of the droplet microfluidics-based single-cell RNA-seq sequencing data derived from control and CX-5461 treated mouse ES cells. Single-cell gene expression matrix was further analyzed with Seurat[83] (https://satijalab.org/seurat/) (v2.3.4). We excluded the genes with expressed cell number smaller than three and the cells with nUMIs smaller than 500 or the expression percentages of mitochondrial genes larger than 0.2 and used 16 principle components (PCs) for tSNE analysis. Especially, we modified the published pipeline by substituting the aligner of Bowtie with HISAT2 (v2.1.0) for calculating the expression of repeat sequences. Considering the multi-mapping of reads derived from repeat sequences, we used all mapped reads for further analysis.

**ChIP-seq and ATAC-seq data analysis**. To tailor and filter ATAC-seq and ChIP-seq reads, we used the same procedure as RNA-seq reads processing. To avoid the potential effects of inconsistent sequencing depths on subsequent data analyses, we randomly sampled equal numbers read pairs from each experimental sample. For each sample, the ATAC-seq and ChIP-seq reads were first aligned to mm10 genomes using Bowtie2 (version 2.3.4.1)[84]. The ATAC-seq reads were aligned with the parameters: -t -q -N 1 -L 25 -X 2000 no-mixed no-discordant. The ChIP-seq reads were aligned to mm10 with the options: -t -q -N 1 -L 25. The ATAC-seq reads were aligned with the parameters: -t -q -N 1 -L 25 -X 2000 no-mixed no-discordant. The ChIP-seq reads were aligned to mm10 with the options: -t -q -N 1 -L 25. For meta-analysis of genome regions, all unmapped reads, multiple mapped reads, and PCR duplicates were removed. For demonstrating the sequencing signal around *Dux* locus in UCSC genome browser, we maintained all multiple mapped reads for visualization. The bamCoverage and bamCompare commands contained in deepTools[85] (version 2.5.3) were adopted for downstream analysis. Using BamCoverage command with the parameters: -normalizeUsing BPM -of bigwig -binSize 100, we normalized the raw reads signal to Bins per Million mapped reads (BPM) signal and converted the alignment bam files to bigwig signal files. The bigwig files were imported into UCSC genome browser for visualization. To minimize the effect of chromatin structure and sequencing bias in our ChIP-seq data, we corrected ChIP-seq signal using log$_2$ ratio transformation between H3K9me3 signal and input signal by BamCompare command. We only considered

the log$_2$ ratio larger than 0 as effective ChIP-seq signals. The "computeMatrix" and "plotProfile" commands of deepTools were used to produce the reads density distribution plot of ATAC-seq and ChIP-seq signal in the given genomic region. For meta-analysis of sequencing signals in L1 regions, we only used their subsets which have overlaps with Inactive Hub regions. Homer (v4.11)[86] was used for motif discovery and enrichment analysis. For motifs across gene promoters, the search space was defined as a 4 kilobase (kb) window centered at the transcription start site (findMotifs.pl geneInput.txt mouse out/ -start -2000 -end 2000 -len 8,12 -p 4).

**Hi-C data analysis**. The paired-end reads of fastq files were aligned, processed, and iteratively corrected using HiC-Pro software (version 2.11.1)[87]. Firstly, short sequencing reads were cleaned and then independently mapped to the mouse mm10 reference genome (https://hgdownload.soe.ucsc.edu/downloads.html#mouse) using bowtie2 aligner (v2.3.5.1) with end-to-end algorithm and "-very-sensitive" option. To rescue the chimeric fragments spanning the ligation junction, the ligation site was detected and the 5′ fraction of the reads was aligned back to the reference genome. Unmapped reads, multiple mapped reads and singletons were then discarded. Pairs of aligned reads were then assigned to *MboI* restriction fragments. Read pairs from the uncut DNA, self-circle ligation and PCR artifacts were filtered out and the valid read pairs involving two different restriction fragments were used to build the contact matrix. To eliminate the possible effects on data analyses of variable sequencing depths, we randomly sampled equal numbers read pairs from each condition for downstream analyses involving comparison analyses between conditions. Valid read pairs were then binned at a 40, 150, and 500 kb resolutions by dividing the genome into bins of equal size. The binned interaction matrices were then corrected by Knight–Ruiz matrix balancing method using hicCorrectMatrix command with the parameter–correctionMethod KR in HiCExplorer (https://hicexplorer.readthedocs.io/en/latest/) (v3.3)[88]. The Observed/Expected (O/E) Hi-C matrix was obtained by HiCExplorer hicTransform command with–method obs_exp_norm option. Pearson correlation coefficients Hi-C matrix was obtained by HiCExplorer hicTransform command with–method pearson option.

A and B compartments were identified using the first eigenvector (PC1) from principal component analysis on correlation Hi-C matrix[89]. We used Homer software (v4.11) with parameters -res 500,000 -window 1,000,000 to obtain the PC1 value based on Pearson Correlation Coefficients (PCC) Hi-C matrix. Sometimes the entry signs of PC1 need to be inverted to ensure that we are assigning the correct signs to individual regions. As GC content is well correlated with A and B compartments[90], we calculated the GC content of each region and inverted the eigenvector sign if the average GC content of negative-eigenvector entries is higher than that of positive-eigenvector entries. To obtain the heatmap plot of enrichment of A/B interaction, an A/B compartment profile for each chromosome was then separated into five bins: (min to 20th percentile), (20th percentile to 40th percentile), etc. For each pair of bins (25 pairs total), the averaged O/E values were then calculated for loci belonging to each pair of bins. The compartment strength was calculated as the natural logarithm of (AA*BB)/AB$^2$. AA was defined as the sum of rows 1–3 and columns 1–3 of compartment enrichment matrix. BB was defined as the sum of rows 3–5 and columns 3–5 of compartment enrichment matrix. AB was defined as the sum of rows 1–3 (rows 3–5) and columns 3–5 (column 1–3) of compartment enrichment matrix. For the error bar in evaluating the compartment strength, we obtained 100 5 × 5 compartment enrichment matrices by bootstrapping. For each pixel of the 5 × 5 compartment enrichment map, we took all the O/E values that contributed to this pixel and took a random sample with replacement of the same size that the contributing values. We then proceeded with downstream for each of the 100 reshuffled maps.

TADs and loops annotated in CH12.LX were obtained from[59] and lifted over to the mm10 genome version using the UCSC genome browser liftOver tool. Aggregated plots of TAD enrichment map were obtained by averaging O/E values over annotated TAD positions at 40 kb resolution[91]. For each domain of length $L$, a map for the region ((Start $- L$) to (End $+ L$)) was obtained. This produced a contact that is three times bigger than a given domain. This contact map was then rescaled to a (90 × 90) pixel map using linear interpolation and block-averaging. In the resulting map, the mid-region pixels 30 to 60 correspond to the TAD body. TAD strength for boxplots was quantified as the ratio of two numbers. The first number is the within-TAD intensity: the sum of the central square of the enrichment map, rows 30–59 and columns 30–59. The second number is the between-TAD intensity, ½ of the sums of the regions [0:30, 30:60] and [30:60, 60:90]. Aggregated plots of loop enrichment map were obtained by averaging O/E values around loop anchors of 310 kb window at 10 kb resolution. To quantify loop strength, we took an average Hi-C O/E values at the loop base, averaging over 31 × 31 kb square centered at the bin containing the loop base. We then divided it by the same average, but in the offset loops (loop positions were offset by the corresponding loop length).

Hi-C de novo boundary aggregate plots at MERVLs are centered on 5′ to 3′-oriented MERVL and show a window of 2 mb around the MERVL element at 40 kb resolution. For illustrating the change of insulation around MERVL genes, the log$_2$ transformed Hi-C O/E matrix was further scaled by z-score normalization across each row. We calculated the insulation score as originally defined[60] with minor modifications. For each region $i$ in the genome, we calculated the average number of O/E interactions in 40 kb Hi-C matrix in a quadratic window with the lower left corner at $(i − 1, i + 1)$, and the top right corner at $(i − 5, i + 5)$, where 5 is the window size in

bins. We normalized insulation scores by dividing each region's score by the average scores of the nearest 50 regions, and log$_2$-transforming the resulting vector, thus accounting for local biases in insulation score. Visualization of Hi-C matrix was carried out by the Juicer tool (https://github.com/aidenlab/juicer) (v1.9.9)[92] and R software (https://www.r-project.org/) (v4.0.2). For heatmap visualization in Juicer, we converted valid read pairs into .hic format files with the juicer tool pre command.

**Statistical analysis**. All statistical analyses for Next Generation Sequencing (NGS) data were performed with R (v4.0.2)/Bioconductor (v3.10) software utilizing custom R scripts. The other statistical analyses were performed with GraphPad Prism 8 software. For *p*-value produced by GraphPad, only 15 digits after decimal, the upper limit of this software, are shown in the figure legend. Details of individual tests are outlined within each figure legend, including number of replications performed (*n*) and the reported error as standard error of the mean (SEM). All statistics are *p < 0.05, **p < 0.01, ***p < 0.001, and were calculated by Wilcox signed rank test (for paired samples), Mann–Whitney *U*-test (for independent samples), two-way ANOVA and chi-square test as described in the figure legends.

**Reporting summary**. Further information on research design is available in the Nature Research Reporting Summary linked to this article.

## Data availability

The data that support this study are available from the corresponding author upon reasonable request. All bulk RNA-seq, single-cell RNA-seq, ChIP-seq, ATAC-seq, and Hi-C data generated in this study have been deposited in the National Center for Biotechnology Information (NCBI) Gene Expression Omnibus (GEO) database under the accession codes GSE166041 and GSE164420. Previously published RNA-Seq data that were re-analyzed here are available under accession codes GSE33923, GSE51682, GSE74278, GSE85632, GSE100939, GSE120953, GSE113671, GSE97778, and GSE66582. Published ChIP-seq data for DUX are available under accession code GSE85632. Published ChIP-seq data for P53 are available under accession code GSE26360. Published ATAC-seq data are available under accession codes GSE66390 and GSE85632. Published Hi-C data of mouse pre-implantation embryos are available under accession code GSE82185. Published Hi-C data of lymphoblastoid cells are available under accession code GSE63525. Supplementary Table 4 provides a summary for all analyzed NGS datasets used in this study. Source data are provided with this paper.

## Code availability

The codes used for the analysis reported in this study were freely available at https://github.com/huayu1111/rRNAproj.

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

## Acknowledgements

We thank Dr. Xiong Ji (Peking University) for providing the Pol I degraded mES cell lines. We thank Dr. Todd Macfarlan, Dr. Lingling Chen and Dr. Juan Guan for kind discussion. We thank Dr. Li Shen (Zhejiang University) for providing 2C::tdTomato Reporter plasmid. We thank Yanwei Li and Guifeng Xiao from the core facility platform of Zhejiang University School of Medicine for their technical support. We thank High-Performance Computing Platform in Center of Cryo-Electron Microscopy of Zhejiang University for providing computational support. J.Z. is supported by the National Key Research and Development Program of China (No.2018YFC1005002, No.2018YFA0107100, No.2018YFA0107103), the National Natural Science Foundation of China (No.31871453, No.91857116), the Zhejiang Natural Science Foundation of China (No.LR19C120001) and the Zhejiang Innovation Team Grant (2019R01004). H.Y. is supported by is supported by the National Natural Science Foundation of China (No.32100632) and the Zhejiang Natural Science Foundation of China (No.LQ21C120002).

## Author contributions

J.Z. and H.Y. conceived and designed the study and experiments. H.Y. and J.Z. wrote the manuscript with contributions from all authors. H.Y. designed and performed all computational analysis. Z.S., T.T., H.P., A.L., Y.Z., and L.C. performed the molecular experiments in mES cells. J.Z., L.Z., and J.C. performed the experiments in mouse embryos. H.Y., Y.X., and Y.L. assisted with the experiment sample preparation. M.C. provided computational support. J.S., Z.X., P.Q., C.L., S.G., and G.D gave critical suggestions about the study design and paper writing. All authors analyzed the results and approved the final version of the manuscript.

## Competing interests

The authors declare no competing interests.
