## [Peer Review File · Nature Communications]

rRNA Biogenesis Regulates Mouse 2C-like State by 3D Structure Reorganization of Peri-Nucleolar HeterochromatinREVIEWER COMMENTS

Reviewer #1 (Remarks to the Author):

The nucleolus is the organelle for ribosome biogenesis and for sensing various types of stress. In this manuscript, Yu et al. provided a novel perspective on its role in regulating stem cell fate. The authors present evidence that nucleolar stress induced by interfering rRNA biogenesis can drive a 2-cell stage embryo-like (2C-like) transcriptional program and induce an expanded 2C-like cell population in mouse embryonic stem (mES) cells. It is well-known that the liquid-liquid phase separation (LLPS) mediated by rRNA and nucleolar proteins maintains the nucleolar integrity and the formation of peri-nucleolar heterochromatin (PNH). The authors found that Pol-I inhibitor CX-5461 but not other cellular stress inducers treatments caused RNA biogenesis defect and disrupted LLPS of the nucleolus, causing dissociation of the NCL/TRIM28 complex from PNH and changes of epigenetic states and reorganization of the 3D structure of PNH. Consequently, Dux, a 2C program transcription factor gene, is released from the PNH region and activates the 2C-like program. The in vivo functional relevance of this regulatory axis was validated by demonstrating that embryos with rRNA biogenesis defect are incompatible to develop from 2-cell (2C) to 4-cell stage, with delayed repression of 2C/ERV genes and a transcriptome skewed toward earlier cleavage embryo signatures. The results in this study highlight that nucleolar integrity maintained by rRNA-mediated LLPS and 3D chromatin structure regulates the fate transition of mES cells to 2C-like cells, and that rRNA biogenesis is a critical regulator during the 2-cell-to-4-cell transition of murine preimplantation embryo development.

Overall, this is an interesting story with novel insights into our understanding of nucleolus functions in cell fate determination and early development. The findings should also provide new ways in reprogramming mES cells back to 2C-like cells by manipulating the ribosome biogenesis process through LLPS and 3D chromatin structure remodeling at the PNH. There do exist a few major and minor issues that should be addressed to further improve the quality of this manuscript.

Major critiques:

1. It is unclear how long CX-5461 treatment was used for the study. Detailed information such as dosage and time length of CX-5461 treatment for the RNA-seq samples have not been mentioned in this manuscript. Since 2CLCs are triggered by CX-5461 in ESCs, what will happen if rRNA biogenesis inhibition is sustained by long-term Pol-I inhibition? Will a higher percentage of 2CLCs be maintained by CX-5461? Will ESCs be apoptotic? differentiating? or enter a diapause status like mTOR inhibition?
2. Have the authors ever compared the rRNA transcription level between the sporadically 2C like cells vs the non-2C like mESCs? This could be an important piece of data to further substantiate the CX-5461 treated datasets.
3. There are hundreds of snoRNAs within the mouse genome, which snoRNA has been knocked out for the snoRNA knockout cell line? In addition, the knockout of snoRNA and degradation of Pol I need to be confirmed by qPCR and WB.

Minor points:

1. Fig. 2d-e mentioned "mislocations", what kind of mislocation should be specified. In addition, it seems obvious that reduced signal density as well besides the mislocation.
2. In Fig. 2f-h, since the authors used Lenti-NCL-mCherry/ Lenti-NPM1-mCherry/ Lenti-FBL-mCherry for the FRAP experiment, it is better to keep the pseudo color consistent with the fluorescence protein (mCherry). And the overexpression level between the control and CX5461-FBL is not comparable in fig. 2h.
3. The references for Inactive Hub, NAD, L1 in figure 4 should be mentioned in the main text.
4. Fig 6c, one mislabeling for a bar chart, it should be "Pre-rRNA". Fig. 6d, the color-coding is hard to distinguish different development stage. Fig. 6e panel "e" is missing.
5. CX-5461 or CX5461 consistency issue.
6. page 9 line 11, (Figure.5m) is actually (Fig.4m).
7. It is better if the author can clearly specify which datasets are their own or from public resources. i.e., provide ref# or GES# in the figure legend, or provide a Suppl Table for all data they have used.
8. The authors did a HiC-seq analysis in ESCs with Pol-I inhibitor treatment. It will be interesting

by more global analysis from their HiC-seq data, but not just focusing on the inductive hub, NAD, and L1 regions.

9. Pol I protein (PRA1): PRA1 does not sound right. Please double check the accuracy.

Reviewer #2 (Remarks to the Author):

The nucleolus is a nuclear body for ribosome biogenesis, which consists of multiphase liquid droplets. Yu et al. reported that nucleolar stress induced by interfering rRNA biogenesis could drive 2C-like transcriptional program and increased 2C-like cell population in mES cells. The process was accompanied by not only the structural changes of nucleoli but also the reorganization of the 3D chromatin structure: a 2C program transcription factor gene *Dux* was dissociated from peri-nucleolar heterochromatin regions and became active. Yu et al. claimed that rRNA biogenesis and matured nucleoli are indispensable for the 2-cell to 4-cell transition in mouse embryos. The authors also claimed that the structural integrity of nucleoli is maintained by rRNA mediated-liquid-liquid phase separation (LLPS) and plays a role in transcriptionally repressing 2C program for the transition from the 2-cell to the 4-cell stage embryos when zygotic genes are activated. Since the LLPS is a hot topic in cell biology, the issues in this manuscript are timely and potentially intriguing in the related fields. However, the novelty of the main message is not clear to me. In addition, there are several concerns to be addressed. My specific comments are as follows:

Major comments:

- 1) It is unclear what is new on how the nucleolar LLPS involves the establishment and the maintenance of peri-nucleolar heterochromatin including *Dux* loci. Although the authors claimed that the nucleolin is a key player, it was already shown that it represses the *Dux*/2C program in Ref. 14 (PMID: 29937225).
- 2) Fig.1a-d. Rapamycin inhibits ribosome biogenesis, including rRNA transcription and processing (e.g., PMID: 15004009; PMID: 14612424). However, the rapamycin treatment did not seem to induce 2C-like cells. The authors should clarify this point.
- 3) "the percentage of tdTomato positive cells was maintained even at 24 hours after CX-5461 withdrawal (Fig.1k)." I do not agree because the density plots after CX-5461 removal in Fig. 1k look different from that with just CX-5461 treatment (Fig. 1j).
- 4) Fig.2b-e. I found that DAPI signals are enriched around peri-nucleolar region after the CX-5461 treatment. This seems to be inconsistent with a feature of 2-cell like cells: a loss of chromocenters as shown in Ref. 14 (PMID: 29937225). I am not convinced that CX-5461 treatment really disassembled peri-nucleolar heterochromatin and subsequently induced the subsequent 2-cell like cells.
- 5) "Altogether, these results demonstrated the rRNA biogenesis defect led to impaired nucleolar phase separation". What did "impaired phase separation" mean? Even after CX-5461 treatment, some parts of the nucleolar structure remained and still phase-separated in Fig. 2a-c. The nucleolar structure after Pol I inhibition or depletion was well characterized in PMID: 33055158.
- 6) Extended Data Fig. S2f. I found only tiny differences between the ChIP-seq signals with and without CX-5461 treatment. Fig. 2i-2p, particularly on the NCL and TRIM28 signals, are not convincing.
- 7) Fig. 4. It was reported that CX-5461 is a G-quadruplex stabilizer in the human genome (PMID: 28211448). I wonder whether the authors can exclude the possibility that CX-5461 treatment drives 3D chromatin reorganization independently of the nucleolar stress.
- 8) Fig. 5k. Pol I degradation increased the percentage of 2C::tdTomato positive cells to about 20%, while CX-5461 treatment increased it to just 2~4 % in Fig. 2l. I wonder why it is so different between them.

Minor comments:

- 1) "we generated two rRNA biogenesis-inhibited mES cell lines: 1) a line with degraded Pol I protein (PRA1)...". "PRA1" is a typo; it should be "RPA1".
- 2) Fig. 5b. I found that the nucleolar shape after Pol I degradation does not look similar to ones treated with CX-5461 shown in Figs. 2b and 4m. Specifically, in Fig. 5b, the nucleolus after Pol I degradation does not show round shape, contrasting with the ones after CX5461 treatment. Could the authors clarify this point?
- 3) Extended Data Fig. S6f. Fibrillarin (FBL) distribution looks similar between control blastocyst embryos and CX-5461-treated blastocyst embryos. Could the authors explain why?
- 4) I wonder whether CX-5461 treatment can affect the level of LINE1 RNA, which regulates Dux silencing in ESC shown in Ref. 14 (PMID: 29937225).
- 5) Fig1j, "0.4um" is confusing.
- 6) Fig. 6e. Panel "e" in Figure 6 is missing.

Reviewer #3 (Remarks to the Author):

In the present manuscript, Hua Yu and colleagues investigate the effect of nucleolar stress on the reprogramming of mouse embryonic stem cells (mES). In particular, the authors reported that treatment of mES cells with CX-5461, a potent inhibitor of rRNA synthesis, induces transcriptional changes consistent with the transition of mES cells into a 2C embryo-like (2C-like) cells. 2C-like cells share molecular features of the totipotent 2C-embryo stage. Using different approaches, the authors demonstrated that upon treatment with CX-5461, markers of the 2-cell stage are activated, including Dux and Gm12794. Consistent with these transcriptional changes, the authors observed an expansion in the 2C-like cell population in mES cells. Combining microscopy, ChIP-seq, and Hi-C, the authors linked nuclear stress to changes in 3D chromatin structure and reorganization of PNH, affecting the epigenome of mES and driving the transcriptional reprogramming toward a 2-cell like state. Based on their results, the authors thus concluded that rRNA biogenesis influences the 2C-like program by regulating nuclear phase separation and 3D structure at the PNH. For the most part, the data support the authors' conclusions and provide a novel mechanism contributing to the reprogramming of mES cells. This reviewer appreciated the excellent Hi-C analysis performed by the authors. However, this reviewer's opinion that the data presented not strongly supports Dux's role. Data presented in Figure 3c are not very convincing and supportive of Dux's role in binding the genes regulated by CX-5461 treatment. Since the authors showed in Figure 3g that depletion of Dux offsets the transcriptional changes induced by CX-5461, the authors should assess Dux binding and chromatin state of CX-5461 genes after Dux silencing. Does Dux silencing affect chromatin accessibility and deposition of H3K27me3 and H3K9me3 at deregulated genes?

In Extended Figure S2f, the ChIP-seq tracks do not decrease the indicated markers between control and CX-5461; however, ChIP-seq analysis is not quantitative unless "spick-ins" are added to normalized ChIP-seq results. Such limitation should be addressed or at least discussed

Reviewer #1 (Remarks to the Author):

The nucleolus is the organelle for ribosome biogenesis and for sensing various types of stress. In this manuscript, Yu et al. provided a novel perspective on its role in regulating stem cell fate. The authors present evidence that nucleolar stress induced by interfering rRNA biogenesis can drive a 2-cell stage embryo-like (2C-like) transcriptional program and induce an expanded 2C-like cell population in mouse embryonic stem (mES) cells. It is well-known that the liquid-liquid phase separation (LLPS) mediated by rRNA and nucleolar proteins maintains the nucleolar integrity and the formation of peri-nucleolar heterochromatin (PNH). The authors found that Pol-I inhibitor CX-5461 but not other cellular stress inducers treatments caused RNA biogenesis defect and disrupted LLPS of the nucleolus, causing dissociation of the NCL/TRIM28 complex from PNH and changes of epigenetic states and reorganization of the 3D structure of PNH. Consequently, Dux, a 2C program transcription factor gene, is released from the PNH region and activates the 2C-like program. The in vivo functional relevance of this regulatory axis was validated by demonstrating that embryos with rRNA biogenesis defect are incompatible to develop from 2-cell (2C) to 4-cell stage, with delayed repression of 2C/ERV genes and a transcriptome skewed toward earlier cleavage embryo signatures. The results in this study highlight that nucleolar integrity maintained by rRNA-mediated LLPS and 3D chromatin structure regulates the fate transition of mES cells to 2C-like cells, and that rRNA biogenesis is a critical regulator during the 2-cell-to-4-cell transition of murine preimplantation embryo development.

Overall, this is an interesting story with novel insights into our understanding of nucleolus functions in cell fate determination and early development. The findings should also provide new ways in reprogramming mES cells back to 2C-like cells by manipulating the ribosome biogenesis process through LLPS and 3D chromatin structure remodeling at the PNH. There do exist a few major and minor issues that should be addressed to further improve the quality of this manuscript.

Response: We thank the reviewer for this excellent comment that our findings presented are novel and interesting and appreciate the critics that a few key questions need to be addressed to make these findings more convincing.

Major critiques:

Comment 1: It is unclear how long CX-5461 treatment was used for the study. Detailed information such as dosage and time length of CX-5461 treatment for the RNA-seq samples have not been mentioned in this manuscript. Since 2CLCs are triggered by CX-

5461 in ESCs, what will happen if rRNA biogenesis inhibition is sustained by long-term Pol-I inhibition? Will a higher percentage of 2CLCs maintained by CX-5461? Will ESCs be apoptotic? differentiating? or enter a diapause status like mTOR inhibition?

Response: We thank the reviewer for this excellent and helpful suggestion. The treatment dosage of CX-5461 for RNA-seq was 2 μ M and the treatment time of CX-5461 for RNA-seq was 12h. We have added the detailed treatment dosages and times of three small molecule chemicals to our new revised manuscript (Page 4 Lines 98-100). Indeed, we have inhibited rRNA biogenesis by CX-5461 with different treatment dosages and times. We used three treatment time points including 12h, 24h and 36h, and five different treatment dosages including 0.4 μ M, 1.2 μ M, 2.0 μ M, 2.8 μ M and 3.6 μ M (Fig 1a in this letter). We observed that the mES cells are almost all apoptotic when treated with CX-5461 36h under 1.2 μ M, 2.0 μ M, 2.8 μ M and 3.6 μ M, and 24h under 2.0 μ M, 2.8 μ M and 3.6 μ M (Fig 1a in this letter). We thus selected other treatment dosages and timepoints to measure 2C markers (Mervi-Pol and Dux), apoptotic marker (Casp8), differentiation markers (Cdx2, Nanog and Gata6) and diapause-like status markers (Myc, Slc38a1 and Slc38a2)^{1,2} using qRT-PCR experiment.

*We have observed consistent and up-regulated expression of Casp8 from 12h to 36h under 0.4 μ M treatment dosage, from 12h to 24h under 1.2 μ M treatment dosage and from 0.4 μ M to 3.6 μ M under 12h treatment (Fig 1b and Fig 1c in this letter). For the differentiation and diapause markers, we have not observed the monotonous trend of increase or decrease of their expression from 12h to 36h under 0.4 μ M treatment dosage and from 0.4 μ M to 3.6 μ M under 12h treatment (Fig 1b and Fig 1c in this letter). However, we have observed obvious upregulation of differentiation marker “Gata6” and diapause-like markers (Fig 1b in this letter) from 12h to 24h under 1.2 μ M treatment dosage. **These results indicated that moderate dosages and times of CX-5461 will induce mES cells into 2C-like state, differentiation, or diapause-like status, yet long-term inhibition of rRNA biogenesis will lead to mES cell apoptosis.***

*As well shown in Fig 1b-1c in this letter, CX-5461 “2 μ M,12h” treatment induced highest 2C marker gene expression and appropriate expression levels of apoptosis, differentiation and diapause markers, we have therefore selected “2 μ M 12h” for subsequent investigation. To check whether the mES cells treated with “2 μ M,12h” CX-5461 have entered diapause-like status globally, we conducted hierarchical clustering of transcriptomes derived from our current study, mouse pre-implantation embryos³ and two previous diapause-like status studies^{1,2}. We found that the transcriptome of CX-5461 treated mES cells was more similar with 2-cell embryos than diapause-like mES cell and paused ICM embryo (Fig 1e in this letter). **Collectively, these results suggested that the mES cells treated with “2 μ M,12h” CX-5461 were mainly not differentiated or entered the diapause-like status but are transition into earlier 2-cell like cell.***

Fig 1. a) Different schemes of treatment with CX-5461; “Dead” denoted that mES cells are almost all apoptotic using the given treatment time and dosage of CX-5461. **b)** qRT-PCR showing the expression levels of 2C markers (*Mervl-pol* and *Dux*), differentiation markers (*Nanog*, *Cdx2* and *Gata6*), diapause state markers (*Myc*, *Slc38a1*, *Slc38a2*) and apoptosis marker (*Casp8*) of mES cells with 0.4 μM CX-5461 at different time points (12h, 24h and 36h); **c)** qRT-PCR showing the expression levels of 2C markers (*Mervl-pol* and *Dux*), differentiation markers (*Nanog*, *Cdx2* and *Gata6*), diapause state markers (*Myc*, *Slc38a1*, *Slc38a2*) and apoptosis marker (*Casp8*) of mES cells treated with 1.2 μM CX-5461 at different time points (12h and 24h); **d)** qRT-PCR showing the expression levels of 2C markers (*Mervl-pol* and *Dux*), differentiation markers (*Nanog*, *Cdx2* and *Gata6*), diapause state markers (*Myc*, *Slc38a1*, *Slc38a2*) and apoptosis marker (*Casp8*) of control mES cells and mES cells treated with 12h CX-5461 in different dosages (0.4 μM, 1.2 μM, 2.0 μM, 2.8 μM, 3.6 μM). **e)** Hierarchical clustering of transcriptomes from our study, mouse pre-implantation embryos³ and two previous diapause-like status studies^{1,2} (GEO accession number: GSE81285 and GSE143494).

Comment 2: Have the authors ever compared the rRNA transcription level between the sporadically 2C like cells vs the non-2C like mESCs? This could be an important piece of data to further substantiate the CX-5461 treated datasets.

Response: We thank the reviewer for this constructive suggestion. We have re-analyzed public RNA-seq dataset of sporadically 2C like cells and non-2C like mESCs downloaded from NCBI GEO database (GEO accession number: GSE33920, PMID: 22722858). Using this dataset, we have first checked the expression levels of different subunit genes of RNA polymerase I (Pol I). We did not observe obviously expressional difference of these genes between sporadically 2C-like cells and non-2C like mESCs (Fig 2a in this letter). We further sorted the sporadically 2C-like cells and non-2C like mESCs and performed qRT-PCR experiment. We observed only a tiny reduction of rRNA expression level in sporadically 2C-like cells when compared with non-2C like mESCs (Fig 2b in this letter). These obtained results might have indicated that the sporadically 2C-like cell population in the mES cell culture were not mainly produced by direct regulating rRNA biogenesis process and other potential regulators need to be further investigated. Interestingly, we have found obviously decreased KEGG RIBOSOME gene expression in both CX-5461 treated mES cells and sporadically 2C-like cells (Fig 2c and Fig 2d), indicating that low ribosomal gene expression was a common molecular feature between CX-5461 treated mES cells and sporadically 2C like cells. Collectively, these results suggested that regulating the ribosome biogenesis process is an effective method in reprogramming mES cells back to 2C-like cells, and our work provided a practicable method to achieve this reprogramming by direct manipulating rRNA biogenesis process.

Fig.2. a) The expression levels of different subunit genes of RNA polymerase I (Pol I) in published RNA-seq dataset (GEO accession number: GSE33920); 2C::tdTomato-: non-2C

like mESCs; 2C::tdTomato-: 2C like cells; TPM: Transcripts Per Kilobase Million. **b)** qRT-PCR quantification of rRNA expression in the non-2C like mESCs and sporadically 2C like cells; 2C::tdTomato-: non-2C like mESCs; 2C::tdTomato+: 2C like cells. **c)** GSEA analysis showing collective changes in the KEGG RIBOSOME gene set in the non-2C like mESCs and sporadically 2C like cells. **d)** GSEA analysis showing collective changes in the KEGG RIBOSOME gene set in the control mES cells and CX-5461 treated mES cells.

Comment 3: There are hundreds of snoRNAs within the mouse genome, which snoRNA has been knockout for the snoRNA knockout cell line? In addition, the knockout of snoRNA and degradation of Pol I need to be confirmed by qPCR and WB.

Response: We thank the reviewer for this constructive suggestion. For snoRNA KO mES cells, we have asked Dr. Pengxu Qian Lab for help. The knocked-out mouse snoRNAs were the homologs of human SNORD113-114 gene cluster. In the snoRNA KO mES cells, a band of 400bp can be detected by PCR experiment. Using the primer provided by Qian Lab, we performed PCR experiment for detecting 400bp band in control mES cells and snoRNA KO mES cells. As expected, we observed that a band with the length of 400bp, indicating that these snoRNA genes was successfully knocked-out (Fig 3a in this letter). Additionally, we have also performed Immunoblotting experiment in control mES cells and Pol I degraded mES cells. As well shown in Fig 3b in this letter, the Pol I protein was successfully degraded in Pol I degradation mES cells. We added these results to our new revised manuscript labeled as Fig S5a and Fig S5b.

Fig 3. a) PCR experiment showing that a 400bp band was observed in the snoRNA KO mES cells, but not in the wild-type (WT) mES cells. This result indicates that the homologs of human SNORD113-114 gene cluster was successfully knocked-out. **b)** Immunoblotting experiment showing Auxin-induced Pol I protein degradation after 24h of Auxin treatment.

Minor points:

Comment 4: Fig. 2d-e mentioned “mislocations”, what kind of mislocation should be specified. In addition, it seems obvious that reduced signal density as well besides the mislocation.

Response: We thank the reviewer for this good and helpful suggestion. Indeed, we have not observed the obvious change of protein abundances for RPA194 and FBL after CX-5461 treatment using Immunoblotting experiment (Fig 4 and Fig 5a in this letter). This result indicated that signal density of FBL and RPA194 is not decreased but the two proteins are

abnormally distributed with aggregated pattern. Therefore, we have revised our claim in our previous version of manuscript to “Immunofluorescence of FBL and RPA194 protein also showed abnormal distribution and morphology with aggregated pattern in the nucleolus upon treatment” in our new revised manuscript (Page 6 Lines 161-162).

Fig 4. a) Immunoblotting showing the expression level of RPA194 protein in control and CX-5461 treated mES cells.

Page 6 Lines 161-162 of our new revised manuscript

“Immunofluorescence of FBL and RPA194 protein also showed abnormal distribution and morphology with aggregated pattern in the nucleolus upon treatment”

Comment 5: In Fig.2f-h, since the authors used Lenti-NCL-mCherry/Lenti-NPM1-mCherry/ Lenti-FBL-mCherry for the FRAP experiment, it is better to keep the pseudo color consistent with the fluorescence protein (mCherry). And the overexpression level between the control and CX5461-FBL is not comparable in fig.2h.

Response: We thank the reviewer for this helpful suggestion. Indeed, we used Lenti-NCL-GFP/Lenti-NPM1-GFP/Lenti-FBL-mCherry for performing FRAP experiment. We are sorry for the writing error in “Methods” section of our previous version of manuscript. We have corrected this error in our new revised manuscript. According to the reviewer’s suggestion, we further compared the overexpression level by Immunoblotting experiment in control and CX-5461 treated mES cells. We observed moderate overexpression in control and CX-5461 treated mES cells for FBL, NCL and NPM1 (Fig 5a-5c in this letter). Though a slight increase of overexpression was observed for NPM1 after CX-5461 treatment, we did not observe obvious change of overexpression level for FBL and NCL between control and CX-5461 treated mES cells (Fig 5a-5c in this letter).

Fig 5. a) Immunoblotting showing the expression level of FBL and FBL-mCherry in E14 (WT), E14 expressing FBL-mCherry and CX-5461 treated E14 expressing FBL-mCherry cells; **b)** Immunoblotting showing the expression level of NPM and NPM-eGFP in E14 (WT), E14 expressing NPM-eGFP and CX-5461 treated E14 expressing NPM-eGFP cells; **c)** Immunoblotting showing the expression level of NCL and NCL-mCherry in E14 (WT), E14 expressing NCL-eGFP and CX-5461 treated E14 expressing NCL-eGFP cells.

Comment 6: The references for Inactive Hub, NAD, L1 in figure 4 should be mentioned in the main text.

Response: We thank the reviewer for this helpful and constructive suggestion. According to the reviewer's suggestion, we added the corresponding references of Inactive Hub, NAD and L1 to our new revised manuscript (please see Page 7 Lines 175-178 of our new revised manuscript).

Page 7 Lines 175-178 of our new revised manuscript

"Moreover, we observed increased H3K4me3 and H3K27ac levels and improved chromatin accessibility at Inactive Hub ⁴⁴, NAD ^{45,47} and L1 regions (downloaded from UCSC Table Browser) in CX-5461 treated cells".

Comment 7: Fig6c, one mislabeling for a bar chart, it should be "Pre-rRNA". Fig. 6d, the color-coding is hard to distinguish different development stage. Fig.6e panel "e" is missing.

Response: We thank the reviewer for this helpful suggestion. According to the reviewer's suggestion, we corrected the mislabeling error in Fig 6c in our new revised figure. We also revised the color-coding of Fig 6d to better distinguish different development stages in our new revised figure. We added the panel "e" to Fig 6e in our new revised figure.

Comment 8: CX-5461 or CX5461 consistency issue.

Response: We thank the reviewer for this helpful suggestion. According to the reviewer's suggestion, we corrected the inconsistency issue of CX-5461 and CX5461 and unified to CX-5461 in our new revised manuscript.

Comment 9: page 9 line 11, (Figure.5m) is actually (Fig.4m).

Response: We thank the reviewer for this helpful and constructive suggestion. According to the reviewer's suggestion, we have revised this label error in our new revised manuscript.

Comment 10: It is better if the author can clearly specify which datasets are their own or from public resources. i.e., provide ref# or GES# in the figure legend, or provide a Suppl Table for all data they have used.

Response: We thank the reviewer for this constructive and helpful suggestion. According

to the reviewer's suggestion, in our new revised manuscript, we have described which datasets were produced by our current study or were obtained from public data resources by providing the NCBI GEO accession number in the figure legend. In addition, we also provided a Supplementary Table (Supplementary Table 4) for all datasets used in our study.

Comment 11: The authors did a Hi-C-seq analysis in ESCs with Pol-I inhibitor treatment. It will be interesting by more global analysis from their HiC-seq data, but not just focusing on the inductive hub, NAD, and L1 regions.

Response: We thank the reviewer for this constructive and helpful suggestion. We have performed a more global analysis and compared control mES cells and CX-5461-treated mES cells Hi-C maps, with lymphoblastoid cells as a reference for full differentiated cells. A global analysis of A(active)/B(inactive) compartment strength showed a slight decrease of Hi-C contacts within the B compartments in CX-5461-treated mES cells compared with control mES cells (Fig 6a-6b in this letter and Extended Data Fig.S4e-S4h in our new revised manuscript). However, at topologically associating domain (TAD) or chromatin loop level, we found a mild increase in their strength in CX-5461-treated mES cells (Fig 6c-6f in this letter and Extended Data Fig.S4i-S4m in our new revised manuscript). In addition, we also compared the difference of 3D interaction and structural correlation between Inactive Hub, NAD and L1 regions and randomly selected genomic regions. When compared with control mES cells, we observed that Inactive Hub, NAD and L1 regions have significantly more decreased 3D chromatin interaction and structural correlation than randomly selected genomic regions in CX-5461 treated mES cells (please see Pages 9-10 Lines 249-266 of our new revised manuscript).

Fig 6. a) A/B interaction profile showing contact enrichment between active and inactive compartments. **b)** Quantification of compartment strength; *: $p < 0.05$, ***: $p < 0.001$, Wilcox signed rank test. **c)** Observed/Expected (O/E) aggregate plot of TADs. **d)** Quantification of TAD strength; *: $p < 0.05$, ***: $p < 0.001$, Wilcox signed rank test. **e)** O/E aggregate plots of chromatin loops. **f)** Quantification of loop strength; ***: $p < 0.001$, Wilcox signed rank test.

Pages 9-10 Lines 249-266 of our new revised manuscript

“We observed markedly decreased higher-order chromatin interactions within PNH region indicated by the Inactive Hub, NAD and L1 regions in the treated cells (Fig.4a-4c, compared with randomly selected genomic regions, Mann-Whitney U test, the replicates of experiment $n=10$, averaged p -values=0, 0, $3.42E-06$ for Inactive Hub, NAD and L1, respectively). Moreover, the Dux locus is significantly further away from the PNH region as characterized by the largely decreased Hi-C contacts (Fig.4a-4c and Fig.4g-4i, compared with randomly selected genomic regions, Mann-Whitney U test, the replicates of experiment $n=10$, averaged p -values= $1.65E-05$, $7.82E-05$, $2.68E-03$ for Inactive Hub, NAD and L1, respectively). We further analyzed the 3D chromatin structural correlation within PNH region, and between the Dux locus and PNH region by comparing Hi-C pearson correlation coefficient (PCC) matrix of control and CX-5461-treated mES cells. We observed the obviously decreased 3D chromatin structural correlation within PNH region (compared with randomly selected genomic regions, Mann-Whitney U test, the replicates of experiment $n=10$, averaged p -values=0 for all Inactive Hub, NAD and L1) and between the Dux locus and PNH region (compared with randomly selected genomic regions, Mann-

Whitney U test, the replicates of experiment n=10, p-values=1.86E-6, 5.03E-6, 4.02E-5 for Inactive Hub, NAD and L1, respectively) (Fig.4d-4f and Fig.4j-4l)."

Comment 12: Pol I protein (PRA1): PRA1 does not sound right. Please double check the accuracy.

Response: We thank the reviewer for this helpful suggestion. According to the reviewer's suggestion, we corrected PRA1 to RPA1 in our new revised manuscript.

Reviewer #2 (Remarks to the Author):

The nucleolus is a nuclear body for ribosome biogenesis, which consists of multiphase liquid droplets. Yu et al. reported that nucleolar stress induced by interfering rRNA biogenesis could drive 2C-like transcriptional program and increased 2C-like cell population in mES cells. The process was accompanied by not only the structural changes of nucleoli but also the reorganization of the 3D chromatin structure: a 2C program transcription factor gene *Dux* was dissociated from peri-nucleolar heterochromatin regions and became active. Yu et al. claimed that rRNA biogenesis and matured nucleoli are indispensable for the 2-cell to 4-cell transition in mouse embryos. The authors also claimed that the structural integrity of nucleoli is maintained by rRNA mediated-liquid-liquid phase separation (LLPS) and plays a role in transcriptionally repressing 2C program for the transition from the 2-cell to the 4-cell stage embryos when zygotic genes are activated. Since the LLPS is a hot topic in cell biology, the issues in this manuscript are timely and potentially intriguing in the related fields. However, the novelty of the main message is not clear to me. In addition, there are several concerns to be addressed. My specific comments are as follows:

Response: We thank the reviewer for his good and positive comment on our novel findings and appreciate the critics that a few key questions need to be addressed to make these novel findings more convincing. To be more clearly demonstrate the novelty of our current work, we highlighted three aspects of the key findings of our story as follow:

First, the most important novelty of our work is that we found that rRNA biogenesis defect triggers 3D structure reorganization of peri-nucleolar heterochromatin (PNH), which leads to *Dux*, a 2C program transcription factor gene, to be released from the PNH region and activation of the 2C-like program. In well support with this finding, we observed that 3D structure of PNH reorganizes after early 2-cell during mouse pre-implantation embryo development, coinciding with the rRNA biogenesis and *Dux* repression.

Second, we reported that nucleolar stress induced by interfering rRNA biogenesis can drive 2-cell stage embryo-like (2C-like) transcriptional program and induce an expanded 2C-like cell population in mES cells. Nucleolus, the largest membrane-less condensate in a cell, is a stress-sensitive organelle and ensure quality control

of nuclear proteome under stress ⁴⁻⁸. *Previous studies on nucleolus in stem cells mainly focused on the role of rRNA and its associated chromatin in the context of ES differentiation* ^{9,10}. *In contrast, we revealed its role in reprogramming ES cells back to 2 cell-like cells.*

Third, by in vivo functional experiment, we demonstrated that rRNA biogenesis is a critical regulator during 2-cell-to-4-cell transition of murine pre-implantation embryo development.

In summary, our work highlighted that nucleolar integrity maintained by rRNA biogenesis and 3D chromatin structure reshaping of PNH compartment regulates the fate transition between mES cells and 2C-like cells. Given the dynamic regulation of nucleolus and its associated chromatin ¹¹⁻¹⁴ in cell fate transitions and environmental perturbations, it is conceivable that the mechanisms elucidated in our study might be applied to other contexts of physiological and pathological processes.

Major comments:

Comment 1: It is unclear what is new on how the nucleolar LLPS involves the establishment and the maintenance of peri-nucleolar heterochromatin including Dux loci. Although the authors claimed that the nucleolin is a key player, it was already shown that it represses the Dux/2C program in Ref. 14 (PMID: 29937225).

Response: *We thank the reviewer for this good and helpful critic. In this work, we do not intend to overstate that the fate transition of mES cell to 2C-like cell triggered by rRNA biogenesis defect are fully explained by nucleolar LLPS ¹⁵. What we observed is that nucleolar integrity mediated by rRNA biogenesis maintains the normal LLPS and 3D structure of peri-nucleolar heterochromatin. As shown in the main title of our previous version of manuscript "rRNA Biogenesis Regulates Mouse 2C-like State by 3D Structure Reorganization of Peri-Nucleolar Heterochromatin", our key hypothesis is that nucleolar integrity defect triggered by inhibiting rRNA biogenesis reshaped 3D chromatin structure of PNH compartment, which regulates the fate transition of mES cells to 2C-like cells. We have carefully revised our previous version of manuscript to correct our statements for supporting this hypothesis.*

In our previous study, we have found that the nucleolus-localized LIN28 protein promotes rRNA biogenesis and represses Dux expression and 2C-like transcriptional program (Fig 7 in this letter). Our LIN28 work have recently been published on Protein & Cell and is accessible by <https://doi.org/10.1007/s13238-021-00864-5>. To further confirm whether rRNA biogenesis regulates mouse 2C-like state by 3D structure reorganization of PNH, we further performed Hi-C experiment in wild-type and Lin28a^{-/-} mES cells. As expected, we observed the similar 3D chromatin structure reorganization of Lin28a^{-/-} mES cells with CX-5461 treated mES cells (Fig 17 in this letter).

Fig 7. a) qRT-PCR showing the expression level of rRNA in wild-type mES cells, Lin28a knocked-out mES cells and Lin28a knocked-out mES cells with Lin28a overexpression; **b)** UCSC Genome Browser view showing RNA-seq results at Dux gene in wild-type and Lin28a knocked-out ES cells; **c)** qRT-PCR showing the expression of Mervl-polymerase gene in wild-type mES cells, Lin28a knocked-out mES cells and Lin28a knocked-out mES cells with Lin28a overexpression; **d)** Scatter plot of RNA-sequencing data comparing gene expression of wild-type and Lin28a knocked-out mES cells.

We were agreed with the reviewer comment that NCL was already shown that it represses the Dux/2C program in Ref. 14 (PMID: 29937225). However, the authors of Ref. 14 have only demonstrated that NCL binds to the Dux locus but not the PNH compartment globally. In Fig 8 of this letter, we have highlighted the main novelty of our current study compared with Ref 14 (PMID: 29937225). The novel concept of “Nuclear compartmentalization as a mechanism of quantitative control of gene expression” has been discussed in a recent review at Nature Reviews Molecular and Cell Biology ¹⁶. Our study showcased a good example of 2C gene expression regulation by the nuclear compartmentalization around the peri-nucleolar heterochromatin.

Fig 8. Graphical abstracts of our current study and Ref.14. In this figure, we highlighted four main novelties of our work compared with Ref.14. Firstly, we reported that inhibiting rRNA biogenesis reshapes the 3D structure of PNH and MERVL regions. However, it has not been mentioned in Ref.14. Secondly, we found that inhibiting rRNA biogenesis causes the dissociation of NCL/TRIM28 complex from PNH. Yet, Ref.14 reported only that LINE1 KD leads to reduced binding of NCL/TRIM28 complex on Dux loci. Thirdly, we observed that inhibiting rRNA biogenesis disrupted normal nucleolar phase separation, which has not been reported in Ref.14. Finally, we found that rRNA biogenesis is critically required at the 2-cell-to-4-cell stage transition of mouse embryo. However, Ref.14 only observed that rRNA biogenesis is required for mouse pre-implantation embryo development.

In this study, we have observed that inhibiting of rRNA biogenesis disrupts normal nucleolar liquid-liquid phase separation (LLPS) and leads to dissociation of NCL/TRIM28 complex from PNH globally. These results might have indicated that LLPS is important for the assembly and function of nucleolus with the implication of gene regulation¹⁷ and is also well in line with the emerging notion that phase-separated condensates regulate gene transcription, epigenetics, and higher-order chromatin structure in various biological contexts¹⁷⁻²⁰. It is worth being mentioned that 3D structure reshaping of PNH mediated by nucleolar LLPS is possibly a common thread of RNA and protein-mediated Dux silencing and 2C program repression (e.g. through rRNA, snoRNA, LINE1 RNA, NCL, TRIM28 and LIN28). In our new revised version of manuscript, we have pointed out that the NCL/TRIM28 complex and other proteins localized at the nucleolus and its peripheral heterochromatin, e.g. NPM1 and HP1, are possibly the key regulating factors coordinately linking nucleolar LLPS and the establishment and maintenance of PNH.

Comment 2: Fig.1a-d. Rapamycin inhibits ribosome biogenesis, including rRNA

transcription and processing (e.g., PMID: 15004009; PMID: 14612424). However, the rapamycin treatment did not seem to induce 2C-like cells. The authors should clarify this point.

Response: We thank the reviewer for this good critic. According to the reviewer's comment, we further surveyed the previous literature (PMID: 15004009; PMID: 14612424), we found that these two published studies were performed in the NIH 3T3 cells and HEK293T cells with a concentration of 20 or 80 nM rapamycin for treatment time up to 24h. In our study, we used 2.0 μ M rapamycin for treating mES cells 12h. We performed qRT-PCR experiment in control mES cells and rapamycin-treated mES cells to measure the expression level of rRNA. We have not observed obvious expression change of rRNA level between control mES cells and rapamycin-treated mES cells (Fig 9a in this letter). In addition, we have also not observed the obvious expression change of ribosome biogenesis genes by GSEA enrichment analysis of RNA-seq datasets of control mES cells and rapamycin-treated mES cells (Fig 9b in this letter). Collectively, these results indicated that different cell lines have different tolerances for rRNA biogenesis inhibition, and rapamycin treatment did not effectively inhibit ribosome biogenesis in mES cells with a 2.0 μ M rapamycin for treating mES cell 12h.

Fig 9. a) qRT-PCR quantification of rRNA expression in control mES cells and Rapamycin-treated mES cells. **b)** GSEA analysis showing the collective changes of KEGG RIBOSOME gene set in control mES cells and Rapamycin-treated mES cells.

Comment 3: "the percentage of tdTomato positive cells was maintained even at 24 hours after CX-5461 withdrawal (Fig.1k)." I do not agree because the density plots after CX-5461 removal in Fig. 1k look different from that with just CX-5461 treatment (Fig. 1j).

Response: We thank the reviewer for this good critic. We have repeated this experiment and found that the percentage of tdTomato positive cells was improved to about 2-fold and 8-fold even at 24 hours after 0.4 μ M and 2.0 μ M CX-5461 withdrawal, respectively (Fig 10a in this letter and Fig 1k in our new revised manuscript). There is only a little decrease when compared with before 2.0 μ M CX-5461 withdrawal (Fig 10b in this letter and Fig 1j in our new revised manuscript). Therefore, we have revised our claim in our previous version of manuscript to "the percentage of tdTomato positive cells was largely maintained even at 24 hours after CX-5461 withdrawal" in our new revised manuscript.

Fig 10. a) FACS analysis on 2C::tdTomato+ cells after 12h treatment and 24h withdrawal of CX-5461, demonstrating the change of percentage of 2C-like cells; **b)** FACS analysis on 2C::tdTomato+ cells upon different treatment doses of CX-5461, demonstrating the change of percentage of 2C-like cells.

Comment 4: Fig.2b-e. I found that DAPI signals are enriched around peri-nucleolar region after the CX-5461 treatment. This seems to be inconsistent with a feature of 2-cell like cells: a loss of chromocenters as shown in Ref. 14 (PMID: 29937225). I am not convinced that CX-5461 treatment really disassembled peri-nucleolar heterochromatin and subsequently induced the subsequent 2-cell like cells.

Response: We thank the reviewer for this good critic. We further analyzed the features of CX-5461-induced 2C::tdTomato+ cells. We observed that these cells have the expected features of 2C-like cells: loss of chromocenters and lack of Oct4 protein, which is well consistent with the previous published works^{21,22} (Fig 11-13 in this letter). We agree with the reviewer's comment "DAPI signals are enriched around peri-nucleolar region after the CX-5461 treatment in Fig.2b-e". A possible explanation for this is cellular heterogeneity in responding to CX-5461 treatment as only <5% mES cells convert to 2C-like cells after CX-5461 treatment. In addition, the phenomenon of cellular heterogeneity has been also widely observed in the studies of 2C-like cells²²⁻²⁴. We have also realized the word "disassembly" in our previous version of manuscript is unsuitably used as we only found changed epigenetic state and 3D chromatin structure of PNH after CX-5461 treatment. To be rigorous, we have revised the unsuitable claim "Deficiency of rRNA biogenesis disrupted nucleolar LLPS and promoted the disassembly of PNH region" of our previous version of manuscript to "Deficiency of rRNA biogenesis disrupted normal state of nucleolar LLPS and epigenetic state of PNH region" in our new revised manuscript (Page 7 Lines 187-188).

Fig 11. a) Immunofluorescence analysis of sporadically 2C-like cells in control mES cells. **b)** Immunofluorescence analysis of CX-5461-induced 2C-like cells. **c)** The percentage of 2C::tdTomato positive cells was quantified using FACS analysis in control mES cells and CX-5461-treated mES cells; Data are means \pm SD; SD: Standard Deviation; **: $p < 0.01$, ***: $p < 0.001$, two-way ANOVA, the replicates of experiment $n = 5$ (Extended data Fig S1I in our new revised manuscript). **d)** Graph showing the percentage of 2C::tdTomato+ cells that have the expected 2C-like features (loss of chromocenters and Oct4 protein). Scale bar, 10 μ m; n, number of cells.

Fig 12. Immunofluorescence analysis of LINE1 KD-induced 2C-like cells. Graph depicts the percentage of GFP+ cells that have the expected features (loss of chromocenters and Oct4 protein). Scale bar, 10 μ m; n, number of cells. This figure and legend were obtained from (Percharde et al. Cell, 2018).

Fig 13. Immunofluorescence quantifying the loss of pluripotency (i.e., the presence (+) or absence (-) of POU5F1 protein) and chromocenters (Chromo) in mESCs after ectopic Dux expression, n = 110 cells. Scale bar, 10 μ m. This figure and legend were obtained from (Hendrickson et al. Nature Genetics, 2017).

Page 7 Lines 187-190 of our new revised manuscript

“Altogether, these results demonstrated the rRNA biogenesis defect affected the normal state of nucleolar phase separation and changed the epigenetic state of the heterochromatic regions at the periphery of nucleolus by breaking up the binding of NCL/TRIM28 complex on the PNH region.”

Comment 5: “Altogether, these results demonstrated the rRNA biogenesis defect led to impaired nucleolar phase separation”. What did “impaired phase separation” mean? Even after CX-5461 treatment, some parts of the nucleolar structure remained and still phase-

separated in Fig. 2a-c. The nucleolar structure after Pol I inhibition or depletion was well characterized in PMID: 33055158.

Response: *We thank the reviewer for this good and helpful critic. We agree with the reviewer's comment that some parts of the nucleolar structure remained and still phase-separated even CX-5461 treatment. In our previous version of manuscript, the "impaired phase separation" is an improper statement. We are sorry for this improper claim, and we have revised this claim to "rRNA biogenesis defect affected the normal state of nucleolar phase separation" in our new revised manuscript (Page 7 Lines 187-188).*

Page 7 Lines 187-190 of our new revised manuscript

"Altogether, these results demonstrated the rRNA biogenesis defect affected the normal state of nucleolar phase separation and changed the epigenetic state of the heterochromatic regions at the periphery of nucleolus by breaking up the binding of NCL/TRIM28 complex on the PNH region."

Comment 6: Extended Data Fig. S2f. I found only tiny differences between the ChIP-seq signals with and without CX-5461 treatment. Fig. 2i-2p, particularly on the NCL and TRIM28 signals, are not convincing.

Response: *We thank the reviewer for this good critic. For epigenetic markers of H3K9me3, H3K27me3, H3K4me3 and H3K27ac, we have selected other regions within PNH to further confirm the observed changes after CX-5461 treatment (Fig 14a-14c in this letter, and Fig S2b and Fig S2d in our new revised manuscript). As also well shown with box plots and rigorous statistical analysis in Fig 2i-2n and Fig S2a-S2d in our new revised manuscript (Fig 15 in this letter), the H3K9me3 and H3K27me3 levels of PNH were remarkably decreased and the H3K4me3 and H3K27me3 levels of PNH were significantly increased after CX-5461 treatment. **For NCL and TRIM28, we have further repeated ChIP-seq experiment. Compared with the results of ChIP-seq experiment in our previous version of manuscript, we have observed more obvious reduction of NCL and TRIM28 binding on PNH (Fig 16 in this letter, Fig 2o-2q, Fig S2e and S2f in our new revised manuscript).** These results fully demonstrated that rRNA biogenesis defect have changed the epigenetic state of the heterochromatic regions at the periphery of nucleolus by influencing the binding of NCL/TRIM28 complex on the PNH.*

Fig 14. a) UCSC Genome Browser viewing of H3K9me3, H3K27me3, H3K4me3 and H3K27ac ChIP-seq signals in control mES cells and CX-5461-treated mES cells around a representative PNH fragment at chr18:7,718,063-8,074,255 (Extended data Fig S2f in our previous version of manuscript); **b)** UCSC Genome Browser viewing of H3K9me3 and H3K27me3 ChIP-seq signals in control mES cells and CX-5461 treated mES cells around a representative PNH fragment at chr15:4,699,343-5,778,242; **c)** UCSC Genome Browser viewing of H3K4me3 and H3K27ac ChIP-seq signals in control mES cells and CX-5461 treated mES cells around a representative PNH fragment at chr10:3,632,504-3,926,058.

Fig 15. a) Heatmap plots demonstrate the levels of H3K9me3 and H3K27me3 on within 1mb region around start and end sites of Inactive Hub. The regions of different lengths of Inactive Hub fragments were fitted to 1mb. **b)** Heatmap plots demonstrate the levels of H3K9me3 and H3K27me3 on within 1mb region around start and end sites of NAD. The regions of different lengths of NAD fragments were fitted to 1mb. **c)** Heatmap plots

demonstrate the levels of H3K9me3 and H3K27me3 on within 1kb region around start and end sites of L1. The regions of different lengths of L1 sequences were fitted to 1kb. **d)** Heatmap plots demonstrate the level of H3K4me3 and H3K27ac on within 1mb region around start and end sites of Inactive Hub. The regions of different lengths of Inactive Hub fragments were fitted to 1mb. **e)** Heatmap plots demonstrate the levels of H3K4me3 and H3K27ac on within 1mb region around start and end sites of NAD. The regions of different lengths of NAD fragments were fitted to 1mb. **f)** Heatmap plots demonstrate the level of H3K4me3 and H3K27ac on within 1kb region around start and end sites of L1. The regions of different lengths of L1 sequences were fitted to 1kb. **g)** Boxplots demonstrate the averaged H3K9me3 and H3K27me3 levels of 101 Inactive Hub fragments, 578 NAD fragments and 34888 L1 sequences; ***: $p < 0.001$, Wilcoxon signed rank test. **h)** Boxplots demonstrate the averaged H3K4me3 and H3K27ac levels of 101 Inactive Hub fragments, 578 NAD fragments and 34888 L1 sequences; ***: $p < 0.001$, Wilcoxon signed rank test.

Fig 16. a) UCSC Genome Browser viewing of NCL and TRIM28 ChIP-seq signals in control mES cells and CX-5461 treated mES cells around a representative PNH fragment at chr18:7,718,063-8,074,255 (Extended data Fig S2f in our previous version of manuscript); **b)** UCSC Genome Browser viewing of NCL and TRIM28 ChIP-seq signals in control mES cells and CX-5461 treated mES cells around a representative PNH fragment at chr18:7,718,063-8,074,255 (Extended data Fig S2f in our new revised manuscript). **c)** Heatmap plots demonstrate the binding signals of NCL and TRIM28 on within 1mb region around start and end sites of Inactive Hub and NAD, and within 1kb region around start and end sites of L1. The regions of different lengths of Inactive Hub and NAD fragments

were fitted to 1mb. The regions of different lengths of L1 sequences were fitted to 1kb (Extended data Fig S2e in our previous version of manuscript); **d**) Heatmap plots demonstrate the binding signals of NCL and TRIM28 on within 1mb region around start and end sites of Inactive Hub and NAD, and within 1kb region around start and end sites of L1. The regions of different lengths of Inactive Hub and NAD fragments were fitted to 1mb. The regions of different lengths of L1 sequences were fitted to 1kb (Extended data Fig 2o-2q in our new revised manuscript); **e**) Boxplots demonstrate the averaged binding signals of NCL and TRIM28 on 101 Inactive Hub fragments, 578 NAD fragments and 34888 L1 sequences; **: $p < 0.01$, ***: $p < 0.001$, Wilcox signed rank test (Fig 2o and 2p in our previous version of manuscript); **f**) Boxplots demonstrate the averaged binding signals of NCL and TRIM28 on 101 Inactive Hub fragments, 578 NAD fragments and 34888 L1 sequences; **: $p < 0.01$, ***: $p < 0.001$, Wilcox signed rank test (Fig S2e in our new revised manuscript).

Comment 7: Fig. 4. It was reported that CX-5461 is a G-quadruplex stabilizer in the human genome (PMID: 28211448). I wonder whether the authors can exclude the possibility that CX-5461 treatment drives 3D chromatin reorganization independently of the nucleolar stress.

Response: We thank the reviewer for this good critic. To exclude the potential possibility that rRNA biogenesis defect (CX-5461 treatment) drives 3D chromatin reorganization independently of nucleolar stress, we performed further experiments of Hi-C using wild-type and *Lin28a*^{-/-} mES cells. The rationality of performing this experiment is that: 1) LIN28A is a classic RNA binding protein and has not been reported directly regulating 3D chromatin structure; 2) LIN28A localized within nucleolus and promoted rRNA biogenesis; 3) *Lin28a* deficiency triggered nucleolar stress, activated 2C-like program, and promote the transition of mES cells to 2C-like cells. The above findings have recently been published at *Protein & Cell* and is accessible by <https://doi.org/10.1007/s13238-021-00864-5>. Thus, *Lin28a*^{-/-} mES cells can be used as a way of validation for nucleolar stress-induced 3D chromatin reorganization, and a way to exclude the possibility of CX-5461-induced other effects such as stabilizing G-quadruplex. As expected, in *Lin28a*^{-/-} mES cells, we observed similar 3D structure reorganization with CX-5461 treated mES cells (Fig 17 in this letter), namely, *Lin28a* knockout cells have reduced interaction within PNH, reduced interaction between PNH and the *Dux* locus and more obvious topological associated domain (TAD) structure around *MERV1* gene loci. These obtained results further confirmed that CX-5461 treatment drives 3D chromatin reorganization is dependent on the nucleolar stress.

Fig 17. a) Observed/Expected (O/E) Hi-C contact maps of Inactive Hub (organized around the nucleolus) and 1.5 Mb genomic regions around *Dux* at 150-kb resolution; **b)** Hi-C Pearson correlation heat maps of Inactive Hub (organized around the nucleolus) and 1.5Mb genomic regions around *Dux* at 150-kb resolution; **c)** Scatter plot demonstrates the $\log_2(\text{fold change})$ of Hi-C O/E contacts between Inactive Hub and different types of genes in *Lin28a^{-/-}* and wild-type mES cells; **d)** Scatter plot demonstrates the Pearson correlation change between Inactive Hub and different types of genes in *Lin28a^{-/-}* and wild-type mES cells; **e)** O/E Hi-C contact maps of NAD regions and 1.5 Mb genomic regions around *Dux* at 150-kb resolution; **f)** Hi-C Pearson correlation heat maps of NAD regions and 1.5Mb genomic regions around *Dux* at 150-kb resolution; **g)** Scatter plot demonstrates the $\log_2(\text{fold change})$ of Hi-C O/E contacts between NAD regions and different types of genes in *Lin28a^{-/-}* and wild-type mES cells; **h)** Scatter plot demonstrates the Pearson correlation change between NAD regions and different types of genes in *Lin28a^{-/-}* and wild-type mES cells; **i)** O/E Hi-C contact maps of L1 regions and 1.5 Mb genomic regions around *Dux* at 150-kb resolution; **j)** Hi-C Pearson correlation heat maps of L1 regions and 1.5Mb genomic regions around *Dux* at 150-kb resolution; **k)** Scatter plot demonstrates the $\log_2(\text{fold change})$ of Hi-C O/E contacts between L1 regions and different types of genes in *Lin28a^{-/-}* and wild-type mES cells; **l)** Scatter plot demonstrates the Pearson correlation change between L1 regions and different types of genes in *Lin28a^{-/-}* and wild-type mES cells; **MERVL-int:** Up-regulated MERVL-int genes, **MT2_Mm:** Up-regulated MT2_Mm genes, **UG:** Up-regulated GENCODE genes, **DG:** Down-regulated GENCODE genes. **m)** Aggregate O/E Hi-C matrices centred on activated MERVL genes in wild-type and *Lin28a^{-/-}* mES cells at 40-kb resolution; **n)** Aggregate O/E Hi-C matrix at *Lin28a^{-/-}* activated MERVL genes throughout mouse early embryonic development. For illustrating the change of insulation around MERVL genes, the \log_2 transformed Hi-C O/E matrix was further scaled by z-score

normalization across each row.

Comment 8: Fig. 5k. Pol I degradation increased the percentage of 2C::tdTomato positive cells to about 20%, while CX-5461 treatment increased it to just 2~4 % in Fig. 2l. I wonder why it is so different between them.

Response: *We thank the reviewer for this good critic. We have repeated this experiment with 5 biological replicates in control mES cells and Pol I degraded mES cells. As expected, we observed that the percentage of 2C::tdTomato positive cells is increased from ~0.5% to ~3.5% (Fig 18 in this letter, Fig 5k and Fig S5e in our new revised manuscript). We have corrected this problem in our new revised manuscript.*

Fig 18. a) The percentage of 2C::tdTomato positive cells, which was quantified using FACS analysis in control mES cells and Pol I degraded mES cells; Data are means \pm SD, SD: Standard Deviation, ***: $p < 0.001$, two-way ANOVA, the replicates of experiment $n=5$; **b)** Representative image showing FACS analysis on 2C::tdTomato+ mES cells in control mES cells and Pol I degraded mES cells.

Minor comments:

Comment 9: “we generated two rRNA biogenesis-inhibited mES cell lines: 1) a line with degraded Pol I protein (PRA1)”. “PRA1” is a typo; it should be “RPA1”.

Response: *We thank the reviewer for this helpful suggestion. According to the reviewer's suggestion, we have corrected this typo in our new revised manuscript.*

Comment 10: Fig. 5b. I found that the nucleolar shape after Pol I degradation does not look similar to ones treated with CX-5461 shown in Figs. 2b and 4m. Specifically, in Fig. 5b, the nucleolus after Pol I degradation does not show round shape, contrasting with the ones after CX5461 treatment. Could the authors clarify this point?

Response: *We thank the reviewer for this good critic. According to the reviewer's comment, we have further repeated the DNA FISH and Immunofluorescence experiments in both Pol I degraded mES cells and snoRNA KO mES cells. We have observed that, in a part of Pol I degraded mES cells and snoRNA KO mES cells, the nucleolus does indeed show round*

shape, similar with the ones after CX5461 treatment (Fig 19a-c in this letter). For 10 observed mES cells, the proportion of NCL protein showing round shape for CX-treated mES cells, Pol I degraded mES cells and snoRNA KO mES cells are around 99%, 70% and 40%, respectively. A possible reason for explaining this is that the effects of Pol I degradation and snoRNA KO on the nucleolus are milder than the “2.0 μ M, 12h” CX-5461 treatment as indicated by increased percentage of 2C::tdTomato positive cells (CX-5461 treatment: from ~0.5% to ~4.0% (Fig 19d in this letter and Fig S1I in our new revised manuscript); Pol I degradation: from ~0.5% to ~3.5% (Fig 18a in this letter and Fig 5k in our new revised manuscript); snoRNA KO: from ~0.6% to ~2.4% (Fig 19e in this letter and Fig 5I in our new revised manuscript)).

Fig 19. a) DNA FISH analysis with a Dux locus probe and Inactive Hub locus probe, and co-immunostained with NCL protein in control mES cells and CX-5461 treated mES cells; **b)** DNA FISH analysis with a Dux locus probe and Inactive Hub locus probe, and co-immunostained with NCL protein in control and Pol I degraded mES cells; **c)** DNA FISH analysis with a Dux locus probe and Inactive Hub locus probe, and co-immunostained with NCL protein in control and snoRNA KO mES cells; **d)** The percentage of 2C::tdTomato positive cells was quantified using FACS analysis in control mES cells and CX-5461-treated mES cells; Data are means \pm SD; SD: Standard Deviation; **: $p < 0.01$, ***: $p < 0.001$, two-way ANOVA, the replicates of experiment $n = 5$; **e)** The percentage of 2C::tdTomato positive cells was quantified using FACS analysis in control mES cells and snoRNA KO mES cells; Data are means \pm SD; SD: Standard Deviation; **: $p < 0.01$, two-way ANOVA, the replicates of experiment $n = 5$.

Comment 11: Extended Data Fig. S6f. Fibrillarin (FBL) distribution looks similar between control blastocyst embryos and CX-5461-treated blastocyst embryos. Could the authors explain why?

Response: We thank the reviewer for this good critic. As shown in Fig 5a in this letter, we

have not observed obvious change of protein abundance for FBL between control mES cells and CX-5461 treated mES cells using immunoblotting experiment. However, we have observed that FBL is abnormally distributed with an aggregated pattern in CX-5461 treated mES cells (Fig 20a in this letter and Fig 2d in our new revised manuscript). Indeed, for a proportion of CX-5461 treated blastocyst embryos, we have also observed the distribution and morphology of FBL are similar with CX-5461 treated mES cells (Fig 20b in this letter). The similar FBL distribution between some CX-5461 treated blastocyst embryos and control blastocyst embryos might be due to the cellular heterogeneity in responding to CX-5461 treatment.

Fig 5. a) Immunoblotting showing the expression level of FBL and FBL-mCherry in E14 (WT), E14 expressing FBL-mCherry and CX-5461 treated E14 expressing FBL-mCherry cells; **b)** Immunoblotting showing the expression level of NPM and NPM-eGFP in E14 (WT), E14 expressing NPM-eGFP and CX-5461 treated E14 expressing NPM-eGFP cells; **c)** Immunoblotting showing the expression level of NCL and NCL-mCherry in E14 (WT), E14 expressing NCL-eGFP and CX-5461 treated E14 expressing NCL-eGFP cells.

Fig 20. a) Immunofluorescence staining of FBL in control mES cells and CX-5461 treated

mES cells; b) Immunofluorescence staining of FBL in control blastocyst embryos and CX-5461-treated blastocyst embryos.

Comment 12: I wonder whether CX-5461 treatment can affect the level of LINE1 RNA, which regulates Dux silencing in ESC shown in Ref. 14 (PMID: 29937225).

Response: *We thank the reviewer for this helpful suggestion. According to the reviewer's suggestion, we further analyzed the level of LINE1 RNA in the control mES cells and CX-5461 treated mES cells. We observed that the expression level of LINE1 is significantly downregulated in CX-5461 treated mES cells (Fig 21a in this letter). This obtained result is consistent with the reported role of LINE1 in regulating Dux expression.*

Fig 21. a) Boxplot shows the expression levels of LINE1 under control mES cells and CX-5461 treated mES cells. *n* denotes the number of sub-classes of LINE1 genes, ***, $p < 0.001$, Wilcoxon signed rank test.

Comment 13: Fig1j, "0.4um" is confusing.

Response: *We thank the reviewer for this helpful suggestion. According to the reviewer's suggestion, we corrected this error in our new revised manuscript.*

Comment 14: Fig. 6e. Panel "e" in Figure 6 is missing.

Response: *We thank the reviewer for this helpful suggestion. According to the reviewer's suggestion, we corrected this error in our new revised manuscript.*

Reviewer #3 (Remarks to the Author):

In the present manuscript, Hua Yu and colleagues investigate the effect of nucleolar stress on the reprogramming of mouse embryonic stem cells (mES). In particular, the authors reported that treatment of mES cells with CX-5461, a potent inhibitor of rRNA synthesis, induces transcriptional changes consistent with the transition of mES cells into a 2C

embryo-like (2C-like) cells. 2C-like cells share molecular features of the totipotent 2C-embryo stage. Using different approaches, the authors demonstrated that upon treatment with CX-5461, markers of the 2-cell stage are activated, including Dux and Gm12794. Consistent with these transcriptional changes, the authors observed an expansion in the 2C-like cell population in mES cells. Combining microscopy, ChIP-seq, and Hi-C, the authors linked nuclear stress to changes in 3D chromatin structure and reorganization of PNH, affecting the epigenome of mES and driving the transcriptional reprogramming toward a 2-cell like state. Based on their results, the authors thus concluded that rRNA biogenesis influences the 2C-like program by regulating nuclear phase separation and 3D structure at the PNH. For the most part, the data support the authors' conclusions and provide a novel mechanism contributing to the reprogramming of mES cells. This reviewer appreciated the excellent Hi-C analysis performed by the authors. However, this reviewer's opinion that the data presented not strongly supports Dux's role.

We thank the reviewer for this excellent comment that our findings in this study provide a novel mechanism contributing to the reprogramming of mES cells and appreciate the critics that a few questions about Dux's role need to be addressed to make these findings more convincing.

Comment 1: Data presented in Figure 3c are not very convincing and supportive of Dux's role in binding the genes regulated by CX-5461 treatment. Since the authors showed in Figure 3g that depletion of Dux offsets the transcriptional changes induced by CX-5461, the authors should assess Dux binding and chromatin state of CX-5461 genes after Dux silencing. Does Dux silencing affect chromatin accessibility and deposition of H3K27me3 and H3K9me3 at deregulated genes?

Response: *We thank the reviewer for this helpful and constructive suggestion. According to the reviewer's suggestion, we have further performed ChIP-qPCR experiments under four conditions including control mES cells, CX-5461 treated mES cells, Dux silenced mES cells and Dux silenced & CX-5461 treated mES cells to assess the changes of H3K9me3 & H3K27me3 levels and Dux binding of CX-5461 induced 2C marker genes. We observed that, after Dux silencing, both of H3K9me3 and H3K27me3 are increased, and Dux binding is decreased for CX-5461 induced 2C genes (Fig 22a-22c). When Dux is silenced in the CX-5461-treated mES cells, we observed that H3K9me3 & H3K27me3 levels and Dux binding on CX-5461 induced 2C genes are partially reversed (Fig 22a-22c in this letter). We then further performed ATAC-seq experiment to assess the changes of chromatin accessibility after Dux silencing. In well support with these results, we observed that chromatin accessibility of the CX-5461 induced 2C genes is reversed by Dux silencing (Fig 22d in this letter). Together, these results further confirmed that CX-5461 induced 2C genes activation is regulated by Dux.*

Fig 22. a) ChIP-PCR showing H3K9me3 levels of Dux or 2C-related genes in Dux silenced mES cells; *: $p < 0.05$, **: $p < 0.01$, ***: $p < 0.001$, two-way ANOVA, the replicates of experiment $n=3$, error bar: standard error of mean. **b)** ChIP-PCR showing H3K27me3 levels of Dux or 2C-related genes in Dux silenced mES cells; *: $p < 0.05$, **: $p < 0.01$, ***: $p < 0.001$, two-way ANOVA, the replicates of experiment $n=3$, error bar: standard error of mean. **c)** ChIP-PCR showing DUX protein binding levels on 2C-related genes in Dux silenced mES cells; *: $p < 0.05$, **: $p < 0.01$, ***: $p < 0.001$, two-way ANOVA, the replicates of experiment $n=3$, error bar: standard error of mean. **d)** Line plots demonstrate the meta-analysis results of chromatin accessibility in Dux silenced mES cells within 5kb region around transcription start sites or transcription start and end sites of 621 commonly induced genes between CX-5461 treatment and Dux overexpression and 10173 CX-5461 induced ERV genes using published ATAC-seq data. The regions of different lengths of ERV genes were fitted to 5kb

Comment 2: In Extended Figure S2f, the ChIP-seq tracks do not decrease the indicated markers between control and CX-5461; however, ChIP-seq analysis is not quantitative unless "spick-in" are added to normalized ChIP-seq results. Such limitation should be addressed or at least discussed.

Response: We thank the reviewer for this good and helpful suggestion. Instead of using spick-in to normalize ChIP-seq results, in this study, we employed "Input DNA" to normalize ChIP-seq signals. We are very sorry for the label error of Extended Data Fig S2f in our previous version of manuscript. We have removed "BPM" and corrected it to "ChIP/input" in our new revised manuscript. For ChIP-seq tracks of H3K9me3, H3K27me3, H3K4me3 and H3K27ac, we have selected other genomic regions within PNH to further confirm the observed changes after CX-5461 treatment (Fig 14a-14c in this letter, Fig S2b and Fig S2d in our new revised manuscript). As also well shown in Fig 15 in this letter (Fig 2i-2n and Fig S2a-S2d in our new revised manuscript), both of H3K9me3 and H3K27me3 levels of

PNH are remarkably decreased and the H3K4me3 and H3K27me3 levels of PNH were significantly increased after CX-5461 treatment. For NCL and TRIM28, we have repeated ChIP-seq experiment. Compared with the obtained results of ChIP-seq experiment in our previous version of manuscript, we observed more obvious reduction of NCL and TRIM28 binding on PNH (Fig 16 in this letter, Fig 2o-2q, Fig S2e and S2f in our new revised manuscript). These results fully demonstrated that rRNA biogenesis defect have changed epigenetic state of the heterochromatic regions at the periphery of nucleolus by influencing or impairing the binding of NCL/TRIM28 complex on the PNH.

Fig 14. **a)** UCSC Genome Browser viewing of H3K9me3, H3K27me3, H3K4me3 and H3K27ac ChIP-seq signals in control mES cells and CX-5461 treated mES cells around a representative PNH fragment at chr18:7,718,063-8,074,255 (Extended data Fig S2f in our previous version of manuscript); **b)** UCSC Genome Browser viewing of H3K9me3 and H3K27me3 ChIP-seq signals in control mES cells and CX-5461 treated mES cells around a representative PNH fragment at chr15:4,699,343-5,778,242; **c)** UCSC Genome Browser viewing of H3K4me3 and H3K27ac ChIP-seq signals in control mES cells and CX-5461 treated mES cells around a representative PNH fragment at chr10:3,632,504-3,926,058.

Fig 15. a) Heatmap plots demonstrate the levels of H3K9me3 and H3K27me3 on within 1mb region around start and end sites of Inactive Hub. The regions of different lengths of Inactive Hub fragments were fitted to 1mb. **b)** Heatmap plots demonstrate the levels of H3K9me3 and H3K27me3 on within 1mb region around start and end sites of NAD. The regions of different lengths of NAD fragments were fitted to 1mb. **c)** Heatmap plots demonstrate the levels of H3K9me3 and H3K27me3 on within 1kb region around start and end sites of L1. The regions of different lengths of L1 sequences were fitted to 1kb. **d)** Heatmap plots demonstrate the level of H3K4me3 and H3K27ac on within 1mb region around start and end sites of Inactive Hub. The regions of different lengths of Inactive Hub fragments were fitted to 1mb. **e)** Heatmap plots demonstrate the levels of H3K4me3 and H3K27ac on within 1mb region around start and end sites of NAD. The regions of different lengths of NAD fragments were fitted to 1mb. **f)** Heatmap plots demonstrate the level of H3K4me3 and H3K27ac on within 1kb region around start and end sites of L1. The regions of different lengths of L1 sequences were fitted to 1kb. **g)** Boxplots demonstrate the averaged H3K9me3 and H3K27me3 levels of 101 Inactive Hub fragments, 578 NAD fragments and 34888 L1 sequences; ***: $p < 0.001$, Wilcoxon signed rank test. **h)** Boxplots demonstrate the averaged H3K4me3 and H3K27ac levels of 101 Inactive Hub fragments, 578 NAD fragments and 34888 L1 sequences; ***: $p < 0.001$, Wilcoxon signed rank test.

Figures in our previous version of manuscript

Figures in our current version of manuscript

Fig 16. a) UCSC Genome Browser viewing of NCL and TRIM28 ChIP-seq signals in control mES cells and CX-5461 treated mES cells around a representative PNH fragment at chr18:7,718,063-8,074,255 (Extended data Fig S2f in our previous version of manuscript); **b)** UCSC Genome Browser viewing of NCL and TRIM28 ChIP-seq signals in control and CX-5461 treated mES cells around a representative PNH fragment at chr18:7,718,063-8,074,255 (Extended data Fig S2f in our new revised manuscript); **c)** Heatmap plots demonstrate the binding signals of NCL and TRIM28 on within 1mb region around start and end sites of Inactive Hub and NAD, and within 1kb region around start and end sites of L1. The regions of different lengths of Inactive Hub and NAD fragments were fitted to 1mb. The regions of different lengths of L1 sequences were fitted to 1kb (Extended data Fig S2e in our previous version of manuscript); **d)** Heatmap plots demonstrate the binding signals of NCL and TRIM28 on within 1mb region around start and end sites of Inactive Hub and NAD, and within 1kb region around start and end sites of L1. The regions of different lengths of Inactive Hub and NAD fragments were fitted to 1mb. The regions of different lengths of L1 sequences were fitted to 1kb (Extended data Fig 2o-2q in our new revised manuscript); **e)** Boxplots demonstrate the averaged binding signals of NCL and TRIM28 on 101 Inactive Hub fragments, 578 NAD fragments and 34888 L1 sequences; **: $p < 0.01$, ***: $p < 0.001$, Wilcox signed rank test (Fig 2o and 2p in our previous version of manuscript); **f)** Boxplots demonstrate the averaged binding signals of NCL and TRIM28 on 101 Inactive Hub fragments, 578 NAD fragments and 34888 L1 sequences; **: $p < 0.01$, ***: $p < 0.001$, Wilcox signed rank test (Fig S2e in our new revised manuscript).

References

1. Bulut-Karslioglu, A. et al. Inhibition of mTOR induces a paused pluripotent state. *Nature* **540**, 119-123, doi:10.1038/nature20578 (2016).
2. Hussein, A. M. et al. Metabolic Control over mTOR-Dependent Diapause-like State. *Developmental Cell* **52**, 236-250, doi:10.1016/j.devcel.2019.12.018 (2020).
3. Wu, J. et al. The landscape of accessible chromatin in mammalian preimplantation embryos. *Nature* **534**, 652-657, doi:10.1038/nature18606 (2016).
4. Nicolas, E. et al. Involvement of human ribosomal proteins in nucleolar structure and p53-dependent nucleolar stress. *Nature Communications* **7**, 11390 (2016).
5. Boulon, S., Westman, B.J., Hutten, S., Boisvert, F.M. & Lamond, A.I. The nucleolus under stress. *Molecular Cell* **40**, 216-227 (2010).
6. Savić, N. et al. lncRNA maturation to initiate heterochromatin formation in the nucleolus is required for exit from pluripotency in ESCs. *Cell Stem Cell* **15**, 720-734 (2014).
7. Yang, A. et al. Nucleolin maintains embryonic stem cell self-renewal by suppression of p53 protein-dependent pathway. *Journal of Biological Chemistry* **286**, 43370-43382 (2011).
8. Sanulli, S. et al. HP1 reshapes nucleosome core to promote phase separation of heterochromatin. *Nature* **575**, 390-394 (2019).
9. Strom, A.R. et al. Phase separation drives heterochromatin domain formation. *Nature* **547**, 241-245 (2017).
10. Hnisz, D., Shrinivas, K., Young, R.A., Chakraborty, A.K. & Sharp, P.A. A Phase Separation Model for Transcriptional Control. *Cell* **169**, 13-23 (2017).
11. Hug, C.B. & Vaquerizas, J.M. The Birth of the 3D Genome during Early Embryonic Development. *Trends in Genetics* **34**, 903-914 (2018).
12. Borsos, M. & Torres-Padilla, M.E. Building up the nucleus: nuclear organization in the establishment of totipotency and pluripotency during mammalian development. *Genes and Development* **30**, 611-621 (2016).
13. Guetg, C. & Santoro, R. Formation of nuclear heterochromatin: the nucleolar point of view. *Epigenetics* **7**, 811-814 (2012).
14. Gupta, S. & Santoro, R. Regulation and Roles of the Nucleolus in Embryonic Stem Cells: From Ribosome Biogenesis to Genome Organization. *Stem Cell Reports* **15**, 1206-1219 (2020).
15. Leslie, M. Separation anxiety. *Science* **371**, 336-338, doi:10.1126/science.371.6527.336 (2021).
16. Bhat P, Honson D, Guttman M. Nuclear compartmentalization as a mechanism of quantitative control of gene expression. *Nat Rev Mol Cell Biol*. doi: 10.1038/s41580-021-00387-1 (2021).

17. Lafontaine, D. L. J. & Riback, J. A. The nucleolus as a multiphase liquid condensate. *Nature Reviews Molecular Cell Biology* **22**, 165-182, doi:10.1038/s41580-020-0272-6 (2021).
18. Frottin, F. *et al.* The nucleolus functions as a phase-separated protein quality control compartment. *Science* **365**, 342-347 (2019).
19. Feric, M. *et al.* Coexisting Liquid Phases Underlie Nucleolar Subcompartments. *Cell* **165**, 1686-1697 (2016).
20. Wu M. *et al.* lncRNA SLERT controls phase separation of FC/DFCs to facilitate Pol I transcription. *Science* **373**, 547-555. doi: 10.1126/science.abf6582 (2021).
21. Hendrickson, P. G. *et al.* Conserved roles of mouse DUX and human DUX4 in activating cleavage-stage genes and MERVL/HERVL retrotransposons. *Nature Genetics* **49**, 925-934, doi:10.1038/ng.3844 (2017).
22. Percharde, M. *et al.* A LINE1-Nucleolin Partnership Regulates Early Development and ESC Identity. *Cell* **174**, 391-405.e319, doi:10.1016/j.cell.2018.05.043 (2018).
23. Ishiuchi, T. *et al.* Early embryonic-like cells are induced by downregulating replication-dependent chromatin assembly. *Nature Structural & Molecular Biology* **22**, 662-671, doi:10.1038/nsmb.3066 (2015).
24. Choi, Y. J. & Lin, C. P. Deficiency of microRNA miR-34a expands cell fate potential in pluripotent stem cells. *Science* **355**, 6325 doi:10.1126/science.aag1927 (2017).

REVIEWERS' COMMENTS

Reviewer #1 (Remarks to the Author):

The authors have obviously devoted tremendous amounts of effort to address all reviewers' concerns raised during the initial review. While I am satisfied with the rebuttal to my concern, I am not so pleased in reading the revised manuscript due to the extremely crowded data figures and often ambiguous/unclear statements.

A major question that remains in my mind is how the CX-5461-induced 2CLCs maintain translation functions if rRNA biogenesis or related RiBi was compromised or totally messed up. Don't totipotent stem cells also require normal translational control?

There are several minor points below:

Fig. 2a: "we observed that CX-5461-treated mES cells displayed abnormal nucleolar structure, missing the outer layer usually associated with dense electron intensity". I feel difficult to appreciate this. Can authors try to label or mark directly in the figure panel so readers can better appreciate what has been described? The same applies to Fig. 4a-f panels.

Fig. 2b-c: the ring structure change is not so obvious or significant in my eyes. Any other way to present this in a more quantitative and straightforward fashion, so readers don't have to stare at the images and guess?

Page 9, Line 255, description of Fig. 4a-c. Dux locus is significantly further away.... How to appreciate this without even seeing Dux in those figure panels? There are many occasions that I feel difficult to follow what the authors claimed. I highly recommend authors add some additional text/labels directly on the figure panels for easy reading.

Page 14, line 380-382. It seems to be a broken English sentence with a grammar error. Double-check the sentence.

Page 15, line 416. "It is worth to mention..." is better to be replaced with "It is worth mentioning...".

Reviewer #2 (Remarks to the Author):

The authors addressed most of my comments with a considerable amount of additional data. However, I am still concerned about the novelty of this paper and its suitability for publication in Nature Communications.

Reviewer #3 (Remarks to the Author):

In this revised version of the manuscript, the authors addressed my previous concerns.

Point-by-point responses to reviewers' comments for NCOMMS-21-13807A

Reviewer #1 (Remarks to the Author):

The authors have obviously devoted tremendous amounts of effort to address all reviewers' concerns raised during the initial review. While I am satisfied with the rebuttal to my concern, I am not so pleased in reading the revised manuscript due to the extremely crowded data figures and often ambiguous/unclear statements.

Response: We thank the reviewer for his positive comment to our work and appreciate the critics about insufficient readability of our previous manuscript. In our revised manuscript, we have carefully addressed this problem to make our work easier reading. We have divided the crowded data of Fig. 1, Fig. 2, Fig. 4, Fig. S1 and Fig. S4 of our previous version of manuscript to multiple figures. We have also further revised our figures and manuscript and figures to make our findings more intuitive, quantitative and clearer.

A major question that remains in my mind is how the CX-5461-induced 2CLCs maintain translation functions if rRNA biogenesis or related RiBi was compromised or totally messed up. Don't totipotent stem cells also require normal translational control?

Response: We thank the reviewer for this excellent comment on our current study. As shown in Figure 2c of our previous "Response to Reviewer's letter", the spontaneous 2C::tdTomato positive 2CLCs have more less translation activity than 2C::tdTomato negative mES cells. These results have suggested that 2CLCs are less dependent on translation than mES cells and these cells are in a more quiescent state compared with mES cells. We have treated mES cells with different dosages and treatment times of CX-5461. We found that long-term or high dosage CX-5461 inhibition of rRNA biogenesis leads to mES cell apoptosis (Figure 1a-1d in our previous "Response to Reviewer's letter"). This result have indicated that translation is necessary for both spontaneous 2C::tdTomato positive 2CLCs and 2C::tdTomato negative mES cells. Although the reduced expression level of ribosomal genes, we can still observe the expression of ribosomal genes under CX-5461 treatment. This have suggested that "2 μ M, 12h" CX-5461 treatment did not completely inhibit translation and the translation activity of 2CLCs induced by "2 μ M, 12h" CX-5461 is enough for its survival.

Fig. 2a: "we observed that CX-5461-treated mES cells displayed abnormal nucleolar structure, missing the outer layer usually associated with dense electron intensity". I feel difficult to appreciate this. Can authors try to label or mark directly in the figure panel so readers can better appreciate what has been described? The same applies to Fig. 4a-f

panels.

Response: We thank the reviewer for this good and helpful suggestion. We have marked directly in the figure panel for Fig.2a-2c (Figure 3a-3c in our revised manuscript) and Fig.4a-4f (Figure 6a-6f in our revised manuscript) to facilitate readers better appreciate what has been described.

Fig. 2b-c: the ring structure change is not so obvious or significant in my eyes. Any other way to present this in a more quantitative and straightforward fashion, so readers don't have to stare at the images and guess?

Response: We thank the reviewer for this good and constructive comment. Indeed, this molecular phenotype was observed in 93% CX-5461 treated mES cells (26 out of 28 detected CX-5461 treated mES cells). We added this result to our revised manuscript to more quantitative show the disappearance of the NPM1- and NCL-marked "ring" structure. We have also marked directly in the figure panel of Fig.2b-2c (Figure 3b-3c in our revised manuscript) to more straightforward indicate the disappearance of the NPM1- and NCL-marked "ring" structure.

Page 9, Line 255, description of Fig. 4a-c. Dux locus is significantly further away.... How to appreciate this without even seeing Dux in those figure panels? There are many occasions that I feel difficult to follow what the authors claimed. I highly recommend authors add some additional text/labels directly on the figure panels for easy reading.

Response: We thank the reviewer for this good and helpful suggestion. We have marked directly in the figure panel for Fig.2a (Figure 3a in our revised manuscript) and Fig.4a-4f (Figure 6a-6f in our revised manuscript) to facilitate readers better appreciate what has been described.

Page 14, line 380-382. It seems to be a broken English sentence with a grammar error. Double-check the sentence.

Response: We thank the reviewer for this helpful suggestion. We have checked and revised this sentence to avoid grammar error.

Page 15, line 416. "It is worth to mention..." is better to be replaced with "It is worth mentioning...".

Response: We thank the reviewer for his helpful comment. We have revised the phrase "It is worth to mention" to "It is worth mentioning" in our revised manuscript.

Reviewer #2 (Remarks to the Author):

The authors addressed most of my comments with a considerable amount of additional

data. However, I am still concerned about the novelty of this paper and its suitability for publication in Nature Communications.

Response: We thank the reviewer for the positive comment on our additional data. The main novelty of our current study is that nucleolar integrity maintained by rRNA-mediated liquid-liquid phase separation (LLPS) and 3D chromatin structure of peri-nucleolar heterochromatin (PNH) regulates the fate transition of mouse embryonic stem cells to 2 cell-stage embryo-like cells, and that rRNA biogenesis regulates the 2-cell-to-4-cell transition of murine preimplantation embryo development. Our work provides a potential way in reprogramming mES cells back to 2C-like cells and meanwhile enhances our understanding of nucleolus functions in stem cell fate determination and early development. Thus, we think that the novelty and contribution of our work to this field are suitability for publication in Nature Communications.

Reviewer #3 (Remarks to the Author):

In this revised version of the manuscript, the authors addressed my previous concerns.

Response: We thank the reviewer for his positive comment on our current work.